# Trust Region Constrained Measure Transport in Path Space for Stochastic Optimal Control and Inference

**Denis Blessing**[†, 1] **Julius Berner**[*, 2] **Lorenz Richter**[*, 3, 4] **Carles Domingo-Enrich**[*, 5]
**Yuanqi Du**[6] **Arash Vahdat**[2] **Gerhard Neumann**[1]

[1]Karlsruhe Institute of Technology  [2]NVIDIA  [3]Zuse Institute Berlin  [4]dida
[5]Microsoft Research New England  [6]Cornell University

## Abstract

Solving stochastic optimal control problems with quadratic control costs can be viewed as approximating a target path space measure, e.g. via gradient-based optimization. In practice, however, this optimization is challenging in particular if the target measure differs substantially from the prior. In this work, we therefore approach the problem by iteratively solving constrained problems incorporating trust regions that aim for approaching the target measure gradually in a systematic way. It turns out that this trust region based strategy can be understood as a geometric annealing from the prior to the target measure, where, however, the incorporated trust regions lead to a principled and educated way of choosing the time steps in the annealing path. We demonstrate in multiple optimal control applications that our novel method can improve performance significantly, including tasks in diffusion-based sampling, transition path sampling, and fine-tuning of diffusion models.

## 1 Introduction

Even though the theory of stochastic optimal control (SOC) dates back several decades [12, 49], it has recently attracted renewed interest within the machine learning community. Building on novel formulations that are well-suited for gradient-based optimization (see [40] for an overview) and drawing connections to diffusion models [15, 36, 92], recent work has led to significant progress in the numerical approximation of high-dimensional control problems using neural networks [42, 87]. Related problems are crucial in many practical applications, ranging from sampling problems (e.g., in statistical physics [48, 68], Bayesian statistics [54, 85], and reinforcement learning [24]) to fine-tuning of diffusion models [38, 41, 132]. In this work, we aim to further advance SOC approximation methods by taking inspiration from trust region methods used in optimization [1, 88, 93, 110, 123], resulting in a principled framework from the perspective of measure transport in path space.

**Stochastic optimal control.** SOC problems (with quadratic control costs) describe optimization problems of the form

$$\min_{u \in \mathcal{U}} \mathbb{E}\left[\int_0^T \left(\tfrac{1}{2}\|u\|^2 + f\right)(X_s^u, s)\,\mathrm{d}s + g(X_T^u)\right] \quad \text{with} \quad \begin{cases} \mathrm{d}X_s^u = (b + \sigma u)\,(X_s^u, s)\mathrm{d}s + \sigma(s)\mathrm{d}W_s \\ X_0 \sim p_0, \end{cases} \tag{1}$$

where one optimizes the control $u$ of the stochastic differential equation (SDE). Since the law of the SDE solution $X^u$ induces a so-called *path measure* $\mathbb{P}^u$ on the space of continuous trajectories (specifying how likely a certain trajectory is), finding the optimal control is equivalent to finding an optimal target path space measure $\mathbb{Q}$. From the SOC literature it is known that the likelihood of

---

[†]Correspondence to `denis.blessing@kit.edu`.  [*]Equal contribution.

39th Conference on Neural Information Processing Systems (NeurIPS 2025).

$\mathbb{Q}$ w.r.t. $\mathbb{P}^u$ can be expressed in closed-form (see [34] and (3) below), which allows to minimize divergences[2] $D(\mathbb{P}^u, \mathbb{Q})$ via gradient-based optimization (also termed *iterative diffusion optimization*).

**Trust region methods.** However, if the target $\mathbb{Q}$ is rather different from the initialization $\mathbb{P}^{u_0}$ (typically the uncontrolled process with $u_0 = \mathbf{0}$), many algorithms face challenges with high variances or mode discovery when directly minimizing $D(\mathbb{P}^u, \mathbb{Q})$, especially in high dimensions. To this end, we propose to approach the target measure gradually by a sequence $(\mathbb{P}^{u_i})_i$, where in the $i$-th step we add the constraint $D_{\mathrm{KL}}(\mathbb{P}^u | \mathbb{P}^{u_{i-1}}) \leq \varepsilon$ to the cost functional (1), with $u_{i-1}$ being the approximated optimal control from the previous iteration and $\varepsilon > 0$ a chosen trust region bound. We prove that the intermediate measures $\mathbb{P}^{u_i}$ define a geometric annealing between the prior $\mathbb{P}^{u_0}$ and target measure $\mathbb{Q}$, where the annealing step-sizes are chosen optimally, in the sense of having an approximately constant change in Fisher-Rao distance (Props. 2.2 and 2.3). Finding an optimal annealing schedule is paramount for the convergence speed of many measure transport and sampling methods [119], and understanding physical processes [30, 106]. While the direct computation of Fisher-Rao distances can be challenging, we show that trust region methods lead to a simple way of obtaining equidistant steps in an information-geometric sense. Moreover, we show that the Lagrangian of the constrained problem can be written as another SOC problem and that the optimal Lagrangian multiplier can be obtained via a dual optimization problem without additional computational overhead (Sec. 2.1). Finally, we adapt successful approaches based on SOC matching [41, 42] and log-variance divergences [87] to the constrained SOC problem to get a practical algorithm (Sec. 2.2).

**Applications.** The resulting *trust region stochastic optimal control* method can be viewed as an extension of various existing algorithms, yielding significant improvements on a range of applications (Sec. 3). In particular, we consider (i) deep learning approaches to classical SOC problems (extending [42, 87]) enabling the usage of cross-entropy losses in high dimensions, (ii) diffusion-based sampling from unnormalized densities (extending [97, 129]) enabling efficient sampling from high-dimensional, multimodal densities with substantially fewer target evaluations, (iii) transition path sampling in molecular dynamics (extending [72, 113]) yielding notably higher transition hit rates, and (iv) reward fine-tuning of text-to-image models (extending [41]) achieving comparable performance while requiring significantly fewer simulations.

**Contributions.** Our contributions can be summarized as follows:

- We develop a general framework for solving measure transport with trust regions and apply it to SOC problems using iterative diffusion optimization.
- We prove that our framework leads to a sequence of SOC problems whose solutions define an equispaced annealing between initialization and optimum w.r.t. the Fisher-Rao distance.
- Relying on different loss functionals, we propose two practical instantiations of our framework and demonstrate state-of-the-art performance on a series of applications, ranging from sampling from unnormalized densities to transition path sampling and reward fine-tuning of text-to-image models.

**Notation.** We denote by $\mathcal{U} \subset C(\mathbb{R}^d \times [0, T]; \mathbb{R}^d)$ the set of admissible controls and by $\mathcal{P}$ the set of all probability measures on $C([0, T], \mathbb{R}^d)$. We define the path space measure $\mathbb{P} \in \mathcal{P}$ as the law of a $\mathbb{R}^d$-valued stochastic process $X = (X_t)_{t \in [0,T]}$ and we denote by $\mathbb{P}_s$ the marginal distribution at time $s$. We refer to App. A for further details on our notation and assumptions.

## 2 Trust region constrained measure transport for optimal control

The idea of *iterative diffusion optimization* in optimal control based on path space measures is to consider loss functionals of the form

$$\mathcal{L}(u) = D(\mathbb{P}^u, \mathbb{Q}) \tag{2}$$

and minimize them with gradient-descent algorithms [87]. The loss functional (2) yields implementable algorithms for SOC problems since the optimal path measure $\mathbb{Q}$ of (1) can be stated explicitly via the Radon-Nikodym derivative

$$\frac{\mathrm{d}\mathbb{Q}}{\mathrm{d}\mathbb{P}}(X) = \frac{e^{-\mathcal{W}(X,0)}}{\mathcal{Z}(X_0)} \quad \text{with} \quad \mathcal{W}(X,t) = \int_t^T f(X_s, s)\,\mathrm{d}s + g(X_T), \tag{3}$$

where $\mathcal{Z} := \mathbb{E}\big[e^{-\mathcal{W}(X,0)}|X_0\big]$ and $\mathbb{P}$ is the path measure of the uncontrolled process $X = X^{\mathbf{0}}$; see App. D. In this work, we extend this attempt by using trust regions that shall make sure that the

---

[2]Note that the cost functional (1) corresponds (up to the normalizing constant) to the reverse Kullback-Leibler (KL) divergence $D = D_{\mathrm{KL}}$.

optimization is conducted in a more "regulated" fashion, where the essential idea is to divide the global problem into smaller (reasonably chosen) chunks. We quantify this in Prop. 2.3 below. To this end, we consider the iterative optimization scheme defined by

$$u_{i+1} = \operatorname*{arg\,min}_{u \in \mathcal{U}} D_{\mathrm{KL}}\left(\mathbb{P}^u | \mathbb{Q}\right) \quad \text{s.t.} \quad D_{\mathrm{KL}}(\mathbb{P}^u | \mathbb{P}^{u_i}) \leq \varepsilon, \tag{4}$$

for any $i \in \mathbb{N}$, where $\varepsilon > 0$ defines a trust region w.r.t. to the previous control iterate and where we often set $u_0 = \mathbf{0}$ (and thus $\mathbb{P}^{u_0} = \mathbb{P}$). This corresponds to dividing the overall optimization problem into parts according to their distance measured in the KL divergence between the respective preceding and succeeding path measures. Due to the convexity of the KL divergence, we can show that in all but the last step we actually have an equality constraint in (4); see App. E.1. Thus, there exists an $I \in \mathbb{N}$ such that $u_I = u^*$ is the optimal control of the global control problem defined in (1).

**Remark 2.1** (Controlling the variance of importance weights). The constraint $D_{\mathrm{KL}}(\mathbb{P}^u | \mathbb{P}^{u_i}) \leq \varepsilon$ can be motivated by the goal to control the variance of importance weights $\mathrm{Var}_{\mathbb{P}^{u_i}}(\mathrm{d}\mathbb{P}^{u_{i+1}}/\mathrm{d}\mathbb{P}^{u_i})$, which can be explained by the inequality $\mathrm{Var}_{\mathbb{P}^{u_i}}(\mathrm{d}\mathbb{P}^{u_{i+1}}/\mathrm{d}\mathbb{P}^{u_i}) \geq e^{D_{\mathrm{KL}}(\mathbb{P}^{u_{i+1}} | \mathbb{P}^{u_i})} - 1$, see, e.g., [60]. For small $\varepsilon$ (which is a common choice in practice) we typically observe $\mathrm{Var}_{\mathbb{P}^{u_i}}(\mathrm{d}\mathbb{P}^{u_{i+1}}/\mathrm{d}\mathbb{P}^{u_i}) \approx 2\varepsilon$ (see App. I.3), which can be explained by a Taylor expansion and assuming that $\mathrm{d}\mathbb{P}^{u_{i+1}}/\mathrm{d}\mathbb{P}^{u_i} \approx 1$. Low variance of importance weights is directly related to efficiency of many measure transport methods and too high variance makes it practically impossible to obtain reliable results. Note also that the reverse KL divergence allows for explicit expressions for the resulting constrained problem (see Sec. 2.1) and we leave alternative divergences for future research.

In practice, under suitable regularity assumptions, we can approach the above constrained optimization problem using a relaxed Lagrangian formalism. To this end, we consider the loss functionals

$$\mathcal{L}_{\mathrm{TR}}^{(i)}(u, \lambda) = D_{\mathrm{KL}}\left(\mathbb{P}^u | \mathbb{Q}\right) + \lambda\left(D_{\mathrm{KL}}(\mathbb{P}^u | \mathbb{P}^{u_i}) - \varepsilon\right), \tag{5}$$

where $\lambda \geq 0$ is a Lagrange multiplier, and solve the saddle point problems

$$\max_{\lambda \geq 0} \min_{u \in \mathcal{U}} \mathcal{L}_{\mathrm{TR}}^{(i)}(u, \lambda). \tag{6}$$

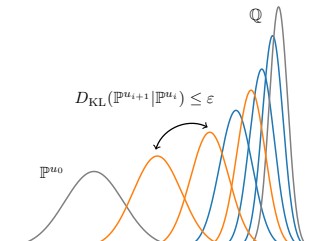

We note that $\mathcal{L}_{\mathrm{TR}}^{(i)}$ is convex in $u$ by convexity of the KL divergence (see App. E.1) and concave in $\lambda$ since it can be expressed as the pointwise minimum $\min_u \mathcal{L}_{\mathrm{TR}}^{(i)}(u, \lambda)$ among a family of linear functions of $\lambda$. Thus, (6) has unique optima which we denote by $u_{i+1}$ and $\lambda_i$, respectively. We can now show the following evolution of the optimal measures.

Figure 1: Illustration of a sequence of distributions $(\mathbb{P}^{u_i})_i$ resulting from our trust region method (orange) and a measure transport corresponding to non-equispaced geometric annealing (blue), leading to high variance in the importance weights for the initial steps.

**Proposition 2.2** (Optimal change of measure as geometric annealing). *Let $\mathbb{Q}$ be the optimal path measure defined in* (3). *The intermediate optimal path measures corresponding to* (4) *then satisfy*[3]

$$\frac{\mathrm{d}\mathbb{P}^{u_{i+1}}}{\mathrm{d}\mathbb{P}^{u_i}} \propto \left(\frac{\mathrm{d}\mathbb{Q}}{\mathrm{d}\mathbb{P}^{u_i}}\right)^{\frac{1}{1+\lambda_i}} \tag{7}$$

*and the optimal change of measure w.r.t. the base measure $\mathbb{P}$ is given by*[4]

$$\frac{\mathrm{d}\mathbb{P}^{u_i}}{\mathrm{d}\mathbb{P}}(X) \propto \left(\frac{\mathrm{d}\mathbb{Q}}{\mathrm{d}\mathbb{P}}(X)\right)^{\beta_i} \left(\frac{\mathrm{d}\mathbb{P}^{u_0}}{\mathrm{d}\mathbb{P}}(X)\right)^{1-\beta_i} \quad \text{with} \quad \beta_i = 1 - \prod_{j=0}^{i-1} \frac{\lambda_j}{1+\lambda_j}. \tag{8}$$

*Proof.* The first statement follows by the definition of the Lagrangian and the second follows by induction; see App. B. $\qquad\square$

Note that the sequence $(\beta_i)_i$ is monotonically increasing with values in $[0, 1]$, where we have $\beta_0 = 0$ and $\beta_I = 1$ (as $\lambda_{I-1} = 0$ due to optimality). Thus, the formula in (8) can be seen as a geometric annealing from the prior to the target measure. Note that when $u_0 = \mathbf{0}$, the second factor vanishes. Importantly, the step-size of the annealing is automatically chosen such that we obtain a well-behaved sequence of distributions; see also Fig. 1.

**Proposition 2.3** (Equidistant steps on statistical manifold). *Up to higher order terms in $\varepsilon$, the sequence of measures $\mathbb{P}^{u_i}$, $i \in \{0, \dots, I-1\}$, are equispaced in the Fisher-Rao distance.*

---

[3]For notational convenience we assume an $X_0$-independent normalizing constant here and hereafter, which is possible whenever the optimal tilting of the initial density $p_0$ is known, cf. App. D.3.

[4]As usual, the empty product is defined as 1 such that $\beta_0 = 0$.

*Proof.* By Prop. 2.2, we obtain $\varepsilon = D_{\mathrm{KL}}(\mathbb{P}^{u_{i+1}}|\mathbb{P}^{u_i}) = \frac{\Delta_i^2}{2}\mathcal{I}(\beta_i) + O(\Delta_i^3)$, where $\Delta_i = \beta_{i+1} - \beta_i$ and $\mathcal{I}(\beta_i)$ is the Fisher information. The Fisher-Rao distance between $\mathbb{P}^{u_i}$ and $\mathbb{P}^{u_{i+1}}$ is then given by $\int_{\beta_i}^{\beta_{i+1}} \sqrt{\mathcal{I}(\tau)}\,\mathrm{d}\tau = \sqrt{\mathcal{I}(\beta_i)}\Delta_i + O(\Delta_i^2) = \sqrt{2\varepsilon} + O(\Delta_i^{3/2})$; see App. F for details. $\square$

**Remark 2.4** (Trust regions for general measures). The observant reader has likely noticed that so far all our arguments do not rely on the fact that we consider path space measures, but work for general probability measures. We could therefore as well write our trust region method stated in (4) as

$$\mathbb{P}_{i+1} = \arg\min_{\mathbb{P}\in\mathcal{P}} D_{\mathrm{KL}}(\mathbb{P}|\mathbb{Q}) \quad \text{s.t.} \quad D_{\mathrm{KL}}(\mathbb{P}|\mathbb{P}_i) \leq \varepsilon. \tag{9}$$

We refer to App. H for a treatment when the measures admit densities on $\mathbb{R}^d$, which can, e.g., be considered for variational inference with normalizing flows.

## 2.1 Constrained stochastic optimal control

While the above formulation in principle works for arbitrary measures, in this work we focus on path space measures corresponding to optimal control problems. In this setting we can compute some of the objectives more explicitly and recover helpful relations.

**Lagrangian as SOC problem.** First, note that, using the Girsanov theorem (see App. A.2), it turns out that, for a fixed Lagrange multiplier $\lambda$, the Lagrangian in (5) defines another SOC problem, i.e.,

$$\mathcal{L}_{\mathrm{TR}}^{(i)}(u,\lambda) = \mathcal{L}_{\mathrm{TRC}}^{(i)}(u,\lambda) - \lambda\varepsilon, \tag{10}$$

where[5]

$$\mathcal{L}_{\mathrm{TRC}}^{(i)}(u,\lambda) = \mathbb{E}\left[\int_0^T \left(\tfrac{1+\lambda}{2}\|u - \tfrac{\lambda}{1+\lambda}u_i\|^2 + \tfrac{\lambda}{2(1+\lambda)}\|u_i\|^2 + f\right)(X_s^u, s)\,\mathrm{d}s + g(X_T^u) + \log\mathcal{Z}(X_0)\right] \tag{11}$$

and $X^u$ is still defined as in (1); see App. E.4 for details. Note that this cost functional is more general than the one stated in (1), which one recovers when setting $\lambda = 0$. We can show that the corresponding SOC problem satisfies the following optimality conditions.

**Proposition 2.5** (Optimality for trust region SOC problems). *For fixed $\lambda$, let us define by*

$$V_{i+1}^\lambda(x,t) := \inf_{u\in\mathcal{U}} \mathbb{E}\left[\int_t^T \left(\tfrac{1+\lambda}{2}\|u - \tfrac{\lambda}{1+\lambda}u_i\|^2 + \tfrac{\lambda}{2(1+\lambda)}\|u_i\|^2 + f\right)(X_s^u, s)\,\mathrm{d}s + g(X_T^u)\bigg|X_t = x\right]$$

*the value function of the SOC problem $\inf_{u\in\mathcal{U}}\mathcal{L}_{\mathrm{TRC}}^{(i)}(u,\lambda)$ corresponding to (11) and by $u_{i+1}^\lambda$ its solution. Then it holds*

*(i) (Estimator for value function)* $V_{i+1}^\lambda(x,t) = -(1+\lambda)\log\mathbb{E}\left[e^{-\frac{1}{1+\lambda}\mathcal{W}_i(X^{u_i},t)}\Big|X_t^{u_i} = x\right]$,

*where* $\mathcal{W}_i(X^{u_i},t) = \int_t^T \tfrac{1}{2}\|u_i(X_s^{u_i},s)\|^2\mathrm{d}s + \int_t^T u_i(X_s^{u_i},s)\cdot\mathrm{d}W_s + \mathcal{W}(X^{u_i},t)$.

*(ii) (Connection between solution and value function) It holds* $u_{i+1}^\lambda = \tfrac{\lambda}{1+\lambda}u_i - \tfrac{1}{1+\lambda}\sigma^\top\nabla V_{i+1}^\lambda$.

*Proof.* The statements can be proven using the verification theorem; see App. E.4 for details. $\square$

We note that Prop. 2.2, the Girsanov theorem, and (3) relate the functional $\mathcal{W}_i$ in Prop. 2.5 to the importance weights

$$\frac{\mathrm{d}\mathbb{Q}}{\mathrm{d}\mathbb{P}^{u_i}}(X^{u_i}) \propto e^{-\mathcal{W}_i(X^{u_i},0)} \quad \text{and} \quad \frac{\mathrm{d}\mathbb{P}^{u_{i+1}}}{\mathrm{d}\mathbb{P}^{u_i}}(X^{u_i}) \propto e^{-\frac{1}{1+\lambda_i}\mathcal{W}_i(X^{u_i},0)}. \tag{12}$$

**Dual problem for Lagrange multiplier.** Next, we will outline how to find the optimal Lagrange multiplier $\lambda$ in (6) in the SOC setting. Plugging the optimal control $u_{i+1}^\lambda$ in the Lagrangian (10) yields the dual function $\mathcal{L}_{\mathrm{Dual}}^{(i)} \in C(\mathbb{R},\mathbb{R})$ given by

$$\mathcal{L}_{\mathrm{Dual}}^{(i)}(\lambda) := \mathcal{L}_{\mathrm{TR}}^{(i)}(u_{i+1}^\lambda,\lambda) = \mathcal{L}_{\mathrm{TRC}}^{(i)}(u_{i+1}^\lambda) - \lambda\varepsilon. \tag{13}$$

We note that evaluating the SOC problem in (11) at the optimal control can be expressed via the value function given in Prop. 2.5, which yields

$$\mathcal{L}_{\mathrm{Dual}}^{(i)}(\lambda) = \mathbb{E}\left[V_{i+1}^\lambda(X_0^{u_i},0)\right] - \lambda\varepsilon = -(1+\lambda)\mathbb{E}\left[\log\mathbb{E}\left[e^{-\frac{1}{1+\lambda}\mathcal{W}_i(X^{u_i},0)}\Big|X_0^{u_i}\right]\right] - \lambda\varepsilon, \tag{14}$$

---

[5]The SOC problem is slightly more general than (1) due to the shift in the quadratic cost.

**Algorithm 1** Trust Region SOC with buffer (see App. E.2 for details)

---

**Require:** Initial path measure $\mathbb{P}^{u_0}$, target path measure $\mathbb{Q}$, divergence $D$, termination threshold $\delta$

    **for** $i = 0, 1, \ldots$ **do**

        Sample trajectories $X \sim \mathbb{P}^{u_i}$ by integrating the SDE in (1) with Brownian motion $W$ and control $u_i$

        Compute importance weights $w = \frac{\mathrm{d}\mathbb{Q}}{\mathrm{d}\mathbb{P}^{u_i}}(X^{u_i}) \propto \exp(-\mathcal{W}_i(X^{u_i}, 0))$ as in (12)

        Initialize buffer $\mathcal{B} = \{W, X, w\}$

        Compute multiplier $\lambda_i = \arg\max_{\lambda \in \mathbb{R}^+} \mathcal{L}_{\mathrm{Dual}}^{(i)}(\lambda)$ as in (14) using $\mathcal{B}$ and a 1-dim. non-linear solver

        Compute $u^{i+1} = \arg\min_u D(\mathbb{P}^u, \mathbb{P}^{u_{i+1}})$ using $\mathcal{B}$ and $\frac{\mathrm{d}\mathbb{P}^{u_{i+1}}}{\mathrm{d}\mathbb{P}^u} \propto w^{\frac{1}{1+\lambda_i}} \frac{\mathrm{d}\mathbb{P}^{u_i}}{\mathrm{d}\mathbb{P}^u}$ as in Sec. 2.2

        **if** $\lambda_i \leq \delta$ **then**

            **return** control $u_{i+1}$ with $\mathbb{P}^{u_{i+1}} \approx \mathbb{Q}$

---

where we note that the expression in the expectation is proportional to the importance weights in (12). Note that we can obtain a Monte Carlo estimate of the dual function using only simulations $X^{u_i}$ from the previous iterations. As it turns out, these simulations are in most cases already required when learning the control $u_{i+1}$ and we can thus store them in a buffer. We can then obtain $\lambda_i = \arg\max_{\lambda \in \mathbb{R}^+} \mathcal{L}_{\mathrm{Dual}}^{(i)}(\lambda)$ using any non-linear solver with minimal computational overhead.

In theory, we can define $u_{i+1} = u_{i+1}^{\lambda_i}$ using the representations in Prop. 2.5 and proceed with the next iteration of our trust region method in (4). However, computing the optimal control $u_{i+1}$ using the representations in Prop. 2.5 requires gradients and Monte Carlo estimators of the value functions. This is problematic since it relies on a large amount of samples *for each state* $x$ due to the (typically) very high variance of the estimator; see App. C for details. Thus, we propose versions of iterative diffusion optimization to learn parametrized approximations to $u_{i+1}$ in the next section.

## 2.2 Learning the constrained optimal control

In this section we propose strategies to learn the optimal control for each iteration. As before, the general idea is to minimize loss functionals based on divergences between path space measures, namely $\mathcal{L}(u) = D(\mathbb{P}^u, \mathbb{P}^{u_{i+1}})$. Such divergences often rely on the Radon-Nikodym derivative

$$
\begin{aligned}
\frac{\mathrm{d}\mathbb{P}^{u_{i+1}}}{\mathrm{d}\mathbb{P}^u}(X^{u_i}) &= \frac{\mathrm{d}\mathbb{P}^{u_{i+1}}}{\mathrm{d}\mathbb{P}^{u_i}}(X^{u_i}) \frac{\mathrm{d}\mathbb{P}^{u_i}}{\mathrm{d}\mathbb{P}^u}(X^{u_i}) \\
&\propto \exp\left( \int_0^T \tfrac{\|u_i - u\|^2}{2}(X_s^{u_i}, s)\mathrm{d}s + \int_0^T (u_i - u)(X_s^{u_i}, s) \cdot \mathrm{d}W_s - \tfrac{\mathcal{W}_i(X^{u_i}, 0)}{1 + \lambda_i} \right),
\end{aligned} \tag{15}
$$

where we used Girsanov's theorem and (12). Note that the Radon-Nikodym derivative in (15) depends only on samples of the process with the already learned $u_i$. Let us now suggest two concrete divergences. Those divergences are desirable for high-dimensional problems since both do not rely on computing derivatives of the stochastic process and can be optimized "off-policy" using trajectories $X^{u_i}$ with the control $u_i$ of the previous iteration, which can be stored in a buffer; see Algorithm 1.

**Log-variance divergence.** This divergence can be considered w.r.t. an arbitrary reference measure, where we choose $\mathbb{P}^{u_i}$ for convenience [87, 98]. We can then define the loss functional

$$
\mathcal{L}_{\mathrm{LV}}(u) := \mathrm{Var}\left[ \log\left( \frac{\mathrm{d}\mathbb{P}^{u_{i+1}}}{\mathrm{d}\mathbb{P}^u}(X^{u_i}) \right) \right], \tag{16}
$$

where the Radon-Nikodym derivative can be explicitly computed as in (15). Note that for $\lambda_i = 0$, this loss reduces to the on-policy log-variance loss typically used in the literature [97]. While this loss has beneficial theoretical properties [87], it requires to keep the full trajectory in memory for the gradient computation.

**Cross-entropy divergence and SOC matching.** Alternatively, we can consider the cross-entropy loss (i.e., the forward KL divergence computed using reweighting)

$$
\mathcal{L}_{\mathrm{CE}}(u) := D_{\mathrm{KL}}(\mathbb{P}^{u_{i+1}} | \mathbb{P}^u) = \mathbb{E}\left[ \left( \log \frac{\mathrm{d}\mathbb{P}^{u_{i+1}}}{\mathrm{d}\mathbb{P}^u}(X^{u_i}) \right) \frac{\mathrm{d}\mathbb{P}^{u_{i+1}}}{\mathrm{d}\mathbb{P}^{u_i}}(X^{u_i}) \right], \tag{17}
$$

where the Radon-Nikodym derivative is again given by (15). Contrary to the log-variance loss, the reweighting $\frac{\mathrm{d}\mathbb{P}^{u_{i+1}}}{\mathrm{d}\mathbb{P}^{u_i}}$ in (12) induces exponential terms. Our trust region constraint makes sure, however, that the variance of those weights stays bounded, see Remark 2.1.

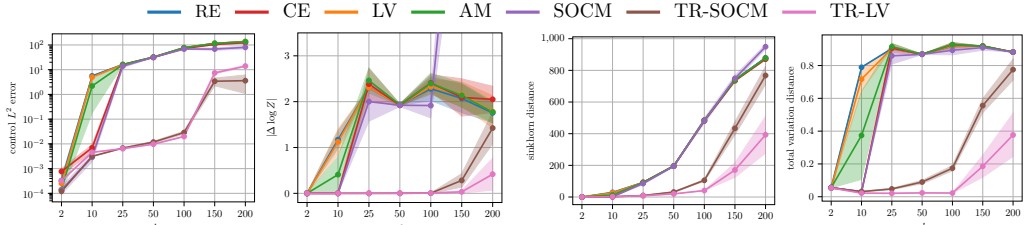

Figure 2: Performance criteria for a Gaussian mixture target density with varying dimension $d$, averaged across four seeds. We show the errors of estimating the optimal control, the log-normalizing constant, as well as the Sinkhorn and total variation distances over different dimensions (from left to right). We observe that our trust region methods (TR-SOCM and TR-LV) are the only methods that perform well in high dimensions.

To efficiently compute this loss, we define the so-called (*lean*[6]) *adjoint state* $a$ as in [41] via

$$\frac{\mathrm{d}}{\mathrm{d}s} a_{i+1}(X_s, s) = - \left[ (\nabla b(X_s, s)^\top a_{i+1}(X_s, s) + \beta_{i+1} \nabla f(X_s, s) \right] \tag{18}$$

with $a_{i+1}(X_T, T) = \beta_{i+1} \nabla g(X_T)$, satisfying $a_{i+1}(X_s, s) = \nabla_{X_s} \beta_{i+1} \mathcal{W}(X, s)$; see [41, Lemma 5] and observe that it differs from the standard lean adjoint by the factor $\beta_i$ defined in Prop. 2.2. Similar to [42], we can use the expression for the optimal control in Prop. 2.5 and the Girsanov theorem to arrive at the *SOC matching loss*[7], a simple regression objective given by

$$\mathcal{L}_{\mathrm{SOCM}}(u) := \mathbb{E} \left[ \frac{1}{2} \int_0^T \| \sigma^\top a_{i+1}(X_s^{u_i}, s) - u(X_s^{u_i}, s) \|^2 \mathrm{d}s \, \frac{\mathrm{d}\mathbb{P}^{u_{i+1}}}{\mathrm{d}\mathbb{P}^{u_i}} (X^{u_i}) \right], \tag{19}$$

see App. G.5 for details. Contrary to the log-variance divergence above, this objective does not require to keep the whole trajectory $X^{u_i}$ in memory for backpropagation but can be computed at times $t \sim \mathrm{Unif}([0, T])$ using a Monte Carlo approximation. We summarize our algorithm in (1) and compare the different losses against existing approaches for SOC problems in the next section.

## 3 Applications

In this section, we explore several applications of SOC, comparing our novel trust-region-based optimization algorithm against existing methods. Specifically, we consider the three tasks sampling from unnormalized densities, transition path sampling, and fine-tuning text-to-image models. For background information, detailed experimental setups, and additional results, we refer to Apps. I to K, respectively. We also include further experiments on classical SOC problems in App. L.

### 3.1 Diffusion-based sampling

Using (3), we can show that sampling problems can be reformulated as SOC problems. To this end, we leverage the following corollary showing that the terminal distributions $\mathbb{Q}_T$ and $\mathbb{P}_T$ of the optimally controlled and uncontrolled processes differ by a tilting.

**Corollary 3.1** (Sampling from tilted distributions). *Let us set $f = 0$ and assume that the terminal distribution of the uncontrolled process $X$ is independent of $p_0$ and admits a density denoted by $\mathbb{P}_T$. Then it holds that $\mathbb{Q}_T \propto \mathbb{P}_T e^{-g}$.*

*Proof.* Using (3) it holds that $\frac{\mathrm{d}\mathbb{Q}}{\mathrm{d}\mathbb{P}}(X) = \frac{e^{-g(X_T)}}{\mathcal{Z}(X_0)}$ with $\mathcal{Z}(X_0) = \mathbb{E} \left[ e^{-g(X_T)} | X_0 \right]$. The results follows from the independence of $X_T$ and $X_0$; see [41] and App. I for details. $\square$

Cor. 3.1 shows that the optimally controlled process $X^{u^*}$ samples from a given unnormalized density $\rho_{\mathrm{target}}$ when using an uncontrolled process with known terminal distribution $\mathbb{P}_T$ and setting $g = \log \frac{\mathbb{P}_T}{\rho_{\mathrm{target}}}$; see [33, 95, 97, 125, 129–131, 144, 149] and App. I for details. Such sampling problems are of immense practical interest, with numerous applications in the natural sciences [109, 151], in Bayesian statistics [54], and reinforcement learning [24].

**Numerical experiments.** Here, we compare existing methods for solving SOC problems with our trust region method on challenging multimodal sampling problems. We use the *Denoising Diffusion Sampler* (DDS) [129] method, which leverages an ergodic Ornstein–Uhlenbeck process initialized at

---

[6]Instead of the uncontrolled process $X$, we could also express the adjoint state w.r.t. the process $X^u$; however, this relies on more costly vector-Jacobian products; see App. G.4.

[7]The loss is similar to the SOCM-Adjoint loss in [42], which, however, involves matrix-valued functions.

|        | $d = 2$ | $d = 50$ | $d = 100$ | $d = 200$ |
|--------|---------|----------|-----------|-----------|
| RE     | $1.364_{\pm 0.002}$ | $3.443_{\pm 0.004}$ | $3.077_{\pm 0.669}$ | $2.908_{\pm 0.679}$ |
| CE     | $0.001_{\pm 0.000}$ | $0.202_{\pm 0.159}$ | $0.526_{\pm 0.181}$ | $0.641_{\pm 0.527}$ |
| LV     | *diverged* | $1.363_{\pm 0.325}$ | $1.809_{\pm 0.737}$ | $1.958_{\pm 0.698}$ |
| AM     | $1.364_{\pm 0.002}$ | $3.432_{\pm 0.020}$ | $3.457_{\pm 0.019}$ | $3.322_{\pm 0.307}$ |
| SOCM   | $0.001_{\pm 0.000}$ | $2.958_{\pm 0.831}$ | $2.971_{\pm 0.846}$ | $3.504_{\pm 0.005}$ |
| TR-LV  | $\mathbf{0.000_{\pm 0.000}}$ | $\mathbf{0.000_{\pm 0.000}}$ | $\mathbf{0.002_{\pm 0.002}}$ | $\mathbf{0.002_{\pm 0.001}}$ |

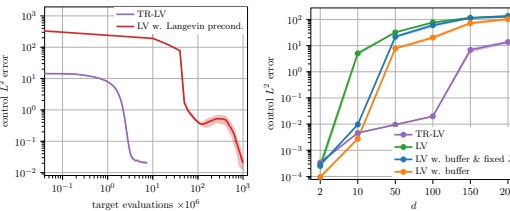

Figure 3: The left table reports $|\Delta \log \mathcal{Z}|$ values for the *Many Well* target across different dimensions $d$. The middle plot compares the log-variance loss of our trust region method (TR-LV) with that of Langevin preconditioning on the GMM target in dimension $d = 100$. The rightmost figure presents an ablation analysis of key components in our method, highlighting the importance of trust regions in preventing mode collapse and achieving low control error. All results are averaged across four seeds.

its equilibrium measure as uncontrolled process $X$. We consider five baselines, specifically, reverse and (importance weighted) forward KL, also known as *relative entropy (RE)* and *cross entropy (CE)* method, respectively. Additionally, we consider the *log-variance loss [98]*, *adjoint matching (AM)* [41], and *stochastic optimal control matching (SOCM)* [42], for the unconstrained problem in (2); see [40] for a comprehensive overview of SOC losses. In all experiments, we deliberately avoid using gradient guidance from the target density in the diffusion process, often referred to as Langevin preconditioning (LP) [66]. Prior work has shown that LP is essential for preventing mode collapse in neural samplers [18, 66]. However, LP is computationally expensive, as it requires querying the target distribution at every discretization step, making such approaches impractical for many problems where evaluating the target gradient is costly.

First, we consider a *Gaussian Mixture Model (GMM)* comprising 10 components and randomized mixing weights. GMMs are particularly compelling as they admit an analytical solution for the optimal control, which enables direct computation of the $L^2$ error between the learned and optimal controls, a reliable metric for detecting mode collapse. In addition, we assess the *Sinkhorn distance* [31] between samples from the target and the model, and the absolute error in estimating the log-normalizing constant, denoted $|\Delta \log \mathcal{Z}|$. Finally, we evaluate the *total variation distance* between the true mixing weights and the model's estimated weights. The results, shown in Fig. 2, indicate that for $d = 2$, all methods closely approximate the optimal control. However, for dimensions beyond $d = 10$, most methods suffer from mode collapse, as reflected by increased control errors, except for those employing trust region updates. Trust region methods maintain robustness across a wide range of dimensions and only begin to show signs of mode collapse in high dimensions ($d \geq 150$).

We additionally evaluate our method on the *Many Well* target [135] with 32 modes. For quantitative analysis, we report the log-normalization error $|\Delta \log \mathcal{Z}|$, as other ground-truth quantities are unavailable. Additionally, for the high-dimensional case $d = 200$, we visualize pairs of marginal distributions in App. I. The results, presented in Fig. 3, demonstrate that our method significantly outperforms competing approaches in estimating the normalizing constant. Furthermore, the visualizations in App. I illustrate that trust region updates effectively prevent mode collapse, even in high dimensions. In contrast, baseline methods either suffer from mode collapse or fail to converge.

Finally, we perform an ablation study on the GMM target, analyzing key components of our proposed method. Specifically, we investigate the effects of incorporating a replay buffer and applying trust region optimization. To this end, we compare a variant using a fixed Lagrangian multiplier $\lambda$, selected via hyperparameter tuning, with one in which $\lambda$ is dynamically optimized using our trust region approach. Additionally, we evaluate the log-variance loss both with and without using a replay buffer. Moreover, we compare our method to LV with Langevin preconditioning on the GMM target with dimensionality $d = 100$. The results, shown in Figure 3, demonstrate that trust region optimization significantly reduces control error and decreases the number of target evaluations by several orders of magnitude.

## 3.2 Transition path sampling

Transition path sampling is of great importance for studying phase transitions and chemical reactions. The key challenge comes from the energy barrier that connects two sets $A$ and $B$ along the energy landscape, which makes direct sampling of transition paths extremely unlikely. These problems can also be formulated as SOC problems [59, 62, 115]. Specifically, we set $b = -\nabla U$, where $U : \mathbb{R}^{N \times 3} \to \mathbb{R}$ is the potential function, and $g = -\log \mathbf{1}_B$ as well as $p_0 \propto \mathbf{1}_A$, which constraints the initial and target states in the sets $A$ and $B$. As in (3), it holds that $\frac{d\mathbb{Q}}{d\mathbb{P}} = \frac{\mathbf{1}_B(X_T)}{\mathcal{Z}(X_0)}$. Recent work

Table 1: Quantitative evaluation on transition path sampling problems. † denotes that results are taken from [113]. The results for TPS-DPS and TR-LV are averaged across three seeds.

| Method | RMSD (Å, ↓) | THP (%, ↑) | ETS (kJ/mol) | Method | RMSD (Å, ↓) | THP (%, ↑) | ETS (kJ/mol) |
|---|---|---|---|---|---|---|---|
| | Alanine Dipeptide | | | | Chignolin | | |
| UMD (3600K)† | $1.19 \pm 0.32$ | 6.25 | $812.47 \pm 148.80$ | UMD (1200K)† | $7.23 \pm 0.93$ | 1.56 | 388.17 |
| SMD† | $0.56 \pm 0.27$ | 54.69 | $78.40 \pm 12.76$ | SMD† | $\mathbf{0.85 \pm 0.24}$ | 34.38 | $179.52 \pm 138.87$ |
| PIPS† | $0.66 \pm 0.15$ | 43.75 | $28.17 \pm 10.86$ | PIPS† | $4.66 \pm 0.17$ | 0.00 | – |
| TPS-DPS | $0.47 \pm 0.18$ | $39.58 \pm 28.13$ | $46.34 \pm 10.16$ | TPS-DPS | $1.06 \pm 0.08$ | $25.00 \pm 10.69$ | $-189.91 \pm 23.01$ |
| TR-LV | $\mathbf{0.29 \pm 0.03}$ | $\mathbf{61.25 \pm 4.05}$ | $49.11 \pm 5.84$ | TR-LV | $0.90 \pm 0.01$ | $\mathbf{43.95 \pm 5.64}$ | $-303.98 \pm 28.65$ |

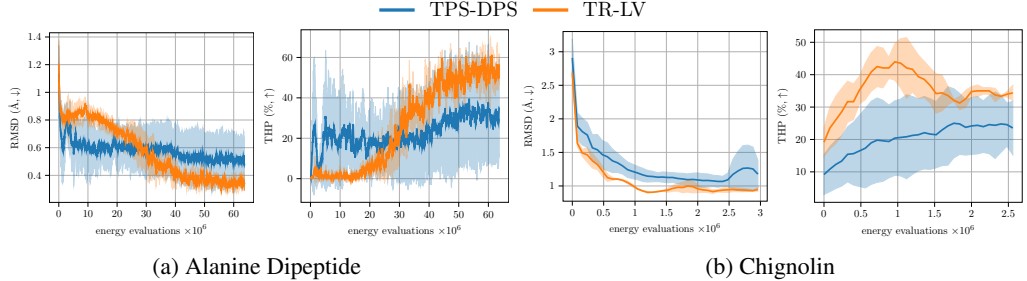

(a) Alanine Dipeptide  (b) Chignolin

Figure 4: We compare our trust region method (TR-LV) with Diffusion Path Sampler (TPS-DPS) [113] on Alanine Dipeptide and Chignolin. All results are averaged over three random seeds, with both the mean and standard deviation reported. Our method identifies transition paths more consistently and robustly, as evidenced by higher THP values and lower standard deviations.

has leveraged neural networks to parameterize a bias force to solve the corresponding SOC problem, employing objectives such as the KL [44, 72, 141], or log-variance divergence [113].

**Numerical experiments.** We evaluate the performance of the trust-region-based log-variance loss (TR-LV) on two transition path sampling problems: Alanine Dipeptide isomerization and Chignolin folding, with 22 and 138 atoms, respectively.

Our evaluation includes three metrics: *Kabsch-aligned root mean squared distance (RMSD)* between the final states of the sampled paths and the target state, *transition hit percentage (THP)* measuring the proportion of final states hitting within the target region, and *energy of transition state (ETS)* identifying the highest energy values along paths that reach the target.

We compare our method to standard molecular dynamics (MD) with increased temperature (UMD), steered MD (SMD) [75] with force applied to collective variables, and PIPS [72] which uses the cross-entropy loss. We also include TPS-DPS [113] as a key baseline, which employs an (unconstrained) log-variance loss to formulate TPS as a stochastic optimal control (SOC) problem. Further experimental details are provided in App. J.

Table 1 shows that TR-LV achieves superior target state RMSD and transition hit percentage compared to the standard log-variance objective (TPS-DPS) for both molecular systems. Notably, SMD performs well due to its use of collective variables with biased force guiding the sampling process. Figure 4 illustrates that the trust region constraint leads to significantly more robust training compared to TPS-DPS as indicated by low standard deviations across different seeds. Moreover, on Alanine Dipeptide, the trust region constraint initially regularizes optimization and accelerates convergence thereafter. Across both systems, the trust region constraint significantly enhances training stability and performance.

### 3.3 Fine-tuning of diffusion models

Interpreting $-g$ as a *reward* and the uncontrolled process $X$ as a pretrained diffusion model (i.e., $b$ includes the pretrained neural network), Cor. 3.1 shows that we can perform reward fine-tuning by solving the SOC problem in (1); see also [38, 41, 132]. Reward fine-tuning has recently shown impressive results, e.g., in image [28, 41] and molecule generation [38], and SOC provides a principled framework. A special case is given by posterior sampling [38]. Setting $g = -\log p(y|x)$, where $p(y|x)$ is the likelihood and we interpret $\mathbb{P}_T$ as a learned (*diffusion*) prior $p(x)$, Bayes' theorem shows that the optimally controlled process samples from the posterior $p(x|y)$.

**Numerical experiments.** We perform reward fine-tuning on Stable Diffusion 1.5 [102], using ImageReward [140], which is a reward model designed to capture prompt alignment and image quality according to human preferences. We take the adjoint matching (AM) method as baseline and

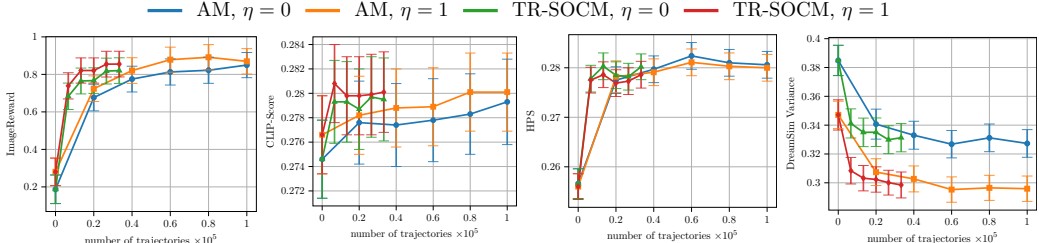

Figure 5: Comparison of Adjoint Matching against Trust Region SOCM for Stable Diffusion 1.5 fine-tuning w.r.t. four quality metrics, where $\eta = 0$ and $\eta = 1$ refer to ODE (DDIM) and SDE (DDPM) inference, respectively.

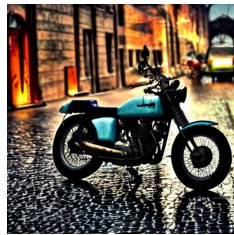 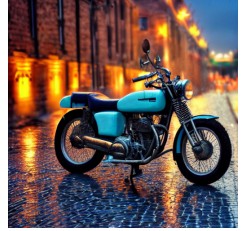 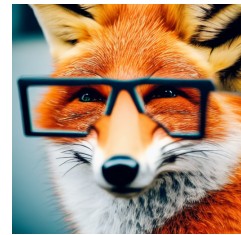 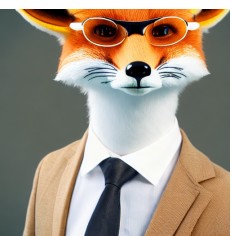

**masterpiece, best quality, realistic photograph, 8k, high detailed vintage motorcycle parked on a wet cobblestone street at dusk, neon reflections, shallow depth of field**

**close up photo of anthropomorphic fox animal dressed in white shirt, fox animal, glasses**

Figure 6: Comparison between images generated by the base Stable Diffusion 1.5 model (left) and its version fine-tuned with TR-SOCM (right), using the same prompts and random seeds. The fine-tuned model generates higher quality images (bike) with better prompt alignment (fox).

compare it against our TR-SOCM loss (19), keeping all other hyperparameters fixed. Our TR-SOCM allows the principled use of buffers, and we perform three passes on each buffer of size $500$, leading to three times fewer trajectories for a fixed number of model updates. For faster convergence, we use a modified version of TR-SOCM with annealing factor $\beta_i = 1$. For each algorithm, we evaluate 5 checkpoints during fine-tuning (with ODE and SDE inference) on ImageReward and three additional metrics: CLIP-Score [69], which measures prompt alignment, Human Preference Score [137], which measures human-perceived image quality, and Dreamsim diversity [51], which measures per-prompt diversity. We observe that TR-SOCM achieves similar performance metrics to AM at a fraction of the cost, as sampling the trajectories and solving the lean adjoint ODE, which dominates the computational costs, is amortized over the buffer passes; see Figs. 5 and 6 as well as App. K for more details.

## 4 Related works

In this section, we discuss the most related works, comparing our approach to existing methods for solving SOC problems. We provide a more extensive comparison in App. C.

**Iterative diffusion optimization.** Many recently developed methods approach SOC problems by simulating the (diffusion) process $X^u$, computing a suitable cost function, and optimizing the parameters of the control function $u$ using variants of stochastic gradient methods. These techniques are collectively referred to as *iterative diffusion optimization* (IDO) methods [87]. While the underlying theory dates back to [33, 91], combinations with deep learning in the context of SOC have been explored by [15, 87, 95, 97, 129, 131, 149, 152]. One can derive most of the related objectives starting from the Radon-Nikodym derivative $\frac{d\mathbb{P}^u}{d\mathbb{Q}}(X^u)$ as in (12) (with $u = u_i$). One can then minimize a loss based on a suitable divergence as in (2). Previous works have, e.g., proposed the log-variance divergence [97, 113] or the forward KL divergence (corresponding to the cross-entropy loss [61, 72, 76, 104, 150]), for which we develop corresponding trust region versions in (16) and (17). The SOC matching loss [42], which we extended to trust regions in (19), is equal to the cross entropy loss in expectation but exhibits lower variance empirically. We refer to [41] for more IDO losses. However, all existing methods have either directly tackled the target measure $\mathbb{Q}$ or relied on a form of hand-tuned annealing.

**Trust region methods.** We show how IDO methods can generally be extended to trust region methods, enabling (1) automatic control on the variance of the importance weights and (2) principled usage of buffers, leading to faster and more stable convergence, in particular avoiding mode collapse

in high dimensions. Trust region methods have a long history as robust optimization algorithms that iteratively minimize an objective within an adaptively sized "trust region"; see [29] for an overview. These methods have also been extended to optimize over spaces of probability distributions, particularly in reinforcement learning [2–4, 7, 83, 88, 90, 93, 110, 111, 138, 139, 142], black-box optimization [1, 118, 134], variational inference [9, 10] and path integral control [123]. To the best of our knowledge, these methods have not yet been extended to path measures or inference problems. Moreover, the connection between trust-region iterates and geometric annealing has not previously been established.

## 5 Conclusion

In this work, we develop a novel framework for solving SOC problems using deep learning. Our framework builds on the fact that we can reformulate specific problems as finding an optimal path space measure induced by a controlled SDE. Instead of finding this optimal measure at once, we divide the unconstrained problem into a sequence of constrained optimization problems by bounding the KL divergence to the measure from the previous iteration. We show that this defines a well-behaved geometric annealing between the prior and the target path measure, resulting in equidistant steps on the Fisher-Rao information manifold. Crucially, each intermediate problem turns out to be an altered SOC problem that can be efficiently solved without simulations by using a buffer of trajectories with the control from the previous iteration. In our experiments, we show that our method significantly improves the learning of the optimal control, including applications in diffusion-based sampling and transition path sampling in molecular dynamics. Further, we show that our method can be scaled to improve the efficiency of reward fine-tuning for text-to-image diffusion models. In the future, we expect our framework to improve even more applications of SOC, potentially including the use of divergences other than the KL divergence for the trust region constraint. Finally, our results for general measures motivate the use of trust region methods for other learned measure transports, e.g., normalizing flows.

## Acknowledgements

D.B. acknowledges support by funding from the pilot program Core Informatics of the Helmholtz Association (HGF) and the state of Baden-Württemberg through bwHPC, as well as the HoreKa supercomputer funded by the Ministry of Science, Research and the Arts Baden-Württemberg and by the German Federal Ministry of Education and Research. The research of L.R. was partially funded by Deutsche Forschungsgemeinschaft (DFG) through the grant CRC 1114 "Scaling Cascades in Complex Systems" (project A05, project number 235221301).

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

# Appendix

# A Assumptions and auxiliary results

## A.1 Additional notation

For vectors $v_1, v_2 \in \mathbb{R}^d$, we denote by $\|v\|$ the Euclidean norm and by $v_1 \cdot v_2$ the Euclidean inner product. For a real-valued matrix $A$, we denote by $\mathrm{Tr}(A)$ and $A^\top$ its trace and transpose.

For a sufficiently smooth function $f \colon \mathbb{R}^d \times [0, T] \to \mathbb{R}$, we denote by $\nabla f = \nabla_x f$ its gradient w.r.t. the spatial variables $x$ and by $\partial_t f$ and $\partial_{x_i} f$ its partial derivatives w.r.t. the time coordinate $t$ and the spatial coordinate $x_i$, respectively.

We denote by $\mathcal{N}(\mu, \Sigma)$ a multivariate normal distribution with mean $\mu \in \mathbb{R}^d$ and covariance matrix $\Sigma \in \mathbb{R}^{d \times d}$. Moreover, we denote by $\mathrm{Unif}([0, T])$ the uniform distribution on $[0, T]$. For random variables $X_1, X_2$, we denote by $\mathbb{E}[X_1]$ and $\mathrm{Var}[X_1]$ the expectation and variance of $X_1$ and by $\mathbb{E}[X_1|X_2]$ the conditional expectation of $X_1$ given $X_2$.

## A.2 Technical assumptions

Throughout our work, we make the same assumptions as in [42, 87], which are needed for all the objects considered to be well-defined. Namely, we assume that:

(i) The set $\mathcal{U}$ of *admissible controls* is given by

$$\mathcal{U} = \{u \in C^1(\mathbb{R}^d \times [0, T]; \mathbb{R}^d) \mid \exists C > 0, \, \forall (x, s) \in \mathbb{R}^d \times [0, T], \, u(x, s) \le C(1 + \|x\|)\}. \quad (20)$$

(ii) The coefficients $b$ and $\sigma$ are continuously differentiable, $\sigma$ has bounded first-order spatial derivatives, and $(\sigma\sigma^\top)(x, s)$ is positive definite for all $(x, s) \in \mathbb{R}^d \times [0, T]$. Furthermore, there exist constants $C, c_1, c_2 > 0$ such that

$$
\begin{aligned}
\|b(x, s)\| &\le C(1 + \|x\|), &\text{(linear growth)} \\
c_1 \|\beta\|^2 &\le \beta^\top (\sigma\sigma^\top)(x, s)\beta \le c_2 \|\beta\|^2, &\text{(ellipticity)}
\end{aligned} \quad (21)
$$

for all $(x, s) \in \mathbb{R}^d \times [0, T]$ and $\beta \in \mathbb{R}^d$.

## A.3 Useful identities

**Definition A.1** (Controlled SDEs). Let $u \in \mathcal{U}$ be a control function. Throughout, we consider controlled and uncontrolled stochastic processes defined via the SDEs

$$
\begin{aligned}
\mathrm{d}X_s^u &= (b + \sigma u)(X_s^u, s)\mathrm{d}s + \sigma(s)\mathrm{d}W_s, &X_0^u &\sim p_0, &(22) \\
\mathrm{d}X_s &= b(X_s, s)\mathrm{d}s + \sigma(s)\mathrm{d}W_s, &X_0 &\sim p_0, &(23)
\end{aligned}
$$

where $X^u \sim \mathbb{P}^u$, $X \sim \mathbb{P}$, with $\mathbb{P}^u$ and $\mathbb{P}$ denoting the respective path space measures, and $W$ is a standard Brownian motion.

**Theorem A.2** (Girsanov's theorem for path measures). *Let $u, v, w \in \mathcal{U}$. Then the Radon-Nikodym derivative between $\mathbb{P}^u$ and $\mathbb{P}^v$, evaluated along $X^w$, is given by:*

$$\log \frac{\mathrm{d}\mathbb{P}^u}{\mathrm{d}\mathbb{P}^v}(X^w) = \int_0^T \sigma^{-1}(u - v)(X_s^w, s) \cdot \mathrm{d}X_s^w - \frac{1}{2}\int_0^T \left(\|\sigma^{-1}b + u\|^2 - \|\sigma^{-1}b + v\|^2\right)(X_s^w, s)\mathrm{d}s, \quad (24)$$

*Proof.* See, e.g., Lemma A.1 in [87] or Appendix E in [131]. $\square$

**Corollary A.3** (Change of measure identities). *Let $u, v \in \mathcal{U}$. From Thm. A.2, we obtain the following useful identities:*

*(i)* $\log \frac{\mathrm{d}\mathbb{P}^u}{\mathrm{d}\mathbb{P}}(X^u) = \int_0^T u(X_s^u, s) \cdot \mathrm{d}W_s + \frac{1}{2}\int_0^T \|u(X_s^u, s)\|^2 \mathrm{d}s$

*(ii)* $\log \frac{\mathrm{d}\mathbb{P}^u}{\mathrm{d}\mathbb{P}}(X) = \int_0^T u(X_s, s) \cdot \mathrm{d}W_s - \frac{1}{2}\int_0^T \|u(X_s, s)\|^2 \mathrm{d}s$

*(iii)* $\log \frac{\mathrm{d}\mathbb{P}^u}{\mathrm{d}\mathbb{P}^v}(X^u) = \int_0^T (u - v)(X_s^u, s) \cdot \mathrm{d}W_s + \frac{1}{2}\int_0^T \|u - v\|^2(X_s^u, s)\mathrm{d}s$

*(iv)* $\log \frac{\mathrm{d}\mathbb{P}^u}{\mathrm{d}\mathbb{P}^v}(X^v) = \int_0^T (u - v)(X_s^v, s) \cdot \mathrm{d}W_s - \frac{1}{2}\int_0^T \|u - v\|^2(X_s^v, s)\mathrm{d}s$

**Lemma A.4** (Itô's formula). *Let $X_s$ solve the SDE*

$$\mathrm{d}X_s = b(X_s, s)\mathrm{d}s + \sigma(s)\mathrm{d}W_s,$$

*and let $f \colon \mathbb{R}^d \times [0, T] \to \mathbb{R}$ be a smooth function. Then*

$$\mathrm{d}f(X_s, s) = (\partial_s + L)f(X_s, s)\mathrm{d}s + \sigma^\top \nabla f(X_s, s) \cdot \mathrm{d}W_s,$$

*where L is the infinitesimal generator given by*

$$L := \frac{1}{2} \sum_{i,j=1}^{d} (\sigma\sigma^\top)_{ij} \partial_{x_i}\partial_{x_j} + \sum_{i=1}^{d} b_i(x,t)\partial_{x_i}.$$

# B  Proofs

*Proof of Prop. 2.2.* Let $\widetilde{\mathbb{P}}$ be the measure defined by $\frac{d\widetilde{\mathbb{P}}}{d\mathbb{P}^{u_i}} = \left(\frac{d\mathbb{Q}}{d\mathbb{P}^{u_i}}\right)^{\frac{1}{1+\lambda_i}}/\widetilde{\mathcal{Z}}$, where $\widetilde{\mathcal{Z}}$ is the normalizing constant. Then we have that

$$(1+\lambda_i)\log\frac{d\mathbb{P}^u}{d\widetilde{\mathbb{P}}} = (1+\lambda_i)\log\left(\frac{d\mathbb{P}^u}{d\mathbb{P}^{u_i}}\frac{d\mathbb{P}^{u_i}}{d\widetilde{\mathbb{P}}}\right) = (1+\lambda_i)\log\frac{d\mathbb{P}^u}{d\mathbb{P}^{u_i}} + \log\frac{d\mathbb{P}^{u_i}}{d\mathbb{Q}} + (1+\lambda_i)\log\widetilde{\mathcal{Z}}$$
(25a)

$$= \lambda_i\log\frac{d\mathbb{P}^u}{d\mathbb{P}^{u_i}} + \log\frac{d\mathbb{P}^u}{d\mathbb{Q}} + (1+\lambda_i)\log\widetilde{\mathcal{Z}}. \quad (25b)$$

Using the definition of the Lagrangian in (5), this implies that

$$(1+\lambda_i)D_{\mathrm{KL}}(\mathbb{P}^u|\widetilde{\mathbb{P}}) = \lambda_i D_{\mathrm{KL}}(\mathbb{P}^u|\mathbb{P}^{u_i}) + D_{\mathrm{KL}}(\mathbb{P}^u|\mathbb{Q}) + (1+\lambda_i)\mathbb{E}\left[\log\widetilde{\mathcal{Z}}(X_0^u)\right] \quad (26a)$$

$$= \mathcal{L}_{\mathrm{TR}}^{(i)}(u,\lambda_i) + (1+\lambda_i)\mathbb{E}\left[\log\widetilde{\mathcal{Z}}(X_0^u)\right] + \lambda_i\varepsilon, \quad (26b)$$

Since we defined the minimizer of the Lagrangian (with optimal multiplier $\lambda_i$) in the last expression as $u_{i+1}$, we have that $u_{i+1} = \arg\min_{u\in\mathcal{U}} D_{\mathrm{KL}}(\mathbb{P}^u|\widetilde{\mathbb{P}})$. This shows that $\widetilde{\mathbb{P}} = \mathbb{P}^{u_{i+1}}$ by the uniqueness of the Radon-Nikodym derivative. For the second statement, we introduce the unnormalized path measure $\widetilde{\mathbb{P}}^{u_{i+1}}$ such that

$$\frac{d\mathbb{P}^{u_{i+1}}}{d\mathbb{P}}(X) = \frac{1}{\widetilde{\mathcal{Z}}_{i+1}(X_0)}\frac{d\widetilde{\mathbb{P}}^{u_{i+1}}}{d\mathbb{P}}(X) \quad\text{with}\quad \widetilde{\mathcal{Z}}_{i+1}(X_0) = \mathbb{E}\left[\frac{d\widetilde{\mathbb{P}}^{u_{i+1}}}{d\mathbb{P}}(X)\Big|X_0\right] \quad (27)$$

and

$$\frac{d\widetilde{\mathbb{P}}^{u_{i+1}}}{d\mathbb{P}}(X) = \left(\frac{d\mathbb{Q}}{d\mathbb{P}}(X)\right)^{\frac{1}{1+\lambda_i}}\left(\frac{d\widetilde{\mathbb{P}}^{u_i}}{d\mathbb{P}}(X)\right)^{\frac{\lambda_i}{1+\lambda_i}}. \quad (28)$$

Assuming $\widetilde{\mathcal{Z}}_0 = 1$, we have $\widetilde{\mathbb{P}}^{u_0} = \mathbb{P}^{u_0}$ for $u_0 \in \mathcal{U}$. By induction, it follows that

$$\frac{d\widetilde{\mathbb{P}}^{u_{i+1}}}{d\mathbb{P}}(X) = \left(\frac{d\mathbb{Q}}{d\mathbb{P}}(X)\right)^{\beta_{i+1}}\left(\frac{d\mathbb{P}^{u_0}}{d\mathbb{P}}(X)\right)^{1-\beta_{i+1}}, \quad (29)$$

with $\beta_{i+1}$ defined as in Prop. 2.2, which proves the second statement. $\square$

**Remark B.1.** Using the left side of (27), we can rewrite the normalized version of (28) as

$$\frac{d\mathbb{P}^{u_{i+1}}}{d\mathbb{P}}(X) = \frac{1}{\widetilde{\mathcal{Z}}_{i+1}(X_0)}\left(\frac{d\mathbb{Q}}{d\mathbb{P}}(X)\right)^{\frac{1}{1+\lambda_i}}\left(\frac{d\widetilde{\mathbb{P}}^{u_i}}{d\mathbb{P}}(X)\right)^{\frac{\lambda_i}{1+\lambda_i}} \quad (30)$$

$$= \frac{1}{\widehat{\mathcal{Z}}_{i+1}(X_0)}\left(\frac{d\mathbb{Q}}{d\mathbb{P}}(X)\right)^{\frac{1}{1+\lambda_i}}\left(\frac{d\mathbb{P}^{u_i}}{d\mathbb{P}}(X)\right)^{\frac{\lambda_i}{1+\lambda_i}}. \quad (31)$$

with $\widehat{\mathcal{Z}}_{i+1} = \widetilde{\mathcal{Z}}_{i+1}/\widetilde{\mathcal{Z}}_i^{\frac{\lambda_i}{1+\lambda_i}}$.

# C  Further related works, broader impact, and limitations

## C.1  Further related works

**Monte Carlo estimator.** In theory, one could directly compute the optimal control using the representations in Prop. 2.5 (for $\lambda = 0$ and $i = 0$; see Item 1 in Thm. D.1) combined with Monte Carlo estimates[8] of the value function in Item 4 in Thm. D.1 [39, 73, 74, 121, 130]. However, in practice this can be problematic since it requires a large amount of samples *for each state $x$* due to the (typically) very high variance of the estimator for $V$ [130]. In particular, we note that the variance translates to a bias in the control due to the logarithmic transformation. Moreover, for nonzero $f$ or

---

[8]One can obtain derivative estimates using adjoint states (as defined in Sec. 2.2) or using reparametrization tricks if the uncontrolled process has suitable, known marginals. For Gaussian marginals, one can also use Stein's lemma [73]. We further note that control variates for such estimators have been analyzed in [96, 105].

general $b$ (e.g., in the fine-tuning setting), one needs to *simulate* the uncontrolled process to obtain samples.

**PDE solver.** One can also leverage the representation of the value function as the solution of an HJB equation (see Item 3 in Thm. D.1). While solving PDEs in high dimensions is very challenging, there exist scalable approaches based on tensor trains and neural networks[9] that leverage backward stochastic differential equations or the Hopf-Cole transform in combination with the Feynman-Kac formula [6, 11, 13, 58, 95, 96, 99, 100]. However, in practice, we only need the value function in the domain where the optimal path measure has sufficiently large values, which is typically not considered for PDE solvers.

**Iterative diffusion optimization.** To focus more on promising regions of the path space, methods for iterative diffusion optimization simulate (sub-)trajectories of the controlled SDE to compute a suitable loss and update the control. Typically, the control is parametrized as a neural network and optimized using variants of stochastic gradient descent. While such methods have been explored for general SOC problems with quadratic control costs [40, 42, 87, 95], many recent works have focused on the special case of sampling from unnormalized densities as described in Sec. 3.1; see, e.g., [8, 15, 86, 112, 130, 131, 144, 149]. From the perspective of path measures, all these works propose to minimize suitable divergences between measures induced by controlled SDEs. While we demonstrate the benefits of leveraging trust region methods for the *Denoising Diffusion Sampler* (DDS) [129], our method could also be extended to other samplers.

**Transition path sampling.** Transition path sampling has been a longstanding problem in physics and chemistry to understand phase transitions and chemical reactions, with applications in energy, catalysis, and drug discovery [20, 128]. Computationally, MCMC-based approaches have been extended to path space to mix the transition path distribution, pioneered by [37]. As discussed in Sec. 3.2, transition path sampling can be formulated as a stochastic optimal control problem and has been numerically solved using reverse KL divergence [141], cross-entropy divergence [71], and log-variance divergence [113]; the optimal control is known to be the Doob's $h$-function [26, 45, 116] (for a review, we refer to [115]). To solve the Doob's $h$-function, [116] proposes a shooting-based method which requires MD simulation to reach the target state, while [45] proposes a Gaussian approximation conditioned on both the initial and target state which satisfies boundary conditions by design and provides a simulation-free optimization algorithm. Similarly to SOC, transition path sampling can also naturally be formulated as a reinforcement learning problem, as shown in [35, 103].

**Diffusion and flow matching reward fine-tuning.** Several of the early works on diffusion fine-tuning focused on directly optimizing the reward model making use of its differentiability [28, 140], without any KL regularization, which can lead to "reward hacking". Some other works [16, 47] framed reward fine-tuning as a reinforcement learning problem, but did not make the probabilistic connection to tilted distributions. [126] provides a probabilistic view of the problem, but proposes an algorithm that is hard to scale. [41] gives a comprehensive view of flow matching reward fine-tuning, introducing memoryless noise schedules as the right ones, as well as a new scalable SOC algorithm that we use and adapt, namely adjoint matching. Using the memoryless noise schedule, a recent work [81] considers GRPO for flow matching fine-tuning. [82, 145] consider alternative algorithms that learn the value functions.

**Diffusion-based sampling from unnormalized densities.** Early work on sampling from unnormalized densities based on a Schrödinger-Föllmer diffusions dates back to [33] and was later implemented using Monte Carlo [39, 73] and deep learning approaches [95, 130, 149]. Another line of work is based on Langevin diffusions [43, 52, 53, 124, 146] and denoising diffusion models based on Ornstein-Uhlenbeck processes [15, 74, 129]. A unifying perspective was proposed in [97, 131], which consider general diffusion bridges. An extension based on underdamped diffusion processes was later proposed by [17]. Recent developments have led to improved loss functions and training schemes [6, 8, 14, 27, 46, 56, 63, 64, 80, 89, 108, 114, 117, 133, 143, 144], exploration capabilities [19, 77, 78], or normalizing constant estimation [57, 65]. Other studies focus on the combination of MCMC and diffusion-based sampling methods [5, 25, 101, 112, 136, 147, 148]. Approaches for discrete state spaces have been proposed in [70, 107, 153]. Combinations of diffusion-based sampling with additional access to ground truth data have been studied in [86, 120]. Lastly, [18, 55] study improved evaluation techniques.

---

[9]Note that some of these approaches correspond to regressions of the Monte Carlo estimators mentioned above [130] or to the IDO methods mentioned below [117].

## C.2 Limitations

While our method for solving stochastic optimal control problems exhibits strong sample efficiency, it relies on storing entire trajectories in the replay buffer during training. In large-scale settings – such as fine-tuning text-to-image models – this necessitates keeping the replay buffer in CPU memory while training occurs on the GPU. This separation introduces additional computational overhead due to data transfers between CPU and GPU; however, the buffer still significantly accelerates the fine-tuning since the main computational cost in such settings stems from the simulation of trajectories.

## C.3 Broader impact

This paper proposes new methodologies and theories that find numerical solutions for stochastic optimal control problems ranging from equilibrium sampling, transition path sampling, to fine-tuning text-to-image generative models. Equilibrium sampling and transition path sampling are important in Bayesian statistics, physics and chemistry where they can be used to estimate free energy, understand phase transition and rare events, thus holding promises to accelerate drug and material discovery. More efficient fine-tuning of text-to-image models democratizes the generation of specialized high-quality visual content for creative applications. However, these capabilities also introduce risks such as the potential for generating convincing misinformation or deepfakes.

# D  Background on SOC

## D.1  Stochastic optimal control

In this work, we consider stochastic optimal control (SOC) problems of the form

$$\min_{u \in \mathcal{U}} \mathcal{L}_{\mathrm{SOC}}(u) = \min_{u \in \mathcal{U}} \mathbb{E}\left[\int_0^T \left(\tfrac{1}{2}\|u(X_s^u, s)\|^2 + f(X_s^u, s)\right) \mathrm{d}s + g(X_T^u)\right], \tag{32}$$

with state costs $f$, terminal costs $g$ and control function $u \in \mathcal{U}$, where $\mathcal{U}$ denotes a set of admissible controls; see App. A.2 for further details. Here, $X^u$ is a controlled SDE of the form

$$\mathrm{d}X_s^u = (b + \sigma u)(X_s^u, s)\mathrm{d}s + \sigma(s)\mathrm{d}W_s, \quad X_0 \sim p_0, \tag{33}$$

with base drift $b$, base distribution $p_0$ (typically a Gaussian or dirac delta distribution), and diffusion coefficient $\sigma$. We denote the path measure induced by (33) by $\mathbb{P}^u \in \mathcal{P}$. Moreover, we simply write $\mathbb{P}$ for the path measure corresponding to the uncontrolled process, i.e.,

$$\mathrm{d}X_s = b(X_s, s)\mathrm{d}s + \sigma(s)\mathrm{d}W_s, \quad X_0 \sim p_0. \tag{34}$$

Given a time $t$ and state $x$, the cost functional $J(u; x, t)$ is the expected cost-to-go for a control $u$ on the time interval $[t, T]$ and is defined as

$$J(u; x, t) = \mathbb{E}\left[\int_t^T \left(\tfrac{1}{2}\|u(X_s^u, s)\|^2 + f(X_s^u, s)\right) \mathrm{d}s + g(X_T^u) \,\Big|\, X_t^u = x\right]. \tag{35}$$

The value function $V$, or, *optimal cost-to-go* is obtained by taking the infimum over all controls in $\mathcal{U}$, that is,

$$V(x, t) = \inf_{u \in \mathcal{U}} J(u; x, t). \tag{36}$$

Then we have the following well-known results on representations of the value function $V$ and solution to the SOC problem $u^*$; see, e.g., [33, 50, 87, 91, 94] for details.

**Theorem D.1** (Optimality for SOC Problems). *Let us define the work functional as*

$$\mathcal{W}(X, t) = \int_t^T f(X_s, s)\,\mathrm{d}s + g(X_T). \tag{37}$$

*Then we have the following representations of the value function $V$ in* (36) *and the solution $u^*$ to the SOC problem in* (32):

1. *(Connection between solution and value function) The solution can be written as $u^* = -\sigma^\top \nabla V$.*

2. *(Optimal change of measure) The Radon-Nikodym derivative of the optimal path measure $\mathbb{Q}$ w.r.t. the uncontrolled path measure $\mathbb{P}$ satisfies*

$$\frac{\mathrm{d}\mathbb{Q}}{\mathrm{d}\mathbb{P}}(X) = \frac{e^{-\mathcal{W}(X, 0)}}{\mathcal{Z}(X_0)} \quad \text{with} \quad \mathcal{Z}(X_0) = \mathbb{E}\big[e^{-\mathcal{W}(X, 0)}|X_0\big]. \tag{38}$$

3. *(PDE for value function) The value function $V$ is the solution to the Hamilton-Jacobi-Bellman (HJB) equation*

$$(\partial_t + L)V(x, t) - \tfrac{1}{2}\|(\sigma^\top \nabla V)(x, t)\|^2 + f(x, t) = 0, \quad V(x, T) = g(x), \tag{39}$$

*where $L := \frac{1}{2}\sum_{i,j=1}^{d}(\sigma\sigma^{\top})_{ij}\partial_{x_i}\partial_{x_j} + \sum_{i=1}^{d}b_i\partial_{x_i}$ denotes the infinitesimal generator of the uncontrolled SDE in (34).*

4. *(Estimator for value function) For every $(x,t) \in \mathbb{R}^d \times [0,T]$ the value function can be written as $V(x,t) = -\log\mathbb{E}\big[e^{-\mathcal{W}(X,t)}\big|X_t = x\big]$, where $X$ is the solution of the uncontrolled SDE in (34).*

Combining the expressions for $u^*$ and $V$ in Thm. D.1, we directly obtain the path integral representation of the optimal control, i.e.,

$$u^*(x,t) = \sigma(t)^{\top}\nabla_x\log\mathbb{E}\left[e^{-\mathcal{W}(X,t)}\Big|X_t = x\right], \tag{40}$$

In practice, computing the optimal control (40) is typically impractical, as it requires running multiple simulations for each state $x$ to obtain a Monte Carlo approximation of the expectation; see App. C.1. To address this challenge, many approaches instead learn a parameterized control function, optimized using stochastic gradient methods. These techniques are collectively referred to as iterative diffusion optimization (IDO) methods and are further discussed in the next section.

### D.2 Iterative diffusion optimization

An alternative view on problem (32) is obtained by considering loss functions on path measures [87]. By the Girsanov theorem (see App. A.3) we have

$$\frac{d\mathbb{P}}{d\mathbb{P}^u}(X^u) = \exp\left(-\int_0^T u(X_s^u,s)\cdot dW_s - \frac{1}{2}\int_0^T\|u(X_s^u,s)\|^2 ds\right). \tag{41}$$

Combining this with the optimal change of measure $d\mathbb{Q}/d\mathbb{P}$ from Thm. D.1, we obtain an expression for $d\mathbb{Q}/d\mathbb{P}^u$, from which we can compute the relative entropy $\mathcal{L}_{\mathrm{RE}}$, i.e., the reverse Kullback-Leibler (KL) divergence

$$\mathcal{L}_{\mathrm{RE}}(u) = D_{\mathrm{KL}}(\mathbb{P}^u|\mathbb{Q}) = \mathbb{E}\left[\int_0^T\left(\tfrac{1}{2}\|u(X_s^u,s)\|^2 + f(X_s^u,s)\right)ds + g(X_T^u) + \log\mathcal{Z}(X_0^u)\right]. \tag{42}$$

Note that minimizing the stochastic optimal control problem in (32) is equal to minimizing the KL divergence, that is,

$$u^* = \arg\min_{u\in\mathcal{U}}\mathcal{L}_{\mathrm{SOC}}(u) = \arg\min_{u\in\mathcal{U}}\mathcal{L}_{\mathrm{RE}}(u), \tag{43}$$

in the sense that both have the same unique optimal control $u^*$ as a minimizer. As such, we can consider an arbitrary divergence $D : \mathcal{P} \times \mathcal{P} \to \mathbb{R}^+$, for which $D(\mathbb{P}_1|\mathbb{P}_2) = 0$ holds if and only if $\mathbb{P}_1 = \mathbb{P}_2$, to solve stochastic optimal control problems. More generally, we can consider any loss function for which the unique minimizer is the optimal control $u^*$. Iterative diffusion optimization builds on this perspective and can be seen as a common framework for solving (potentially high-dimensional) SOC problems by leveraging parameterized control functions and stochastic gradient methods to minimize different loss functions.

### D.3 On the initial value dependence of the normalizing constant

In general, the normalizing constant $\mathcal{Z}(X_0)$ in the optimal change of measure (3) depends on the initial value $X_0$. Let us demonstrate in the following why this is the case. To this end, let us first assume a generic normalization constant $\mathcal{Z}$ that may or may not depend on $X_0$. As in (3), it then holds

$$\frac{d\mathbb{Q}}{d\mathbb{P}}(X) = \frac{e^{-\mathcal{W}(X,0)}}{\mathcal{Z}}. \tag{44}$$

We can then compute

$$\frac{\mathbb{Q}_0(X_0)}{\mathbb{P}_0(X_0)} = \mathbb{E}\left[\frac{d\mathbb{Q}}{d\mathbb{P}}(X)\Big|X_0\right] = \mathbb{E}\left[\frac{e^{-\mathcal{W}(X,0)}}{\mathcal{Z}}\Big|X_0\right]. \tag{45}$$

Now, for a chosen $p_0 = \mathbb{P}_0$ we want that $\mathbb{Q}_0(X_0) = \mathbb{P}_0(X_0)$, which requires

$$\mathcal{Z} = \mathbb{E}\left[e^{-\mathcal{W}(X,0)}\Big|X_0\right] = e^{-V(X_0,0)}. \tag{46}$$

Clearly, the right-hand side depends on $X_0$. Hence, in general, $\mathbb{Q}_0(X_0) = \mathbb{P}_0(X_0)$ can only hold if $\mathcal{Z}$ depends on $X_0$. Conversely, if we wanted to have a global normalizing constant $\mathcal{Z}$, which is independent of $X_0$, we would need to tilt the initial marginal of $\mathbb{Q}$ as well, namely via

$$\mathbb{Q}_0(X_0) = \mathbb{P}_0(X_0)\frac{\mathbb{E}[e^{-\mathcal{W}(X,0)}|X_0]}{\mathcal{Z}} = \mathbb{P}_0(X_0)\frac{e^{-V(X_0,0)}}{\mathcal{Z}}. \tag{47}$$

However, the function $V(\cdot,0)$ is typically not known in practice.

# E   Details on trust region SOC algorithms

## E.1   Characterizing the solutions of the trust region optimization problem

**Proposition E.1** (Characterizing the solutions of the trust region optimization problem). *The solution* $\mathbb{P}^{u_{i+1}}$ *of the problem* (9) *is unique and it satisfies the following:*

- *If* $D_{\mathrm{KL}}(\mathbb{Q}|\mathbb{P}^{u_i}) \leq \varepsilon$, *then* $\mathbb{P}^{u_{i+1}} = \mathbb{Q}$.
- *If* $D_{\mathrm{KL}}(\mathbb{Q}|\mathbb{P}^{u_i}) \geq \varepsilon$, *then* $D_{\mathrm{KL}}(\mathbb{P}^{u_{i+1}}|\mathbb{P}^{u_i}) = \varepsilon$, *i.e.* $\mathbb{P}^{u_{i+1}}$ *is also the unique solution of the problem*

$$\arg\min_{u \in \mathcal{U}} D_{\mathrm{KL}}(\mathbb{P}^u|\mathbb{Q}) \quad s.t. \quad D_{\mathrm{KL}}(\mathbb{P}^u|\mathbb{P}^{u_i}) = \varepsilon. \tag{48}$$

*Proof.* To prove the first case, observe that $\mathbb{Q}$ is the only solution of the unconstrained problem $\arg\min_{\mathbb{P} \in \mathcal{P}} D_{\mathrm{KL}}(\mathbb{P}|\mathbb{Q})$, which means that it is also the unique solution of the problem (9) since it satisfies the constraint $D_{\mathrm{KL}}(\mathbb{Q}|\mathbb{P}^{u_i}) \leq \epsilon$. To prove the second case, by the Karush-Kuhn-Tucker (KKT) conditions, we have that either $\lambda = 0$, or $D_{\mathrm{KL}}(\mathbb{P}^{u_{i+1}}|\mathbb{P}^{u_i}) = \varepsilon$. We assume that $\lambda = 0$ and $D_{\mathrm{KL}}(\mathbb{P}^{u_{i+1}}|\mathbb{P}^{u_i}) < \varepsilon$ to reach a contradiction, which will imply that $D_{\mathrm{KL}}(\mathbb{P}^{u_{i+1}}|\mathbb{P}^{u_i}) = \varepsilon$. The first-order optimality condition for the problem is as follows: for any perturbation $v$ of the control $u_{i+1}$, we have that

$$0 = \frac{\mathrm{d}}{\mathrm{d}\eta}\left(D_{\mathrm{KL}}\left(\mathbb{P}^{u_{i+1}+\eta v}|\mathbb{Q}\right) + \lambda\left(D_{\mathrm{KL}}(\mathbb{P}^{u_{i+1}+\eta v}|\mathbb{P}^{u_i}) - \varepsilon\right)\right)|_{\eta=0} = \frac{\mathrm{d}}{\mathrm{d}\eta}D_{\mathrm{KL}}\left(\mathbb{P}^{u_{i+1}+\eta v}|\mathbb{Q}\right)|_{\eta=0}, \tag{49}$$

which means that $u_{i+1}$ satisfies the first-order optimality condition for the relative entropy loss $u \mapsto D_{\mathrm{KL}}(\mathbb{P}^u|\mathbb{Q})$. By [41, Prop. 2], the only control that satisfies the first-order optimality condition for the relative entropy loss is the optimal control $u^*$, which implies that $\mathbb{P}^{u_{i+1}} = \mathbb{Q}$, which yields a contradiction because $\varepsilon > D_{\mathrm{KL}}(\mathbb{P}^{u_{i+1}}|\mathbb{P}^{u_i}) = D_{\mathrm{KL}}(\mathbb{Q}|\mathbb{P}^{u_i}) \geq \varepsilon$.

Hence, we conclude that $D_{\mathrm{KL}}(\mathbb{P}^{u_{i+1}}|\mathbb{P}^{u_i}) = \varepsilon$. To show that the solution $\mathbb{P}^{u_{i+1}}$ is unique, we use that $\mathbb{P} \mapsto D_{\mathrm{KL}}(\mathbb{P}|\mathbb{P}^{u_i})$ is strictly convex, and that $\{\mathbb{P}|D_{\mathrm{KL}}(\mathbb{P}|\mathbb{P}^{u_i}) \leq \varepsilon\}$ is a convex set because it is the sublevel set of a convex mapping. $\square$

## E.2   Implementation

We provide a detailed version of Algorithm 1 in Algorithm 2. The hyperparameters and used repositories for the experiments on unnormalized densities, transition path sampling, and fine-tuning can be found in the respective sections in Apps. I to K.

## E.3   Variance of the importance weights and trust region bounds

As mentioned in Remark 2.1, one motivation of the trust region constrain $D_{\mathrm{KL}}(\mathbb{P}^u|\mathbb{P}^{u_i}) \leq \varepsilon$ defined in (4) is to keep the variance of the importance weights between two consecutive measures $\mathbb{P}^{u_i}$ and $\mathbb{P}^{u_{i+1}}$ small. This can be motivated by the inequality

$$\mathrm{Var}_{\mathbb{P}^{u_i}}\left(\frac{\mathrm{d}\mathbb{P}^{u_{i+1}}}{\mathrm{d}\mathbb{P}^{u_i}}\right) = \mathbb{E}_{\mathbb{P}^{u_i}}\left[\left(\frac{\mathrm{d}\mathbb{P}^{u_{i+1}}}{\mathrm{d}\mathbb{P}^{u_i}}\right)^2 - 1\right] = \mathbb{E}_{\mathbb{P}^{u_{i+1}}}\left[\frac{\mathrm{d}\mathbb{P}^{u_{i+1}}}{\mathrm{d}\mathbb{P}^{u_i}} - 1\right] \tag{50a}$$

$$\geq \exp\left(\mathbb{E}_{\mathbb{P}^{u_{i+1}}}\left[\log\frac{\mathrm{d}\mathbb{P}^{u_{i+1}}}{\mathrm{d}\mathbb{P}^{u_i}}\right]\right) - 1 = \exp\left(D_{\mathrm{KL}}(\mathbb{P}^{u_{i+1}}|\mathbb{P}^{u_i})\right) - 1, \tag{50b}$$

which follows by Jensen's inequality. While a lower bound on the variance is not straight forward for path space measures (cf. [60]), we can consider the following heuristics. Let us assume that

$$\frac{\mathrm{d}\mathbb{P}^{u_{i+1}}}{\mathrm{d}\mathbb{P}^{u_i}} \approx 1 \tag{51}$$

$\mathbb{P}^{u_i}$- and $\mathbb{P}^{u_{i+1}}$-almost surely, which is reasonable if $D_{\mathrm{KL}}(\mathbb{P}^{u_{i+1}}|\mathbb{P}^{u_i}) \leq \varepsilon$ with $\varepsilon \ll 1$. By a Taylor approximation it then holds

$$\left(\frac{\mathrm{d}\mathbb{P}^{u_{i+1}}}{\mathrm{d}\mathbb{P}^{u_i}}\right)^2 = \exp\left(2\log\frac{\mathrm{d}\mathbb{P}^{u_{i+1}}}{\mathrm{d}\mathbb{P}^{u_i}}\right) \approx 1 + 2\log\frac{\mathrm{d}\mathbb{P}^{u_{i+1}}}{\mathrm{d}\mathbb{P}^{u_i}}. \tag{52}$$

Now, taking expectations w.r.t. $\mathbb{P}^{u_i} \approx \mathbb{P}^{u_{i+1}}$, respectively, using computations similar to (50), and assuming $D_{\mathrm{KL}}(\mathbb{P}^{u_{i+1}}|\mathbb{P}^{u_i}) = \varepsilon$, as argued in App. E.1, yields

$$\mathrm{Var}_{\mathbb{P}^{u_i}}\left(\frac{\mathrm{d}\mathbb{P}^{u_{i+1}}}{\mathrm{d}\mathbb{P}^{u_i}}\right) \approx 2\varepsilon. \tag{53}$$

---

**Algorithm 2** Trust Region SOC with buffer

---

**Require:** Neural network $u_\theta$ with parameters $\theta$, target path measure $\mathbb{Q}$, buffer size $K$, time discretization $S = (s_j)_{j=0}^J \subset [0, T]$, number of gradient steps $M$ per trust region iteration, termination threshold $\delta$

  Initialize $i = 0$ and $\lambda_0 = \infty$

  **for** $i = 0, 1, \ldots$ **do**

    Define $u_i = u_\theta$ (detached)

    Simulate $K$ trajectories $(X_s^{(k)})_{s \in S}$ of the SDE in (1) with Brownian motion $W_s^{(k)}$ and control $u_i$

    Compute importance weights $w^{(k)} = \frac{d\mathbb{Q}}{d\mathbb{P}^{u_i}}(X^{(k)}) \propto \exp(-\mathcal{W}_i(X^{(k)}, 0))$ as in (12)

    Initialize buffer $\mathcal{B} = \big\{ (W^{(k)}, X^{(k)}, w^{(k)}) \big\}_{k=1}^K$

    Compute multiplier $\lambda_i = \arg\max_{\lambda \in \mathbb{R}^+} \mathcal{L}_{\mathrm{Dual}}^{(i)}(\lambda)$ as in (14) using $\mathcal{B}$ and a 1-dim. non-linear solver

    **if** $\lambda_i \leq \delta$ **then**

        **return** control $u_i$ with $\mathbb{P}^{u_i} \approx \mathbb{Q}$

    **if** adjoint matching loss **then**

        Compute annealing $\beta_{i+1} = 1 - \prod_{j=0}^i \frac{\lambda_i}{1 + \lambda_i}$ as in Prop. 2.2

        Compute lean adjoint states $a_s^{(k)} = a_{i+1}(X_s^{(k)}, s)$, $s \in S$, as in (18) and store in $\mathcal{B}$

    **for** $m = 1, \ldots, M$ **do**

        **if** adjoint matching loss **then**

            Estimate $\mathcal{L}(\theta) = \mathbb{E}_{(X,w,a) \sim \mathcal{B},\ s \sim \mathrm{Unif}(S)} \big[ \| \sigma^\top a_s - u_\theta(X_s, s) \|^2 w^{\frac{1}{1+\lambda_i}} \big]$ as in (19)

        **if** log-variance loss **then**

            Estimate $\mathcal{L}(\theta) = \mathrm{Var}_{(W,X,w) \sim \mathcal{B}} \big[ \sum_{j=1}^J \big( \frac{\|\Delta_j\|^2 (s_j - s_{j-1})}{2} + \Delta_j \cdot (W_{s_j} - W_{s_{j-1}}) \big) + \frac{1}{1+\lambda_i} \log w \big]$
            with $\Delta_j = u_i(X_{s_j}, s_j) - u_\theta(X_{s_j}, s_j)$ as in (16)

        Perform a gradient-descent step on $\mathcal{L}(\theta)$

---

### E.4 Lagragian formulation

Using the Girsanov theorem (see App. A.3), we first note that we can write the Lagrangian as

$$
\mathcal{L}_{\mathrm{TR}}^{(i)}(u, \lambda) = \mathbb{E}\left[ \int_0^T \tfrac{1}{2} \| u(X_s^u, s) \|^2 ds + \mathcal{W}(X^u, 0) + \log \mathcal{Z}(X_0^u) \right] + \lambda \left( D_{\mathrm{KL}}(\mathbb{P}^u | \mathbb{P}^{u_i}) - \varepsilon \right) \tag{54}
$$

$$
= \mathbb{E}\left[ \int_0^T \left( \tfrac{1}{2} \| u(X_s^u, s) \|^2 + \tfrac{\lambda}{2} \| u(X_s^u, s) - u_i(X_s^u, s) \|^2 \right) ds + \mathcal{W}(X^u, 0) + \log \mathcal{Z}(X_0^u) \right] - \lambda \varepsilon \tag{55}
$$

$$
= \mathbb{E}\left[ \int_0^T \left( \tfrac{1+\lambda}{2} \| u(X_s^u, s) - \tfrac{\lambda}{1+\lambda} u_i(X_s^u, s) \|^2 + f_i(X_s^u, s) \right) ds + g(X_T^u) + \log \mathcal{Z}(X_0^u) \right] - \lambda \varepsilon \tag{56}
$$

$$
= \mathcal{L}_{\mathrm{TRC}}^{(i)}(u, \lambda) - \lambda \varepsilon, \tag{57}
$$

where $\mathcal{L}_{\mathrm{TRC}}^{(i)}(u, \lambda)$ is defined as in (11), $\lambda \in \mathbb{R}^+$ is the Lagrangian multiplier for the trust region constraint, and we abbreviate $f_i := \frac{\lambda}{2(1+\lambda)} \| u_i \|^2 + f$. For fixed $\lambda$, optimizing the Lagrangian $\mathcal{L}_{\mathrm{TR}}^{(i)}(u, \lambda)$ with respect to $u$ is again an SOC problem. As such, for given $u_i$ and $\lambda$, we can define the value function as

$$
V_{i+1}^\lambda(x, t) = \inf_{u \in \mathcal{U}} \mathbb{E}\left[ \int_t^T \left( \tfrac{1+\lambda}{2} \| u(X_s^u, s) - \tfrac{\lambda}{1+\lambda} u_i(X_s^u, s) \|^2 + f_i(X_s^u, s) \right) ds + g(X_T^u) | X_t = x \right]. \tag{58}
$$

The next proposition provides representations for the value function and the solution to the SOC problem.

**Proposition E.2** (Optimality for trust region SOC problems)**.** *For fixed $\lambda$, let us define by*

$$
V_{i+1}^\lambda(x, t) := \inf_{u \in \mathcal{U}} \mathbb{E}\left[ \int_0^T \left( \tfrac{1+\lambda}{2} \| u - \tfrac{\lambda}{1+\lambda} u_i \|^2 + \tfrac{\lambda}{2(1+\lambda)} \| u_i \|^2 + f \right)(X_s^u, s) ds + g(X_T^u) \,\middle|\, X_t = x \right]
$$

*the value function of the SOC problem $\inf_{u \in \mathcal{U}} \mathcal{L}_{\mathrm{TRC}}^{(i)}(u, \lambda)$ corresponding to (11) and by $u_{i+1}^\lambda$ its solution. Then it holds that*

(i) (Estimator for value function) $V_{i+1}^\lambda(x,t) = -(1+\lambda)\log\mathbb{E}\left[e^{-\frac{1}{1+\lambda}\mathcal{W}_i(X^{u_i},t)}\Big|X_t^{u_i} = x\right],$

where $\quad \mathcal{W}_i(X^{u_i},t) = \int_t^T \frac{1}{2}\|u_i(X_s^{u_i},s)\|^2\mathrm{d}s + \int_t^T u_i(X_s^{u_i},s)\cdot\mathrm{d}W_s + \mathcal{W}(X^{u_i},t).$

(ii) (Connection between solution and value function) It holds $u_{i+1}^\lambda = \frac{\lambda}{1+\lambda}u_i - \frac{1}{1+\lambda}\sigma^\top\nabla V_{i+1}^\lambda.$

Moreover, for $u_0 = \mathbf{0}$ and the optimal Lagrange multiplier $\lambda_i$, let us define the value function

$$\widetilde{V}_{i+1}(x,t) := \inf_{u\in\mathcal{U}}\mathbb{E}\left[\int_0^T\left(\frac{1}{2}\|u\|^2 + \beta_{i+1}f\right)(X_s^u,s)\,\mathrm{d}s + \beta_{i+1}g(X_T^u)\Big|X_t = x\right]$$

of the SOC problem given by the optimal change of measure

$$\frac{\mathrm{d}\mathbb{P}^{u_{i+1}}}{\mathrm{d}\mathbb{P}}(X) = \frac{1}{\widetilde{\mathcal{Z}}_{i+1}(X_0)}\left(\frac{\mathrm{d}\mathbb{Q}}{\mathrm{d}\mathbb{P}}(X)\right)^{\beta_{i+1}} = \frac{e^{-\beta_{i+1}\mathcal{W}(X,0)}}{\widetilde{\mathcal{Z}}_{i+1}(X_0)} \tag{59}$$

as in Prop. 2.2 and (3), and $\widetilde{\mathcal{Z}}_{i+1}(X_0)$ as defined in (27). Then it holds that

(iii) (Estimator for value function) $\widetilde{V}_{i+1}(x,t) = -\log\mathbb{E}\left[e^{-\beta_{i+1}\mathcal{W}(X_t,t)}|X_t = x\right],$

(iv) (Connection between solution and value function) $u_{i+1} = u_{i+1}^{\lambda_i} = -\sigma^\top\nabla\widetilde{V}_{i+1}.$

*Proof.* For notational convenience, we abbreviate $V = V_{i+1}^\lambda$ in this proof. From the verification theorem (see, e.g., [94, Theorem 3.5.2]), we obtain that the value function is the solution to the HJB equation

$$(\partial_t + L)V = -\inf_{\alpha\in\mathbb{R}^d}\left\{f_i + \frac{1+\lambda}{2}\|\alpha - \frac{\lambda}{1+\lambda}u_i\|^2 + \sigma\alpha\cdot\nabla V\right\} \tag{60a}$$

$$= -f_i - \inf_{\alpha\in\mathbb{R}^d}\left\{\frac{1+\lambda}{2}\|\alpha - \frac{\lambda}{1+\lambda}u_i\|^2 + \sigma\alpha\cdot\nabla V\right\}, \quad V(\cdot,T) = g, \tag{60b}$$

where the infimum is pointwise for every $(x,t)\in\mathbb{R}^d\times[0,T]$ and the optimal $\alpha^*$ defines the solution $u^*$. Solving for $\alpha$ yields $\alpha^* = \frac{\lambda}{1+\lambda}u_i - \frac{1}{1+\lambda}\sigma^\top\nabla V$, which proves Item (ii).

Plugging this result back into the HJB equation, we obtain

$$(\partial_t + L)V = -f - \frac{\lambda}{2(1+\lambda)}\|u_i\|^2 - \frac{1}{2(1+\lambda)}\|\sigma^\top\nabla V\|^2 - \sigma\left(\frac{\lambda}{1+\lambda}u_i - \frac{1}{1+\lambda}\sigma^\top\nabla V\right)\cdot\nabla V \tag{61a}$$

$$= -f - \frac{\lambda}{2(1+\lambda)}\|u_i\|^2 + \frac{1}{2(1+\lambda)}\|\sigma^\top\nabla V\|^2 - \frac{\lambda}{1+\lambda}\sigma u_i\cdot\nabla V \tag{61b}$$

$$= -f - \frac{\lambda}{2(1+\lambda)}\|u_i\|^2 + \frac{1}{2(1+\lambda)}\|\sigma^\top\nabla V\|^2 - \sigma u_i\cdot\nabla V + \frac{1}{1+\lambda}\sigma u_i\cdot\nabla V. \tag{61c}$$

Now, we define the infinitesimal generator of the SDE

$$\mathrm{d}X_s^{u_i} = (b(X_s^{u_i},s) + \sigma u_i(X_s^{u_i},s))\,\mathrm{d}s + \sigma\mathrm{d}W_s \tag{62}$$

as

$$\bar{L} := \frac{1}{2}\sum_{i,j=1}^d(\sigma\sigma^\top)_{ij}\partial_{x_i}\partial_{x_j} + \sum_{i=1}^d(b_i + (\sigma u_i)_i)\partial_{x_i} = L + \sum_{i=1}^d(\sigma u_i)_i\partial_{x_i}. \tag{63}$$

Using (63), we can rewrite (61) as

$$(\partial_t + \bar{L})V = -f - \frac{\lambda}{2(1+\lambda)}\|u_i\|^2 + \frac{1}{2(1+\lambda)}\|\sigma^\top\nabla V\|^2 + \sigma\frac{1}{1+\lambda}u_i\cdot\nabla V \tag{64a}$$

$$= -f - \frac{1}{2}\|u_i\|^2 + \frac{1}{2(1+\lambda)}\|u_i + \sigma^\top\nabla V\|^2 \tag{64b}$$

By Itô's formula (see App. A.3), we have

$$\mathrm{d}V(X_s^{u_i},s) = (\partial_s + \bar{L})V(X_s^{u_i},s)\mathrm{d}s + \sigma^\top\nabla V(X_s^{u_i},s)\cdot\mathrm{d}W_s. \tag{65}$$

Plugging (64) into (65) and defining $Y_s := V(X_s^{u_i},s)$ and $Z_s := (-u_i - \sigma^\top\nabla V)(X_s^{u_i},s)$, we obtain the pair of forward-backward SDEs (FBSDEs)

$$\mathrm{d}X_s^{u_i} = (b(X_s^{u_i},s) + \sigma u_i(X_s^{u_i},s))\,\mathrm{d}s + \sigma(s)\mathrm{d}W_s, \quad X_0^{u_i}\sim p_0, \tag{66}$$

$$\mathrm{d}Y_s = \left(-f(X_s^{u_i},s) - \frac{1}{2}\|u_i(X_s^{u_i},s)\|^2 + \frac{1}{2(1+\lambda)}\|Z_s\|^2\right)\mathrm{d}s - (u_i(X_s^{u_i},s) + Z_s)\cdot\mathrm{d}W_s, \tag{67}$$

with $Y_T = g(X_T^{u_i})$. This shows that

$$g(X_T^{u_i}) = Y_t - \int_t^T\left(f(X_s^{u_i},s) + \frac{1}{2}\|u_i(X_s^{u_i},s)\|^2 - \frac{1}{2(1+\lambda)}\|Z_s\|^2\right)\mathrm{d}s - \int_t^T(u_i(X_s^{u_i},s) + Z_s)\cdot\mathrm{d}W_s,$$

which can be rewritten as

$$\mathcal{W}_i(X^{u_i}, t) = Y_t + \int_t^T \tfrac{1}{2(1+\lambda)} \|Z_s\|^2 \mathrm{d}s - \int_t^T Z_s \cdot \mathrm{d}W_s. \tag{68}$$

Using the definition of $Y_t$, we can now write

$$\mathbb{E}\left[e^{-\frac{1}{1+\lambda}\mathcal{W}_i(X^{u_i},t)} \Big| X_t^{u_i} = x\right] = e^{-\frac{1}{1+\lambda}V(X_t^{u_i},t)} \mathbb{E}\left[e^{\frac{1}{1+\lambda}\int_t^T Z_s \cdot \mathrm{d}W_s - \frac{1}{(1+\lambda)^2}\int_t^T \frac{1}{2}\|Z_s\|^2 \mathrm{d}s} \Big| X_t^{u_i} = x\right]$$

$$= e^{-\frac{1}{1+\lambda}V(X_t^{u_i},t)},$$

where we leveraged Novikov's theorem to show that the Doléans-Dade exponential is a martingale with vanishing expectation. This concludes the proof of Item (i). The proof of Items (iii) and (iv) follows directly from Thm. D.1. □

# F   Trust region SOC sequences and Fisher-Rao geometry

For a fixed $\varepsilon$, suppose that we construct the sequence of controls $(u_{i+1})_{i\geq 0}$ as the solutions of the problem (4). As shown in Prop. 2.2, we have that

$$\frac{\mathrm{d}\mathbb{P}^{u_i}}{\mathrm{d}\mathbb{P}} \propto \left(\frac{\mathrm{d}\mathbb{Q}}{\mathrm{d}\mathbb{P}}\right)^{\beta_i} \left(\frac{\mathrm{d}\mathbb{P}^{u_0}}{\mathrm{d}\mathbb{P}}\right)^{1-\beta_i}, \quad \text{with} \quad \beta_i = 1 - \prod_{j=0}^{i-1} \tfrac{\lambda_j}{1+\lambda_j} \tag{69}$$

If we define the family $(\mathbb{Q}^{(\tau)})_{\tau\in[0,1]}$ such that

$$\frac{\mathrm{d}\mathbb{Q}^{(\tau)}}{\mathrm{d}\mathbb{P}} \propto \left(\frac{\mathrm{d}\mathbb{Q}}{\mathrm{d}\mathbb{P}}\right)^{\tau} \left(\frac{\mathrm{d}\mathbb{P}^{u_0}}{\mathrm{d}\mathbb{P}}\right)^{1-\tau}, \tag{70}$$

we can write $\mathbb{P}^{u_i} = \mathbb{Q}^{(\beta_i)}$. Hence, we can regard the sequence $(\mathbb{P}^{u_i})_{i\geq 0}$ as a discretization of the family $(\mathbb{Q}^{(\tau)})_{\tau\in[0,1]}$. Next, we characterize this discretization more precisely using tools from information geometry.

## F.1   Basics on information geometry

Let $\{p(x;\theta)\}_{\theta\in\Theta}$ be a parametric family of probability densities (or mass functions) on the sample space $\mathcal{X}$, and let $X$ be a random variable with distribution $p(x;\theta)$.

**Definition F.1** (Fisher information matrix). The *Fisher information matrix* at $\theta$ is defined as

$$\mathcal{I}(\theta) = \mathbb{E}_{X\sim p(\cdot;\theta)}\left[\nabla_\theta \log p(X;\theta)\left(\nabla_\theta \log p(X;\theta)\right)^\top\right] = -\mathbb{E}_{X\sim p(\cdot;\theta)}\left[\nabla_\theta^2 \log p(X;\theta)\right],$$

where $\nabla_\theta$ denotes the column gradient with respect to $\theta$, and $\nabla_\theta^2$ the Hessian.

As an average of positive semi-definite matrices, $\mathcal{I}(\theta)$ is positive semi-definite, which makes it possible to define a geometric structure:

**Definition F.2** (Statistical manifold). Let $\{p(x;\theta)\}_{\theta\in\Theta}$ be a smooth parametric family of probability densities on $\mathcal{X}$, with parameter space $\Theta \subseteq \mathbb{R}^d$. Then $\Theta$ itself can be viewed as a $d$-dimensional differentiable manifold

$$\mathcal{M} = \{p(\cdot;\theta) : \theta \in \Theta\} \cong \Theta,$$

called the *statistical manifold* of the model. Endow $\mathcal{M}$ with the Riemannian metric

$$g_{ij}(\theta) = \mathcal{I}_{ij}(\theta) = \mathbb{E}_{X\sim p(\cdot;\theta)}\left[\partial_i \log p(X;\theta)\, \partial_j \log p(X;\theta)\right],$$

where $\partial_i = \frac{\partial}{\partial\theta_i}$. This $g$ is known as the *Fisher–Rao metric*, turning $(\mathcal{M}, g)$ into the canonical information-geometric manifold of the model.

Next, we review the definition of the length of a curve on a Riemannian manifold.

**Definition F.3** (Length of a curve on a Riemannian manifold). Let $(\mathcal{M}, g)$ be a $d$-dimensional Riemannian manifold, and let $\gamma\colon [a,b] \longrightarrow \mathcal{M}$ be a piecewise smooth curve. Choose local coordinates $\theta = (\theta^1, \ldots, \theta^d)$ on an open set $\mathcal{U} \subset \mathcal{M}$ containing the image of $\gamma$, so that $\gamma(t) \mapsto \theta(t) = (\theta^1(t), \ldots, \theta^d(t))$. Then the *length* of $\gamma$ is

$$L(\gamma) = \int_a^b \sqrt{g_{ij}\big(\theta(t)\big)\, \dot\theta^i(t)\, \dot\theta^j(t)}\; dt,$$

where $\dot\theta^i(t) = \frac{d\theta^i}{dt}(t)$ and we employ the Einstein summation convention on repeated indices $i, j = 1, \ldots, d$.

A geodesic between two points $\theta_1, \theta_2 \in \mathcal{M}$ is a piecewise smooth curve $\gamma \colon [a, b] \longrightarrow \mathcal{M}$ such that $\gamma(a) = \theta_1, \gamma(b) = \theta_2$ that minimizes the length functional $L$ locally. Any time reparameterization of a geodesic is also a geodesic, because the geodesic distance between $\theta_1, \theta_2$ is the infimum over the lengths of all geodesics (or all piecewise smooth curves) between $\theta_1, \theta_2$.

**Definition F.4** (Fisher-Rao distance). The geodesic distance induced by the Fisher-Rao metric is known as the *Fisher–Rao distance*.

Lastly, we present another statement which connects the Kullback–Leibler divergence and the Fisher information matrix using a local expansion of the KL divergence.

**Proposition F.5** (Second-order expansion of KL). *Let $\{p(x; \theta)\}_{\theta \in \Theta}$ be a smooth parametric family of densities, and fix $\theta \in \Theta$. For a small increment $\delta \in \mathbb{R}^d$, consider*

$$\mathrm{KL}\big(p(\cdot\,; \theta + \delta) \,\|\, p(\cdot\,; \theta)\big) \;=\; \int_{\mathcal{X}} p(x; \theta + \delta) \, \log \frac{p(x; \theta + \delta)}{p(x; \theta)} \, dx.$$

*Then one has the Taylor expansion*

$$\mathrm{KL}\big(p(\theta + \delta) \| p(\theta)\big) = \underbrace{0}_{\text{constant term}} \;+\; \underbrace{0}_{\text{linear term}} \;+\; \frac{1}{2} \, \delta^i \, \mathcal{I}_{ij}(\theta) \, \delta^j \;+\; O\big(\|\delta\|^3\big),$$

*where*

$$\mathcal{I}_{ij}(\theta) \;=\; \mathbb{E}_{X \sim p(\cdot\,; \theta)} \big[ \partial_i \log p(X; \theta) \, \partial_j \log p(X; \theta) \big]$$

*is the Fisher information matrix. Equivalently,*

$$\left. \frac{\partial \mathrm{KL}}{\partial \delta^i} \right|_{\delta=0} = 0, \qquad \left. \frac{\partial^2 \mathrm{KL}}{\partial \delta^i \partial \delta^j} \right|_{\delta=0} = \mathcal{I}_{ij}(\theta).$$

*Sketch.* Expand both $p(x; \theta + \delta)$ and $\log p(x; \theta + \delta)$ to second order in $\delta$, substitute into the integral, and use $\int p \, \partial_i \log p \, dx = 0$ and $\int p \, \partial_i \partial_j \log p \, dx = -\mathcal{I}_{ij}(\theta)$ to verify cancellation of constant and linear terms, leaving the stated quadratic form. $\qquad\square$

## F.2 Fisher–Rao geometry of an exponential family

**Definition F.6** (The exponential-family manifold). Let

$$p(x; \theta) \;=\; \exp\big(\theta^i T_i(x) \,-\, A(\theta)\big) \, h(x), \quad \theta = (\theta^1, \ldots, \theta^d) \in \Theta \subseteq \mathbb{R}^d$$

be a regular $d$-parameter exponential family on $\mathcal{X}$. The parameter space $\Theta$ (equipped with the atlas coming from the coordinates $\theta^i$) is a $d$-dimensional differentiable manifold, which we identify with the statistical model

$$\mathcal{M} \;=\; \big\{ p(\,\cdot\,; \theta) \mid \theta \in \Theta \big\}.$$

Its tangent space at $\theta$ is $T_\theta \mathcal{M} \;\cong\; \mathbb{R}^d$, with basis $\{\partial / \partial \theta^i\}$.

**Definition F.7** (Fisher–Rao metric). The Fisher–Rao metric on $\mathcal{M}$ is the Riemannian metric whose components in the natural coordinate chart $\theta$ are

$$g_{ij}(\theta) \;=\; \mathbb{E}_{X \sim p(\cdot\,; \theta)} \big[ \partial_i \log p(X; \theta) \, \partial_j \log p(X; \theta) \big] \;=\; -\mathbb{E}_{X \sim p(\cdot\,; \theta)} \big[ \partial_{ij} \log p(X; \theta) \big] \;=\; \frac{\partial^2 A(\theta)}{\partial \theta^i \, \partial \theta^j}.$$

Equivalently, $g(\theta) = \nabla^2 A(\theta)$, the Hessian of the log-partition function.

For general exponential families, the Fisher-Rao distance and the geodesics do not admit a closed form. Yet, one-dimensional families can be handled explicitly, because geodesics are trivial:

**Proposition F.8** (One-parameter exponential family). *If $d = 1$ then $\theta \in (a, b) \subseteq \mathbb{R}$, and $g(\theta) = A''(\theta)$. Hence*

$$\mathrm{FR}(\theta_1, \theta_2) \;=\; \left| \int_{\theta_1}^{\theta_2} \sqrt{A''(\theta)} \, d\theta \right|.$$

## F.3 Fisher–Rao geometry of an exponential family of path measures

We can view the family $(\mathbb{Q}^{(\tau)})_{\tau \in [0,1]}$ defined in (70) as a one-parameter exponential family [22] by rewriting $\mathbb{Q}^{(\tau)}$ as

$$\frac{d\mathbb{Q}^{(\tau)}}{d\mathbb{P}^{u_0}} = \exp\left( \tau \left( \log \frac{d\mathbb{Q}}{d\mathbb{P}^{u_0}} \right) - A(\tau) \right), \tag{71}$$

where the log-partition function $A(\tau)$ is defined as

$$A(\tau) = \log \mathbb{E}_{\mathbb{P}^{u_0}} \left[ \left( \frac{d\mathbb{Q}}{d\mathbb{P}^{u_0}} \right)^\tau \right]. \tag{72}$$

Equivalently, we can write it as an exponential family centered on an arbitrary $\tau \in [0, 1]$:

$$\frac{d\mathbb{Q}^{(\tau+\Delta\tau)}}{d\mathbb{Q}^{(\tau)}} = \exp\left(\Delta\tau\left(\log\frac{d\mathbb{Q}}{d\mathbb{P}^{u_0}}\right) - A_\tau(\Delta\tau)\right), \tag{73}$$

where

$$A_\tau(\Delta\tau) := \log\mathbb{E}_{\mathbb{Q}^{(\tau)}}\left[\left(\frac{d\mathbb{Q}}{d\mathbb{P}^{u_0}}\right)^{\Delta\tau}\right]. \tag{74}$$

**Deriving an expression for the Fisher information.** Observe that by construction

$$A_\tau(\Delta\tau) := \log\mathbb{E}_{\mathbb{P}^{u_0}}\left[\left(\frac{d\mathbb{Q}}{d\mathbb{P}^{u_0}}\right)^{\Delta\tau}\frac{d\mathbb{Q}^{(\tau)}}{d\mathbb{P}^{u_0}}\right] = \log\mathbb{E}_{\mathbb{P}^{u_0}}\left[\left(\frac{d\mathbb{Q}}{d\mathbb{P}^{u_0}}\right)^{\Delta\tau}\left(\frac{d\mathbb{Q}}{d\mathbb{P}^{u_0}}\right)^{\tau}\exp\left(-A(\tau)\right)\right] \tag{75}$$

$$= A(\tau + \Delta\tau) - A(\tau),$$

which means that $A'_\tau(0) = A'(\tau)$ for all $\tau \in (0, 1)$. Thus, by Prop. F.8, we conclude that the Fisher information matrix, which is a scalar because the manifold is one-dimensional, reads

$$\mathcal{I}(\tau) = A''(\tau) = A''_\tau(0). \tag{76}$$

Computing the first and second derivatives of $A_\tau$ is straight-forward:

$$A'_\tau(\Delta\tau) = \frac{\mathbb{E}_{\mathbb{Q}^{(\tau)}}\left[\log\left(\frac{d\mathbb{Q}}{d\mathbb{P}^{u_0}}\right)\left(\frac{d\mathbb{Q}}{d\mathbb{P}^{u_0}}\right)^{\Delta\tau}\right]}{\mathbb{E}_{\mathbb{Q}^{(\tau)}}\left[\left(\frac{d\mathbb{Q}}{d\mathbb{P}^{u_0}}\right)^{\Delta\tau}\right]}, \tag{77}$$

$$A''_\tau(0) = \mathbb{E}_{\mathbb{Q}^{(\tau)}}\left[\log\left(\frac{d\mathbb{Q}}{d\mathbb{P}^{u_0}}\right)^2\right] - \mathbb{E}_{\mathbb{Q}^{(\tau)}}\left[\log\left(\frac{d\mathbb{Q}}{d\mathbb{P}^{u_0}}\right)\right]^2,$$

and this implies that

$$\mathcal{I}(\tau) = \mathrm{Var}_{\mathbb{Q}^{(\tau)}}\left[\log\left(\frac{d\mathbb{Q}}{d\mathbb{P}^{u_0}}\right)\right]. \tag{78}$$

**Connecting the trust region constraint to the Fisher information.** Applying Proposition F.5, we obtain that

$$\mathrm{KL}\left(\mathbb{Q}^{(\tau+\Delta\tau)}|\mathbb{Q}^{(\tau)}\right) = \frac{\Delta\tau^2}{2}\mathcal{I}(\tau) + O(\Delta\tau^3), \tag{79}$$

When we set $\tau + \Delta\tau = \beta_{i+1}$, $\tau = \beta_i$, we have that

$$\varepsilon = \mathrm{KL}\left(\mathbb{P}^{u_{i+1}}|\mathbb{P}^{u_i}\right) = \frac{\Delta\tau^2}{2}\mathcal{I}(\tau) + O(\Delta\tau^3). \tag{80}$$

Thus,

$$\Delta\tau = \sqrt{\frac{2\varepsilon}{\mathcal{I}(\tau)}} + O(\Delta\tau^{3/2}), \tag{81}$$

Moreover, the Fisher-Rao distance between $\mathbb{P}^{u_0}$ and $\mathbb{P}^{(i)}$, or rather, between $0$ and $\beta_i$,

$$\mathrm{FR}(0, \beta_i) = \int_0^{\beta_i}\sqrt{\mathcal{I}(\tau)}d\tau. \tag{82}$$

Then, the difference between Fisher-Rao distances $\mathrm{FR}(0, \beta_{i+1})$ and $\mathrm{FR}(0, \beta_i)$ which is equal to the Fisher-Rao distance $\mathrm{FR}(\beta_i, \beta_{i+1})$ is

$$\mathrm{FR}(0, \beta_{i+1}) - \mathrm{FR}(0, \beta_i) = \mathrm{FR}(\beta_i, \beta_{i+1}) = \int_{\beta_i}^{\beta_{i+1}}\sqrt{\mathcal{I}(\tau)}d\tau$$

$$= \left(\sqrt{\mathcal{I}(\beta_i)} + O(\beta_{i+1} - \beta_i)\right)(\beta_{i+1} - \beta_i) = \sqrt{\mathcal{I}(\beta_i)}\Delta\tau + O(\Delta\tau^2) \tag{83}$$

$$= \sqrt{\mathcal{I}(\beta_i)}\sqrt{\frac{2\varepsilon}{\mathcal{I}(\beta_i)}} + O(\Delta\tau^{3/2}) = \sqrt{2\varepsilon} + O(\Delta\tau^{3/2}).$$

In continuous time, we have a curve $\beta : \mathbb{R}^{>0} \to [0, 1]$, and

$$\frac{d}{dt}\mathrm{FR}(0, \beta(t)) = \sqrt{\mathcal{I}(\beta(t))}\beta'(t) = \sqrt{\mathcal{I}(\beta(t))}\sqrt{\frac{2}{\mathcal{I}(\beta(t))}} = \sqrt{2} \tag{84}$$

Thus, we have shown the following result:

**Proposition F.9.** *Up to high order terms, the elements of sequence $(\mathbb{P}^{u_i})_{0\leq i\leq I-1}$ are equispaced in the Fisher-Rao distance. The last term $\mathbb{P}^{u_I}$ is equal to the target distribution $\mathbb{Q}$.*

**A Monte Carlo estimate for the Fisher information.** By equation (71), we have that $\log \frac{\mathrm{d}\mathbb{Q}^{(\tau)}}{\mathrm{d}\mathbb{P}^{u_0}} = \tau\big(\log\frac{\mathrm{d}\mathbb{Q}}{\mathrm{d}\mathbb{P}^{u_0}}\big) - A(\tau)$. Hence, we can rewrite (78) as

$$\mathcal{I}(\tau) = \frac{1}{\tau^2}\mathrm{Var}_{\mathbb{Q}^{(\tau)}}\left[\log\left(\frac{\mathrm{d}\mathbb{Q}^{(\tau)}}{\mathrm{d}\mathbb{P}^{u_0}}\right)\right],\tag{85}$$

which provides a way to estimate $\mathcal{I}(\tau)$, leveraging the Girsanov theorem to estimate $\log\big(\frac{\mathrm{d}\mathbb{Q}^{(\tau)}}{\mathrm{d}\mathbb{P}^{u_0}}\big) = \log\big(\frac{\mathrm{d}\mathbb{Q}^{(\tau)}}{\mathrm{d}\mathbb{P}}\big) - \log\big(\frac{\mathrm{d}\mathbb{P}^{u_0}}{\mathrm{d}\mathbb{P}}\big)$.

**Remark F.10** (Analytical computation of annealing sequence). Using $\tau + \Delta\tau = \beta_{i+1}$, $\tau = \beta_i$ paired with (81) and (78) we can analytically compute the annealing sequence $(\beta_i)_i$, up to high order terms, as

$$\beta_{i+1} = \beta_i + \sqrt{\frac{2\varepsilon}{\mathcal{I}(\beta_i)}} + O((\beta_{i+1} - \beta_i)^{3/2}),\tag{86}$$

with

$$\mathcal{I}(\beta_i) = \mathrm{Var}_{\mathbb{P}^{u_i}}\left[\log\left(\frac{\mathrm{d}\mathbb{Q}}{\mathrm{d}\mathbb{P}^{u_0}}\right)\right],\tag{87}$$

where we used that $\mathbb{Q}^{(\tau)} = \mathbb{P}^{u_i}$.

# G    Trust region SOC losses

In this section, we provide a non-exhaustive list of losses that can be readily applied to solve SOC problems within our trust region framework. More specifically, we aim for minimizing a divergence $D : \mathcal{P} \times \mathcal{P} \to \mathbb{R}^+$ between the path measures induced by the control $u_{i+1}$ and the learnable control $u$. For a comprehensive overview of SOC losses without trust regions, see [40].

## G.1    Log-variance loss

Here, we provide further details on the log-variance loss [87, 97, 98] within our trust region framework. The log-variance loss is defined as

$$\mathcal{L}_{\mathrm{LV}}(u) = \mathrm{Var}\left[\log\left(\frac{\mathrm{d}\mathbb{P}^{u_{i+1}}}{\mathrm{d}\mathbb{P}^u}(X^w)\right)\right] \quad \text{with} \quad X^w \sim \mathbb{P}^w,\tag{88}$$

where $X^w$ is defined as in (1), with $u$ replaced by $w \in \mathcal{U}$, referred to as the *reference process*. Although the choice of $w$ is arbitrary, we discuss two particularly suitable options that facilitate sample reuse with replay buffers in combination with trust regions.

**Using $w = u_i$ as reference control.** First, we replace the reference control $w$ with the control function of the previous iteration $u_i$. Thus, the log-variance loss becomes

$$\mathcal{L}_{\mathrm{LV}}(u) = \mathrm{Var}\left[\log\left(\frac{\mathrm{d}\mathbb{P}^{u_{i+1}}}{\mathrm{d}\mathbb{P}^u}(X^{u_i})\right)\right] = \mathrm{Var}\left[\log\left(\frac{\mathrm{d}\mathbb{P}^{u_{i+1}}}{\mathrm{d}\mathbb{P}^{u_i}}(X^{u_i})\frac{\mathrm{d}\mathbb{P}^{u_i}}{\mathrm{d}\mathbb{P}^u}(X^{u_i})\right)\right].\tag{89}$$

The Girsanov theorem (see App. A.3) shows that

$$\log\left(\frac{\mathrm{d}\mathbb{P}^{u_i}}{\mathrm{d}\mathbb{P}^u}(X^{u_i})\right) = \frac{1}{2}\int_0^T \|u_i(X_s^{u_i}, s) - u(X_s^{u_i}, s)\|^2 \mathrm{d}s + \int_0^T (u_i - u)(X_s^{u_i}, s)\cdot\mathrm{d}W_s.\tag{90}$$

Combining this result for $u = 0$ with Prop. 2.2, we obtain

$$\frac{\mathrm{d}\mathbb{P}^{u_{i+1}}}{\mathrm{d}\mathbb{P}^{u_i}}(X^{u_i}) \propto \left(\frac{\mathrm{d}\mathbb{Q}}{\mathrm{d}\mathbb{P}}\frac{\mathrm{d}\mathbb{P}}{\mathrm{d}\mathbb{P}^{u_i}}(X^{u_i})\right)^{\frac{1}{1+\lambda_i}}\tag{91a}$$

$$= e^{-\frac{1}{1+\lambda_i}\left(\int_0^T \frac{1}{2}\|u_i(X_s^{u_i}, s)\|^2 \mathrm{d}s + \int_0^T u_i(X_s^{u_i}, s)\cdot\mathrm{d}W_s + \mathcal{W}(X^{u_i}, 0)\right)}.\tag{91b}$$

Noting that the variance is shift-invariant, (90) and (91) imply that

$$\mathcal{L}_{\mathrm{LV}}(u) = \mathrm{Var}\left[-\frac{1}{1+\lambda_i}\left(\frac{1}{2}\int_0^T \|u_i(X_s^{u_i}, s)\|^2 \mathrm{d}s + \int_0^T u_i(X_s^{u_i}, s)\cdot\mathrm{d}W_s + \mathcal{W}(X^{u_i}, 0)\right)\right.$$
$$\left. + \frac{1}{2}\int_0^T \|u_i(X_s^{u_i}, s) - u(X_s^{u_i}, s)\|^2 \mathrm{d}s + \int_0^T (u_i - u)(X_s^{u_i}, s)\cdot\mathrm{d}W_s\right],\tag{92}$$

which can be implemented by discretizing the integrals; see App. E.2.

Please note that the loss reduces to

$$\mathcal{L}_{\mathrm{LV}}(u) = \mathrm{Var}\left[\log\left(\frac{\mathrm{d}\mathbb{Q}}{\mathbb{P}^{u_i}}(X^{u_i})\frac{\mathrm{d}\mathbb{P}^{u_i}}{\mathrm{d}\mathbb{P}^u}(X^{u_i})\right)\right] = \mathrm{Var}\left[\log\left(\frac{\mathrm{d}\mathbb{Q}}{\mathrm{d}\mathbb{P}^u}(X^{u_i})\right)\right]\tag{93}$$

for $\lambda_i = 0$, which is how the loss is mostly used in the literature, where the variance is computed using the most recent control, see e.g. [97].

**Using $w = u_{i+1}$ as reference control.** Alternatively, when setting the reference control to the optimal control $u_{i+1}$, the log-variance loss is given by

$$\mathcal{L}_{\mathrm{LV}}(u) = \mathrm{Var}\left[\log\left(\frac{\mathrm{d}\mathbb{P}^{u_{i+1}}}{\mathrm{d}\mathbb{P}^u}(X^{u_{i+1}})\right)\right] = \tag{94a}$$

$$= \mathbb{E}\left[\log\left(\frac{\mathrm{d}\mathbb{P}^{u_{i+1}}}{\mathrm{d}\mathbb{P}^u}(X^{u_{i+1}})\right)^2\right] - \left(\mathbb{E}\left[\log\left(\frac{\mathrm{d}\mathbb{P}^{u_{i+1}}}{\mathrm{d}\mathbb{P}^u}(X^{u_{i+1}})\right)\right]\right)^2 \tag{94b}$$

$$= \mathbb{E}\left[\log\left(\frac{\mathrm{d}\mathbb{P}^{u_{i+1}}}{\mathrm{d}\mathbb{P}^u}(X^{u_i})\right)^2 \frac{\mathrm{d}\mathbb{P}^{u_{i+1}}}{\mathrm{d}\mathbb{P}^{u_i}}(X^{u_i})\right] - \left(\mathbb{E}\left[\log\left(\frac{\mathrm{d}\mathbb{P}^{u_{i+1}}}{\mathrm{d}\mathbb{P}^u}(X^{u_i})\right)\frac{\mathrm{d}\mathbb{P}^{u_{i+1}}}{\mathrm{d}\mathbb{P}^{u_i}}(X^{u_i})\right]\right)^2 \tag{94c}$$

which can be computed using (90) and (91). Hence, in contrast to using $w = u_i$, (94) additionally incorporates the smoothed importance weights $\mathrm{d}\mathbb{P}^{u_{i+1}}/\mathrm{d}\mathbb{P}^{u_i}$.

## G.2 Moment loss

The *moment loss* was introduced in [59] and is defined as

$$\mathcal{L}_{\mathrm{moment}}(u) = \mathbb{E}\left[\log\left(\frac{\mathrm{d}\mathbb{P}^{u_{i+1}}}{\mathrm{d}\mathbb{P}^u}(X^w)\right)^2\right] \quad \text{with} \quad X^w \sim \mathbb{P}^w, \tag{95}$$

where $w \in \mathcal{U}$ is again an arbitrary reference control; see G.1. We distinguish again between $u = u_i$ and $w = u_{i+1}$.

**Using $w = u_i$ as reference control.** In this case, $\mathcal{L}_{\mathrm{moment}}$ becomes

$$\mathcal{L}_{\mathrm{moment}}(u) = \mathbb{E}\left[\log\left(\frac{\mathrm{d}\mathbb{P}^{u_{i+1}}}{\mathrm{d}\mathbb{P}^u}(X^{u_i})\right)^2\right] = \mathbb{E}\left[\log\left(\frac{\mathrm{d}\mathbb{P}^{u_{i+1}}}{\mathrm{d}\mathbb{P}^{u_i}}(X^{u_i})\frac{\mathrm{d}\mathbb{P}^{u_i}}{\mathrm{d}\mathbb{P}^u}(X^{u_i})\right)^2\right], \tag{96}$$

with

$$\frac{\mathrm{d}\mathbb{P}^{u_{i+1}}}{\mathrm{d}\mathbb{P}^{u_i}}(X^{u_i}) = \frac{e^{-\frac{1}{1+\lambda_i}\mathcal{W}_i(X^{u_i},0)}}{\mathcal{Z}_{i+1}} \quad \text{with} \quad \mathcal{Z}_{i+1} = \mathbb{E}\left[e^{-\frac{1}{1+\lambda_i}\mathcal{W}_i(X^{u_i},0)}\right]. \tag{97}$$

Contrary to the log-variance loss, the moment loss is not shift-invariant, thus requiring $\mathcal{Z}_{i+1}$ which is commonly not available. As such, [59] proposes to treat $\mathcal{Z}_{i+1}$ as a learnable parameter. Using (90) and (91) imply that

$$\mathcal{L}_{\mathrm{moment}}(u, \mathcal{Z}_{i+1}) = \mathbb{E}\left[\left(-\frac{1}{1+\lambda_i}\left(\frac{1}{2}\int_0^T \|u_i(X_s^{u_i},s)\|^2 \mathrm{d}s + \int_0^T u_i(X_s^{u_i},s)\cdot \mathrm{d}W_s + \mathcal{W}(X^{u_i},0)\right)\right.\right.$$
$$\left.\left. + \frac{1}{2}\int_0^T \|u_i(X_s^{u_i},s) - u(X_s^{u_i},s)\|^2 \mathrm{d}s + \int_0^T (u_i - u)(X_s^{u_i},s)\cdot \mathrm{d}W_s - \log\mathcal{Z}_{i+1}\right)^2\right], \tag{98}$$

which is optimized as $\min_{u \in \mathcal{U},\ \mathcal{Z}_{i+1} \in \mathbb{R}} \mathcal{L}_{\mathrm{moment}}(u, \mathcal{Z}_{i+1})$.

**Using $w = u_{i+1}$ as reference control.** Using $w = u_{i+1}$ yields

$$\mathcal{L}_{\mathrm{moment}}(u, \mathcal{Z}_{i+1}) = \mathbb{E}\left[\log\left(\frac{\mathrm{d}\mathbb{P}^{u_{i+1}}}{\mathrm{d}\mathbb{P}^u}(X^{u_{i+1}})\right)^2\right] \tag{99}$$

$$= \mathbb{E}\left[\log\left(\frac{\mathrm{d}\mathbb{P}^{u_{i+1}}}{\mathrm{d}\mathbb{P}^{u_i}}(X^{u_i})\frac{\mathrm{d}\mathbb{P}^{u_i}}{\mathrm{d}\mathbb{P}^u}(X^{u_i})\right)^2 \frac{\mathrm{d}\mathbb{P}^{u_{i+1}}}{\mathrm{d}\mathbb{P}^{u_i}}(X^{u_i})\right], \tag{100}$$

where $\mathrm{d}\mathbb{P}^{u_{i+1}}/\mathrm{d}\mathbb{P}^{u_i}$ depends on $\mathcal{Z}_{i+1}$, see (97). Hence, the difference between using $w = u_i$ and $w = u_{i+1}$ lies in the additional importance weights $\mathrm{d}\mathbb{P}^{u_{i+1}}/\mathrm{d}\mathbb{P}^{u_i}$.

### G.3 Cross-entropy loss

The cross entropy loss is defined as the forward KL divergence between $u_{i+1}$ and $u$, i.e.,

$$\mathcal{L}_{\mathrm{CE}}(u) = D_{\mathrm{KL}}\left(\mathbb{P}^{u_{i+1}}|\mathbb{P}^u\right) = \mathbb{E}\left[\log \frac{\mathrm{d}\mathbb{P}^{u_{i+1}}}{\mathrm{d}\mathbb{P}^u}(X^{u_{i+1}})\right] \tag{101a}$$

$$= \mathbb{E}\left[\log\left(\frac{\mathrm{d}\mathbb{P}^{u_{i+1}}}{\mathrm{d}\mathbb{P}^{u_i}}(X^{u_{i+1}})\frac{\mathrm{d}\mathbb{P}^{u_i}}{\mathrm{d}\mathbb{P}^u}(X^{u_{i+1}})\right)\right] \tag{101b}$$

$$= \mathbb{E}\left[\log\left(\frac{\mathrm{d}\mathbb{P}^{u_{i+1}}}{\mathrm{d}\mathbb{P}^{u_i}}(X^{u_i})\frac{\mathrm{d}\mathbb{P}^{u_i}}{\mathrm{d}\mathbb{P}^u}(X^{u_i})\right)\frac{\mathrm{d}\mathbb{P}^{u_{i+1}}}{\mathrm{d}\mathbb{P}^{u_i}}(X^{u_i})\right]. \tag{101c}$$

Using (90) and (91) implies that

$$\mathcal{L}_{\mathrm{CE}}(u) = \mathbb{E}\left[\left(-\frac{1}{1+\lambda_i}\left(\frac{1}{2}\int_0^T \|u_i(X_s^{u_i},s)\|^2\mathrm{d}s + \int_0^T u_i(X_s^{u_i},s)\cdot\mathrm{d}W_s + \mathcal{W}(X^{u_i},0)\right)\right.\right.$$
$$\left.\left. + \frac{1}{2}\int_0^T \|u_i(X_s^{u_i},s) - u(X_s^{u_i},s)\|^2\mathrm{d}s + \int_0^T (u_i - u)(X_s^{u_i},s)\cdot\mathrm{d}W_s\right)\frac{\mathrm{d}\mathbb{P}^{u_{i+1}}}{\mathrm{d}\mathbb{P}^{u_i}}(X^{u_i})\right] - \log\mathcal{Z}, \tag{102}$$

with importance weights $\mathrm{d}\mathbb{P}^{u_{i+1}}/\mathrm{d}\mathbb{P}^{u_i}$.

### G.4 Stochastic optimal control matching via adjoint method

Here, we provide further details on the trust region version of the stochastic optimal control matching (SOCM) loss introduced in [42]. We start from the cross-entropy loss, i.e., the forward KL divergence between $u_{i+1}$ and $u$, that is,

$$\mathcal{L}_{\mathrm{CE}}(u) = D_{\mathrm{KL}}\left(\mathbb{P}^{u_{i+1}}|\mathbb{P}^u\right) = \mathbb{E}\left[\log \frac{\mathrm{d}\mathbb{P}^{u_{i+1}}}{\mathrm{d}\mathbb{P}^u}(X^{u_{i+1}})\right]. \tag{103}$$

Using Girsanov's theorem (see App. A.3), the cross-entropy loss can be written as

$$\mathcal{L}_{\mathrm{CE}}(u) = \mathbb{E}\left[\frac{1}{2}\int_0^T \|u_{i+1}(X_s^{u_{i+1}},s) - u(X_s^{u_{i+1}},s)\|^2\mathrm{d}s\right] \tag{104a}$$

$$= \mathbb{E}\left[\frac{1}{2}\int_0^T \|u_{i+1}(X_s^{u_i},s) - u(X_s^{u_i},s)\|^2\mathrm{d}s\frac{\mathrm{d}\mathbb{P}^{u_{i+1}}}{\mathrm{d}\mathbb{P}^{u_i}}\right]. \tag{104b}$$

Using the expression for the optimal control, $u_{i+1} = \frac{\lambda_i}{1+\lambda_i}u_i - \frac{1}{1+\lambda_i}\nabla V_{i+1}$, see Prop. 2.5, yields

$$\mathcal{L}_{\mathrm{CE}}(u) = \mathbb{E}\left[\frac{1}{2}\int_0^T \|\tfrac{\lambda_i}{1+\lambda_i}u_i(X_s^{u_i},s) - \tfrac{1}{1+\lambda_i}\sigma^\top\nabla V_{i+1}(X_s^{u_i},s) - u(X_s^{u_i},s)\|^2\mathrm{d}s\frac{\mathrm{d}\mathbb{P}^{u_{i+1}}}{\mathrm{d}\mathbb{P}^{u_i}}\right] \tag{105}$$

with

$$\nabla_x V_{i+1}(x,t) = -(1+\lambda_i)\frac{\nabla_x\mathbb{E}\left[e^{-\frac{1}{1+\lambda_i}\mathcal{W}_i(X^{u_i},t)}\Big|X_t^{u_i}=x\right]}{\mathbb{E}\left[e^{-\frac{1}{1+\lambda_i}\mathcal{W}_i(X^{u_i},t)}\Big|X_t^{u_i}=x\right]}. \tag{106}$$

We use the adjoint method [41, see Lemma 5] to evaluate the conditional expectation (106)[10], giving

$$\nabla_x\mathbb{E}\left[e^{-\frac{1}{1+\lambda_i}\mathcal{W}_i(X^{u_i},t)}\Big|X_t^{u_i}=x\right] = \mathbb{E}\left[\widetilde{a}(t,u_i,X^{u_i})e^{-\frac{1}{1+\lambda_i}\mathcal{W}_i(X^{u_i},t)}\Big|X_t^{u_i}=x\right] \tag{107}$$

where the adjoint state $\widetilde{a}(t,u_i,X^{u_i})$ satisfies the ordinary differential equation (ODE)

$$\frac{\mathrm{d}}{\mathrm{d}s}\widetilde{a}(s,u_i,X_s^{u_i}) = -\Big[(\nabla(b(X_s^{u_i},s) + \sigma u_i(X_s^{u_i},s))^\top\widetilde{a}(u_i,X_s^{u_i},s) \tag{108}$$

$$+ \tfrac{1}{1+\lambda_i}\nabla(f(X_s^{u_i},s) + \tfrac{1}{2}\|u_i(X_s^{u_i},s)\|^2)\Big] \tag{109}$$

with $\widetilde{a}(T,u_i,X_T^{u_i}) = \frac{1}{1+\lambda_i}\nabla g(X_T)$. Using the argument from [42, Theorem 1], replacing the path-wise reparameterization trick with the adjoint method, we arrive at the trust region version of the stochastic optimal control loss given by

$$\mathcal{L}_{\mathrm{SOCM}}(u) = \mathbb{E}\left[\frac{1}{2}\int_0^T \|\tfrac{\lambda_i}{1+\lambda_i}u_i(X^{u_i},s) - \sigma^\top\widetilde{a}(u_i,X_s^{u_i},s) - u(X^{u_i},s)\|^2\mathrm{d}s\frac{\mathrm{d}\mathbb{P}^{u_{i+1}}}{\mathrm{d}\mathbb{P}^{u_i}}\right] + K \tag{110}$$

---

[10]Note that there exist other methods for computing derivatives of functionals of stochastic processes. We refer the interested reader to [42].

for some $K$ independent of $u$. However, the adjoint state contains the Jacobian $\nabla u_i$ and the derivative $\nabla \|u_i\|$, which can be expensive in practice. In what follows, we rewrite the objective such that we can get rid of these terms.

### G.5 Stochastic optimal control matching via lean adjoint method

Starting again from the cross-entropy loss, we now employ the alternative expression for the optimal control as stated in Item (ii). This yields the objective

$$\mathcal{L}_{\mathrm{CE}}(u) = \mathbb{E}\left[\frac{1}{2}\int_0^T \| -\sigma^\top \nabla \widetilde{V}_{i+1}(X_s, s) - u(X_s, s)\|^2 \mathrm{d}s \frac{\mathrm{d}\mathbb{P}^{u_{i+1}}}{\mathrm{d}\mathbb{P}}\right], \tag{111}$$

where the gradient of the smoothed value function is given by

$$\nabla_x \widetilde{V}_{i+1}(x, t) = -\frac{\nabla_x \mathbb{E}\left[e^{-\beta_{i+1}\mathcal{W}(X_t, 0)}|X_t = x\right]}{\mathbb{E}\left[e^{-\beta_{i+1}\mathcal{W}(X_t, 0)}|X_t = x\right]}. \tag{112}$$

We evaluate the conditional expectation using the adjoint method:

$$\nabla_x \mathbb{E}\left[e^{-\beta_{i+1}\mathcal{W}(X_t, 0)}|X_t = x\right] = \mathbb{E}\left[a_{i+1}(X_s, s)e^{-\beta_{i+1}\mathcal{W}(X_t, 0)}|X_t = x\right], \tag{113}$$

where $a_{i+1}(X_s, s)$ denotes the lean adjoint state [41], which satisfies the backward differential equation

$$\frac{\mathrm{d}}{\mathrm{d}s}a_{i+1}(X_s, s) = -\left[(\nabla b(X_s, s)^\top a_{i+1}(X_s, s) + \beta_{i+1}\nabla f(X_s, s)\right] \tag{114}$$

with terminal condition $a_{i+1}(X_T, T) = \beta_{i+1}\nabla g(X_T)$. Following the derivations in [42], we arrive at the objective

$$\mathcal{L}_{\mathrm{SOCM}}(u) = \mathbb{E}\left[\frac{1}{2}\int_0^T \|\sigma^\top a_{i+1}(X_s, s) - u(X_s, s)\|^2 \mathrm{d}s \frac{\mathrm{d}\mathbb{P}^{u_{i+1}}}{\mathrm{d}\mathbb{P}}(X)\right]. \tag{115}$$

Finally, performing a change of measure to the previous control $u_i$ gives the expression:

$$\mathcal{L}_{\mathrm{SOCM}}(u) = \mathbb{E}\left[\frac{1}{2}\int_0^T \|\sigma^\top a_{i+1}(X_s^{u_i}, s) - u(X_s^{u_i}, s)\|^2 \mathrm{d}s \frac{\mathrm{d}\mathbb{P}^{u_{i+1}}}{\mathrm{d}\mathbb{P}^{u_i}}(X^{u_i})\right]. \tag{116}$$

We remark that the adjoint ODE in (114) can be solved as

$$a_{i+1}(X_s, s) = \beta_{i+1}\exp\left(\int_s^T \nabla b(X_t, t)^\top \mathrm{d}t\right)\nabla g(X_T) \tag{117}$$

if $f = 0$ and $\nabla b(X_t, t)\nabla b(X_s, s) = \nabla b(X_s, s)\nabla b(X_t, t)$ for all $s, t \in [0, T]$ (i.e., the matrices at different times commute). This allows us to solve the adjoint ODE exactly for our applications of sampling from unnormalized densities; see App. I.

**Extensions for diffusion-based sampling.** Consider the case where $f = 0$ and $b(x, t) = b_1(t)x$ with $b_1 \in C([0, T], \mathbb{R})$, which holds in certain settings for diffusion-based sampling [129, 149]. In this case (117) becomes

$$a_{i+1}(X_T, s) = \beta_{i+1}\gamma(s)\nabla g(X_T) \quad \text{with} \quad \gamma(s) := \exp\left(\int_s^T b_1(t)\mathrm{d}t\right). \tag{118}$$

The SOCM loss in (115) therefore reads

$$\mathcal{L}_{\mathrm{SOCM}}(u) = \mathbb{E}_{\mathbb{P}^{u_{i+1}}}\left[\frac{1}{2}\int_0^T \|\beta_{i+1}\gamma(s)\sigma^\top \nabla g(X_T^{u_{i+1}}) - u(X_s^{u_{i+1}}, s)\|^2 \mathrm{d}s\right]. \tag{119}$$

From Prop. E.2 Item (ii) it directly follows that

$$\frac{\mathrm{d}\mathbb{P}^{u_{i+1}}}{\mathrm{d}\mathbb{P}^{u_i}}(X) = \frac{\mathrm{d}\mathbb{P}^{u_{i+1}}}{\mathrm{d}\mathbb{P}}(X)\frac{\mathrm{d}\mathbb{P}}{\mathrm{d}\mathbb{P}^{u_i}}(X) \propto e^{-(\beta_{i+1}-\beta_i)g(X_T)} \tag{120}$$

for $u_0 = 0$. Thus, the SOCM loss can be rewritten as

$$\mathcal{L}_{\mathrm{SOCM}}(u) = \mathbb{E}_{\mathbb{P}^{u_i}}\left[\frac{1}{2}\int_0^T \|\beta_{i+1}\gamma(s)\sigma^\top \nabla g(X_T^{u_i}) - u(X_s^{u_i}, s)\|^2 \mathrm{d}s \frac{\mathrm{d}\mathbb{P}^{u_{i+1}}}{\mathrm{d}\mathbb{P}^{u_i}}(X^{u_i})\right] \tag{121a}$$

$$\propto \mathbb{E}_{\mathbb{P}^{u_i}}\left[\frac{1}{2}\int_0^T \|\beta_{i+1}\gamma(s)\sigma^\top \nabla g(X_T^{u_i}) - u(X_s^{u_i}, s)\|^2 \mathrm{d}s\, e^{-(\beta_{i+1}-\beta_i)g(X_T^{u_i})}\right]. \tag{121b}$$

Lastly, using that $\mathbb{P}^{u_i} = \mathbb{P}^{u_i}_{\cdot|T}\mathbb{P}^{u_i}_T = \mathbb{P}_{\cdot|T}\mathbb{P}^{u_i}_T$, we arrive at

$$\mathcal{L}_{\text{SOCM}}(u) \propto \mathbb{E}_{\mathbb{P}^{u_i}_T \mathbb{P}_{\cdot|T}}\left[\frac{1}{2}\int_0^T \|\beta_{i+1}\gamma(s)\sigma^\top\nabla g(X_T^{u_i}) - u(X_s^{u_i}, s)\|^2 \mathrm{d}s \, e^{-(\beta_{i+1}-\beta_i)g(X_T^{u_i})}\right] \quad (122\text{a})$$

$$= \int_0^T \mathbb{E}_{\mathbb{P}^{u_i}_T \mathbb{P}_{s|T}}\left[\frac{1}{2}\|\beta_{i+1}\gamma(s)\sigma^\top\nabla g(X_T^{u_i}) - u(X_s^{u_i}, s)\|^2 \, e^{-(\beta_{i+1}-\beta_i)g(X_T^{u_i})}\right]\mathrm{d}s, \quad (122\text{b})$$

where we marginalized out all $t \in [0, T)$ except for $t = s$. Note that for certain $b_1$, $\mathbb{P}_{s|T}$ can be sampled directly, removing the necessity for storing intermediate samples in the buffer.

## H   Trust regions for probability measures

Our goal is to sample from a probability density of the form

$$p_{\text{target}}(x) = \frac{\rho_{\text{target}}(x)}{\mathcal{Z}}, \quad \text{with} \quad \mathcal{Z} = \int \rho_{\text{target}}(x)\mathrm{d}x, \quad (123)$$

where we can evaluate $\rho_{\text{target}}$ but typically do not have access to samples from $p_{\text{target}}$. To tackle this problem, one can again formulate this problem as a variational problem by minimizing a divergence between some $q$ and the target density $p_{\text{target}}$. We can again incorporate an additional trust region constraint, that is, an upper bound on the change of the variational distribution $q$ within a single update step. Formally, we are trying to solve the following problem:

$$q_{i+1} = \arg\min_q D_{\text{KL}}(q\|p_{\text{target}}) \quad \text{s.t.} \quad D_{\text{KL}}(q\|q_i) \leq \varepsilon, \quad \int \mathrm{d}q = 1, \quad (124)$$

where $q_i$ is the variational distribution from the previous iteration. We again tackle the constrained optimization problem in (124) using Lagrangian multipliers. The Lagrangian is given by

$$\mathcal{L}_{\text{TR}}^{(i)}(q, \lambda, \omega) = D_{\text{KL}}(q\|p_{\text{target}}) + \lambda\left(D_{\text{KL}}(q\|q_i) - \varepsilon\right) + \omega\left(\int \mathrm{d}q - 1\right) \quad (125)$$

with Lagrangian multipliers $\lambda, \omega$. Taking the functional derivative $\delta\mathcal{L}_{\text{TR}}^{(i)}(q, \lambda, \omega)/\delta q$ and setting it to zero admits a closed-form solution for the optimal density $q_{i+1}$ as the geometric average between the old distribution and the (unnormalized) optimal distribution, that is,[11]

$$q_{i+1}(\lambda) = \arg\min_q \mathcal{L}_{\text{TR}}^{(i)}(q, \lambda) = \frac{q_i^{\frac{\lambda}{1+\lambda}}\rho_{\text{target}}^{\frac{1}{1+\lambda}}}{\mathcal{Z}_i(\lambda)}, \quad \text{with} \quad \mathcal{Z}_i(\lambda) = \int \mathrm{d}q_i^{\frac{\lambda}{1+\lambda}}\rho_{\text{target}}^{\frac{1}{1+\lambda}}. \quad (126)$$

Plugging the optimal distribution back into the Lagrangian yields the dual function

$$\mathcal{L}_{\text{Dual}}^{(i)}(\lambda) = \mathcal{L}_{\text{TR}}^{(i)}(q_{i+1}(\lambda), \lambda) = -(1 + \lambda)\log \mathcal{Z}_i(\lambda) - \lambda\varepsilon. \quad (127)$$

Note that we can use any non-linear optimizer for solving for the optimal Lagrangian multiplier by maximizing the dual function, i.e.,

$$\lambda_i = \arg\max_{\lambda \in \mathbb{R}^+} \mathcal{L}_{\text{Dual}}^{(i)}(\lambda). \quad (128)$$

## I   Diffusion-based sampling

We consider the task of sampling from densities of the form

$$p_{\text{target}} = \frac{\rho_{\text{target}}}{\mathcal{Z}} \quad \text{with} \quad \mathcal{Z} := \int_{\mathbb{R}^d} \rho_{\text{target}}(x)\mathrm{d}x, \quad (129)$$

where $\rho_{\text{target}} \in C(\mathbb{R}^d, \mathbb{R}_{\geq 0})$ can be evaluated pointwise, but the normalizing constant $\mathcal{Z}$ is typically intractable.

Here, we approach the sampling problem by using denoising diffusion-based sampling based on the work of [129] (see [15, 97] for a generalization). To that end, we consider a controlled ergodic Ornstein-Uhlenbeck (OU) process $X = (X_s)_{s\in[0,T]}$, i.e.,

$$\mathrm{d}X_s^u = (-\zeta(s)X_s^u + u(X_s^u, s))\,\mathrm{d}s + \eta\sqrt{2\zeta(s)}\,\mathrm{d}W_s, \qquad X_0 \sim p_0, \quad (130)$$

with noise schedule $\zeta \in C([0, T], \mathbb{R})$, $p_0(x) = \mathcal{N}(0, \eta^2 I)$ and corresponding path measure $\mathbb{P}^u$. The target path space measure $\mathbb{Q}$ is induced by an uncontrolled ergodic Ornstein-Uhlenbeck (OU) process, starting from the target $p_{\text{target}}$ and running backward in time, that is,

$$\mathrm{d}X_s = \zeta(s)X_s\,\mathrm{d}s + \eta\sqrt{2\zeta(s)}\,\mathrm{d}W_s, \qquad X_T \sim p_{\text{target}}, \quad (131)$$

---

[11]Note the dependence of $\mathcal{L}_{\text{TR}}^{(i)}$ on $\omega$ vanishes as $q_{i+1}$ satisfies the normalization constraint.

which fulfills $\mathbb{Q}_0 \approx p_0$ for a suitable choice of $\zeta$. For integration, we follow [129] and use an exponential integrator. Lastly, it can be shown that the optimal control fulfills

$$u^*(x, s) = \eta\sqrt{2\zeta(s)}\nabla_x \log \frac{\mathbb{Q}_s}{\mathbb{P}_s}(x), \tag{132}$$

which is later used to analytically compute the optimal control for Gaussian mixture model target densities, see e.g. [129]. Please note that $\mathbb{P}_s = \mathcal{N}(0, \eta^2 I)$ for all $s \in [0, T]$ as the uncontrolled SDE is initialized at its equilibrium distribution.

## I.1   Experimental setup

Here, we provide further details on our experimental setup.

**General setting.** The codebase used in this work was developed from scratch but is loosely inspired by `github.com/facebookresearch/SOC-matching`. All experiments are conducted using the Jax library [21] and are run on a single 40GB NVIDIA A40 GPU. Our default experimental setup, unless specified otherwise, is as follows: We use the Adam optimizer [79] with a learning rate of $5 \times 10^{-4}$ and gradient clipping with a value of 1. We utilized 50 discretization steps using exponential integrators. The control function $u$ is parameterized as a fully-connected 6-layer neural network with 256 neurons and GELU activations [67]. Time embedding is achieved via Fourier features [122]. For all experiments, we used a time horizon of $T = 1$.

The control is parameterized as

$$u^\theta(x, t) = f_1^\theta(x, t) + f_2^\theta(t)\frac{x}{\eta^2}, \tag{133}$$

and for experiments using Langevin preconditioning (LP), it is parameterized as

$$u_{\mathrm{LP}}^\theta(x, t) = f_1^\theta(x, t) + f_2^\theta(t)\left(\frac{x}{\eta^2} + \nabla_x \log \rho_{\mathrm{target}}(x)\right), \tag{134}$$

where $f_1^\theta$ and $f_2^\theta$ are neural networks parameterized by $\theta$.

For non-trust methods, we train for $60k$ gradient steps with a batch size of 2000, amounting to a total of $120M$ target evaluations. In contrast, trust region methods use a buffer of length 50k refreshed 150 times during training, resulting in a total of $60k \times 150 = 7.5M$ target evaluations. To optimize for the next control $u_{i+1}$, we perform 400 gradient steps on the replay buffer using randomly sampled batches of size 2000. All experiments use a trust region bound of $\varepsilon = 0.1$. The dual function is optimized using a line search method.

For the *Many Well* target, we set the standard deviation of the prior distribution to 1 and to 2.5 for the Gaussian mixture target. For the randomization of the mixing weights, we uniformly sample positive values that are normalized and rescaled such that the ratio between the maximum mixing weight and the minimum is 3. The diffusivity is scheduled according to $\zeta(t) = (C_{\mathrm{max}} - C_{\mathrm{min}})\cos^2\left(\frac{t\pi}{2T}\right) + C_{\mathrm{min}}$ with $C_{\mathrm{min}} = 0.01$ and $C_{\mathrm{max}} = 10$.

**Evaluation protocol and model selection.** We follow the evaluation protocol of prior work [18] and evaluate all performance criteria 100 times during training, using 2000 samples for each evaluation. We apply a running average with a window of 5 evaluations to smooth out short-term fluctuations and obtain more robust results within a single run. We conducted each experiment using four different random seeds and averaged the best results for each run.

**Benchmark problem details.** The *Many Well* target involves a $d$-dimensional *double well* potential, corresponding to the (unnormalized) density

$$\rho_{\mathrm{target}}(x) = \exp\left(-\sum_{i=1}^{m}(x_i^2 - \delta)^2 - \frac{1}{2}\sum_{i=m+1}^{d} x_i^2\right),$$

with $m \in \mathbb{N}$ representing the number of combined double wells (resulting in $2^m$ modes), and a separation parameter $\delta \in (0, \infty)$ (see also [135]). In our experiments, we set $m = 5$ leading to $2^m = 32$ modes. The separation parameter is set to $\delta = 4$. Since $\rho_{\mathrm{target}}$ factorizes across dimensions, we can compute a reference solution for $\log \mathcal{Z}$ via numerical integration, as described in [84].

Moreover, we consider a Gaussian mixture model (GMM) target of the form

$$p_{\mathrm{target}}(x) = \sum_{k=1}^{K} \pi_k \mathcal{N}(x \mid \mu_k, \Sigma_k), \tag{135}$$

where $\mu_k \in \mathbb{R}^d$, $\Sigma_k \in \mathbb{R}^{d \times d}$, $\pi_k \geq 0$, and $\sum_{k=1}^K \pi_k = 1$. To compute the optimal control $u^*$, we exploit the fact that the optimal marginal path measures $\mathbb{Q}_t(x)$ can be derived analytically [86],

$$\mathbb{Q}_t(x) = \sum_{k=1}^K \pi_k \mathcal{N}\left(x \,\Big|\, \mu_k e^{-\int_t^T \zeta(s)\mathrm{d}s}, \Sigma_k e^{-2\int_t^T \zeta(s)\mathrm{d}s} + \eta^2 \int_t^T 2\zeta(s) e^{-2\int_t^s \zeta(u)\mathrm{d}u}\mathrm{d}s\right) \tag{136}$$

and used this for computing the optimal control $u^*$. Finally, to compute the total variation distance, we leverage the known true mixing weights $\pi_k$ and define the mode partitions $S_k \subset \mathbb{R}^d$ as

$$S_k = \{x \in \mathbb{R}^d | \arg\max_j \pi_j \mathcal{N}(x|\mu_j, \Sigma_j) = k\}. \tag{137}$$

## I.2 Evaluation criteria

Here, we provide further information on how our evaluation criteria are computed.

**Control $L^2$ error.** Assuming access to the optimal control $u^*$, we can compute the $L^2$ error between the optimal and the learned control, i.e.,

$$\text{control } L^2 \text{ error} := \mathbb{E}\left[\frac{1}{2}\int_0^T \|u^* - u\|^2(X^{u^*}, s)\mathrm{d}s\right], \tag{138}$$

where $X^{u^*}$ is obtained by simulating the controlled process with $u^*$, and compute the error using a Monte Carlo estimate. Note that this quantity is equivalent to the forward Kullback-Leibler divergence

$$D_{\text{KL}}(\mathbb{Q}|\mathbb{P}^u) = \mathbb{E}\left[\log\frac{\mathrm{d}\mathbb{Q}}{\mathrm{d}\mathbb{P}^u}(X^{u^*})\right]. \tag{139}$$

Via Girsanov's theorem (see App. A.3) we have that

$$\frac{\mathrm{d}\mathbb{Q}}{\mathrm{d}\mathbb{P}^u}(X^{u^*}) = \int_0^T (u^* - u)(X^{u^*}, s) \cdot \mathrm{d}W_s + \frac{1}{2}\int_0^T \|u^* - u\|^2(X^{u^*}, s)\mathrm{d}s. \tag{140}$$

The desired equivalence follows from the fact that, under mild regularity assumptions, the stochastic integral in (140) is a martingale and has vanishing expectation.

**Log-normalizing constant.** By definition, the log-normalizing constant is given by

$$\mathcal{Z}(X_0) = \mathbb{E}\left[e^{-\mathcal{W}(X,0)}\Big|X_0\right]. \tag{141}$$

Applying a change of measure to the controlled process yields

$$\mathcal{Z}(X_0) = \mathbb{E}\left[e^{-\mathcal{W}(X^u,0)}\frac{\mathrm{d}\mathbb{P}}{\mathrm{d}\mathbb{P}^u}(X^u)\Big|X_0\right] = \mathbb{E}\left[e^{-\int_0^T \frac{1}{2}\|u(X_s^u,s)\|^2\mathrm{d}s - \int_0^T u(X_s^u,s)\cdot\mathrm{d}W_s - \mathcal{W}(X^u,0)}\Big|X_0\right], \tag{142}$$

which can be estimated via Monte Carlo using samples from the current control $u$.

**Sinkhorn distance.** We estimate the Sinkhorn distance $\mathcal{W}_\gamma^2$ [31], an entropy-regularized optimal transport distance, between model and target samples using the JAX-based `ott` library [32].

**Total variation distance.** Inspired by recent work [18, 55], we assume access to ground truth mixing weights $\pi_k$, $k \in \{1, \ldots, K\}$, along with a partition $\{S_1, \ldots, S_K\}$ of $\mathbb{R}^d$, where each region $S_k \subset \mathbb{R}^d$ corresponds to the $k$-th mode of the target distribution. We estimate the empirical mixing weights using

$$\widehat{\pi}_k = \frac{\mathbb{E}\left[\mathbb{1}_{S_k}(X_T^u)\right]}{\sum_{k'=1}^K \mathbb{E}\left[\mathbb{1}_{S_{k'}}(X_T^u)\right]}. \tag{143}$$

Using these estimates, we compute the total variation distance (TVD) between the empirical and true mode weights as

$$\text{TVD} = \sum_{k=1}^K |\pi_k - \widehat{\pi}_k|. \tag{144}$$

Details on how the ground truth mixing weights and the corresponding mode regions $S_k$ are defined can be found in the descriptions of the target densities.

## I.3 Additional experiments

Here, we provide results for additional numerical experiments.

**Gaussian Mixture 40 (GMM40).** We further evaluate the performance of trust-region-based losses by comparing them to existing SOC losses on the well-established GMM40 benchmark [84]. In this task, the target distribution is a Gaussian mixture model with 40 components, where the means are uniformly sampled from the interval $[-40, 40]$, and each component has an initial variance of $1$. We

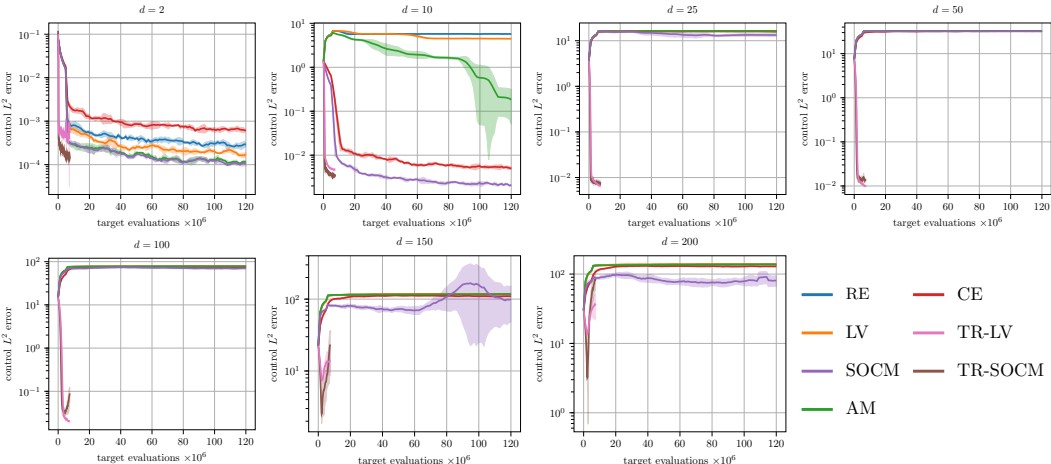

Figure 7: Control $L^2$ error as a function of the number of target evaluations for the GMM target across varying problem dimensionalities $d$. All results are averaged across four random seeds.

set the prior's standard deviation to $\eta = 30$. The results, presented in Fig. 8, show that only two losses, Cross-Entropy (CE) and trust region with log-variance (TR-LV), can consistently learn all 40 modes. Notably, TR-LV achieves this with approximately ten times fewer target evaluations than CE.

**Control $L^2$ error vs. target evaluations.** We extend the results presented in Sec. 3.1 for the GMM benchmark by providing a detailed analysis of the control $L^2$ error as a function of the number of target evaluations across varying problem dimensionalities $d$. For $d = 2$, all SOC losses achieve low control error. However, at $d = 10$, some methods begin to exhibit elevated control error due to mode collapse. As the dimensionality increases further, only trust-region-based losses consistently maintain low control error. While these methods show partial mode collapse for $d \geq 150$, we anticipate that this issue can be mitigated by refining the control function architecture or by employing larger buffer and batch sizes. Importantly, trust region methods also require significantly fewer target evaluations – a key advantage in many real-world applications where evaluations are costly.

**Influence of trust region bounds.** We further investigate the effect of different trust region bound values $\varepsilon$ on the GMM target using TR-LV. The results are presented in Fig. 10. The left figure shows that smaller trust region bounds significantly improve performance: $\varepsilon = 0.01$ yields up to an order of magnitude lower control error compared to $\varepsilon = 1$. Additionally, smaller $\varepsilon$ values help stabilize training, as evidenced by the reduced standard deviation across random seeds. In contrast, training with $\varepsilon = 1$ becomes unstable. However, this improved stability comes at the cost of slower convergence – smaller bounds require more training iterations to effectively anneal from the prior to the target path measure, as illustrated in the middle figure. Finally, the right figure shows that the empirically observed smoothed effective sample size (ESS) aligns well with its Taylor series approximation, $\text{ESS} = \left( \text{Var} \left( \frac{\mathrm{d}\mathbb{P}^{(i+1)}}{\mathrm{d}\mathbb{P}^{(i)}} \right) + 1 \right)^{-1} \approx \frac{1}{2\varepsilon+1}$ for small values of $\varepsilon$; see see App. E.3 for further details.

## J   Transition path sampling

### J.1   Experimental setup

We build upon the codebase provided by TPS-DPS [113] (`github.com/kiyoung98/tps-dps`). Our experimental setups also follow [113] to ensure a fair comparison. The dual function is optimized using Brent's method [23].

**MD simulation setup.** We run molecular dynamics simulation on the OpenMM platform. Both simulations are run at temperature 300K. For Alanine Dipeptide, we use the 'amber99sbildn.xml' forcefield with a VVVR integrator to simulate in vaccum. Each timestep is set as 1 femtosecond. Each path sampled is of length 1,000. For Chignolin, we use the 'protein.ff14SBonlysc.xml' forcefield with

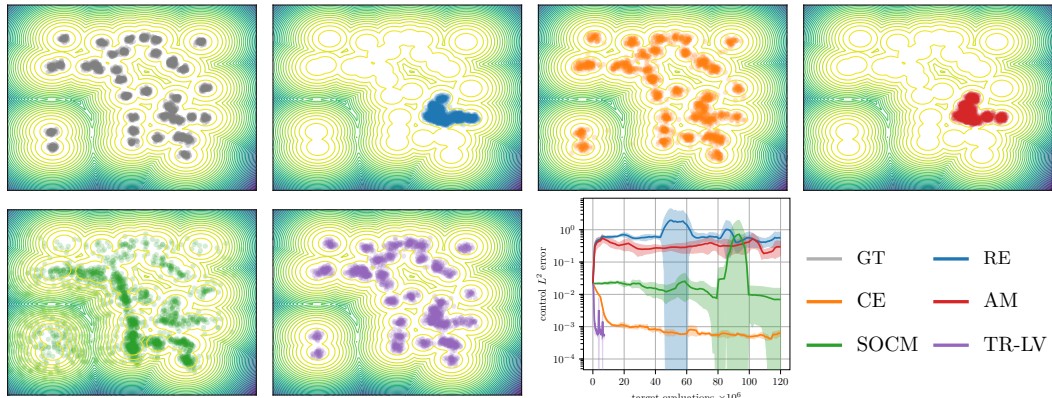

Figure 8: Qualitative and quantitative results for the GMM40 target. The qualitative plots demonstrate that only the CE (orange) and TR-LV (purple) losses successfully capture all 40 modes of the ground truth (GT, grey) distribution. This is further supported by the low $L^2$ control error observed for these two methods. Results are averaged across four random seeds and are not reported for the log-variance loss due to numerical instabilities.

implicit solvent model 'gbn2.xml' with a VVVR integrator. Each timestep is set as 1 femtosecond. Each path sampled is of length 5,000.

**Target hit.** For Alanine Dipeptide, target hit is defined over the two dihedral angles $\phi$ and $\psi$ and a distance radius within 0.75Å. For Chignolin, a long MD simulation is pre-loaded with Time-lagged independent component analysis (TICA) to select the first two dimensions that capture most variance. The region is then defined over the two dimensions with a radius of 0.75.

**Training process.** Annealing is applied from 600K to 300K. A replay buffer is used with buffer size 1,000 and 200 for Alanine Dipeptide and Chignolin, respectively, and training over buffer per iteration is 1,000 times.

**Hyperparameters.** The trust region constraint is set to $\varepsilon = 0.01$ for Alanine Dipeptide and $\varepsilon = 0.2$ for Chignolin. Batch size for both systems is set to 16, Alanine Dipeptide is trained for 2000 iterations, while Chignolin is trained for 50 iterations.

**Computing resources.** Each experiment is run on a single 80GB NVIDIA H100 GPU.

### J.2 Additional experimental result discussion

We discuss our results in comparison to [113]. First of all, we evaluate three seed average as we notice the high variance nature of the transition path sampling problem–running several times can have huge variance in results (also evidenced in Fig. 4). We can also observe the trust region constraint helps to stabilize the training significantly and thus have much smaller variance across three runs. Notably, for Alanine Dipeptide, both methods start with zero hitting percentage, while in Chignolin, in the beginning both methods already have some trajectories that hit the target, trust region constraint is already effective in improving the efficiency. We use almost the exact same setup as in [113] with the only difference being the batch size for Chignolin is 16 instead of 4. We do not tune the model as our goal is to show the trust region constraint improves the training stability and thus the efficiency and accuracy in terms of number of energy calls.

## K  Fine-tuning of diffusion models

We take the adjoint matching (AM) implementation in `github.com/microsoft/soc-fine-tuning-sd` as our baseline, and we modify it to implement TR-SOCM.

**Fine-tuning experimental details.** We generate images using classifier-free guidance, with guidance scale 7.5. We use 50 inference timesteps to sample the trajectories during fine-tuning, and the evaluation samples are also generated at 50 inference timesteps.

We fine-tune using the default hyperparameters in the repo: we use AdamW, using learning rate $3 \times 10^{-6}$, beta 1 set to 0.9, beta 2 set to 0.95, and weight decay 0. We use an effective batch size of 500 trajectories and 4 model backpropagations per trajectory. For the TR-SOCM loss, we use

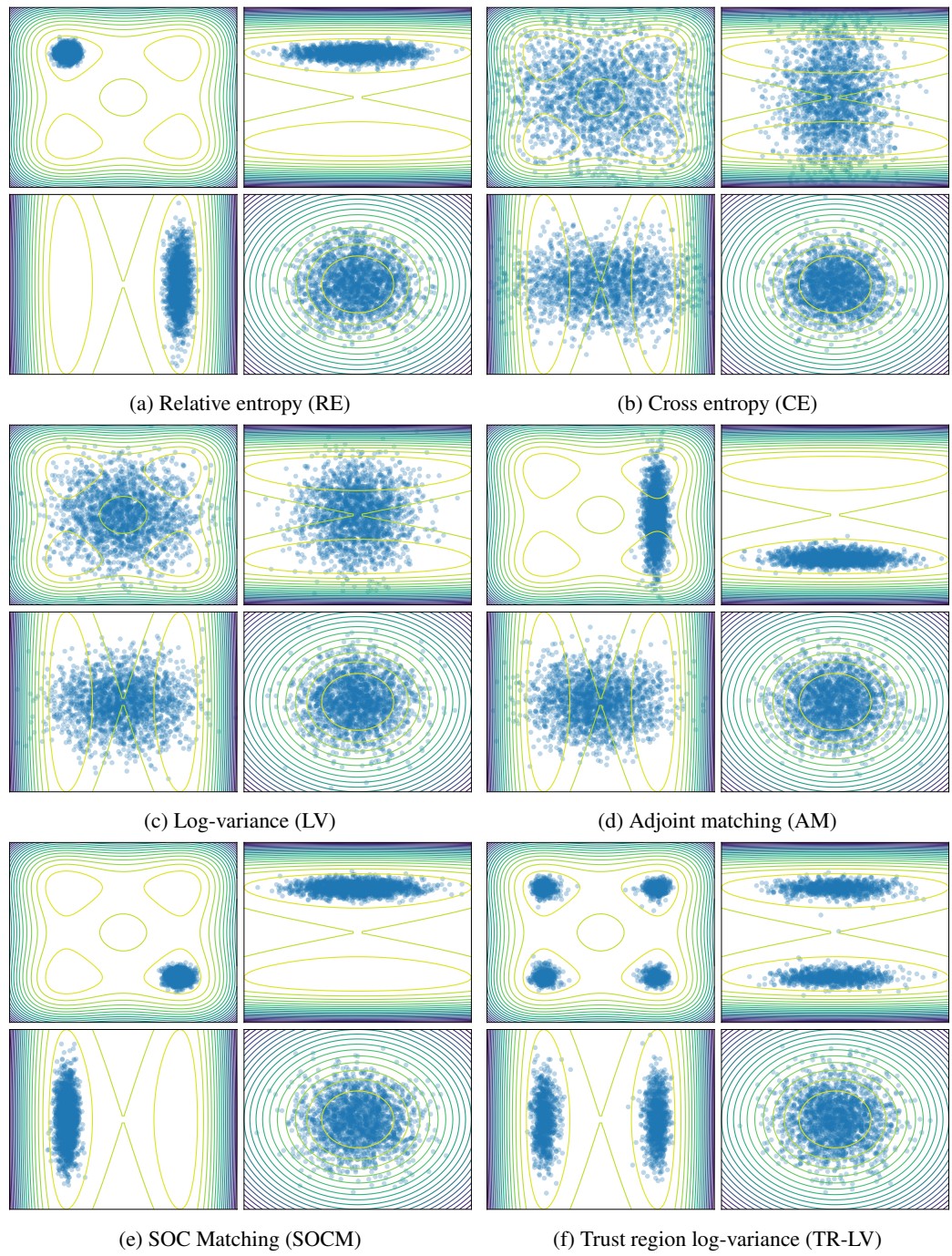

Figure 9: Qualitative results for the *Many Well* target with $d = 200$. Level plots depict the ground truth density for pairs of marginal distributions, while blue dots represent samples generated by models trained using the respective loss functions (indicated in the sub-captions). Among all methods, only the trust-region-based log-variance loss successfully avoids mode collapse and convergence issues. Interestingly, although the cross-entropy loss achieves the second-lowest estimation error for $\log \hat{\mathcal{Z}}$ (see Fig. 3), the qualitative results suggest that the model fails to adequately capture the target distribution – likely due to the high variance of the importance weights. All visualizations are generated using the same random seed for consistency.

a trust-region bound $\varepsilon = 0.1$, a buffer size of 100, and 10 passes per buffer. The dual function is optimized using Brent's method [23].

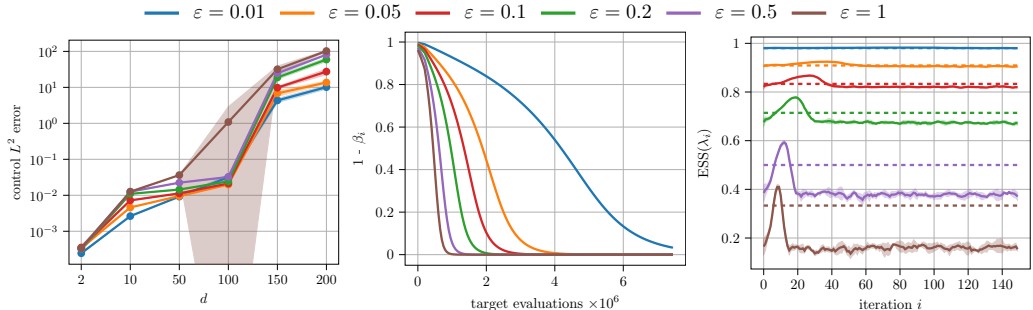

Figure 10: Influence of different trust region bound values $\varepsilon$ on the GMM target for TR-LV. The left figure considers varying problem dimensionalities $d$ whereas the middle and right figure report results for $d = 100$. The figure on the right shows the empirically observed smoothed effective sample size (ESS) and its approximation via Taylor series approximation, i.e., ESS $= \left( \text{Var}\left( \frac{\mathrm{d}\mathbb{P}^{u_{i+1}}}{\mathrm{d}\mathbb{P}^{u_i}} \right) + 1 \right)^{-1} \approx \frac{1}{2\varepsilon+1}$, with solid and dashed lines, respectively. All results are averaged across four random seeds.

We use the 10000 fine-tuning prompts taken from the repository for [140], and the 100 validation prompts from the same repository (see `https://github.com/THUDM/ImageReward`). The two prompts used in Figure 6 are "masterpiece, best quality, realistic photograph, 8k, high detailed vintage motorcycle parked on a wet cobblestone street at dusk, neon reflections, shallow depth of field" and "close up photo of anthropomorphic fox animal dressed in white shirt, fox animal, glasses".

# L    Classical SOC problems

Here, we consider classical SOC problems, for which the optimal control can be computed analytically. These problems have been widely used in recent studies to compare different loss functions [40, 42, 87]. Here, we leverage them to showcase that importance sampling works in high dimensions when using trust-region-based losses. To that end, we consider the comparison between the SOCM loss and its trust-region-based counterpart.

## L.1    Experimental setup

The experimental setup follows the setup used for diffusion-based sampling, as explained in App. I.1, including control function architecture, hyperparameter evaluation protocol, and model selection.

For discretizing the SDE, we leverage the Euler-Maruyama scheme, i.e.,

$$\widehat{X}_{n+1} = \widehat{X}_n + (b + \sigma u)\left(\widehat{X}_n, n\Delta t\right)\Delta t + \sigma(n)\sqrt{\Delta t}\xi_n, \quad \xi_n \sim \mathcal{N}(0, I). \tag{145}$$

Since the considered benchmark problems admit analytical solutions for the optimal control $u^*$, we consider the $L^2$ error between the learned and the optimal control for evaluating the models as explained in App. I.1.

## L.2    Benchmark problem details

We consider two problems taken from [87], the *Quadratic Ornstein-Uhlenbeck (OU) easy* and *Quadratic Ornstein-Uhlenbeck (OU) hard*. For convenience, we briefly introduce them again here.

**Quadratic Ornstein-Uhlenbeck (OU)** The choices for the functions of the control problem are

$$b(x,t) = Ax, \quad f(x,t) = x^\top P x, \quad g(x) = x^\top Q x, \quad \sigma(t) = \sigma_0, \tag{146}$$

where $Q$ is a positive definite matrix. Control problems of this form are better known as linear quadratic regulator (LQR) and they admit a closed form solution [127]. The optimal control is given by

$$u^*(x,t) = -2\sigma_0^\top F(t)x, \tag{147}$$

where $F(t)$ is the solution of the Ricatti equation

$$\frac{\mathrm{d}F(t)}{\mathrm{d}t} + A^\top F(t) + F(t)A - 2F(t)\sigma_0\sigma_0^\top F(t) + P = 0 \tag{148}$$

with the final condition $F(T) = Q$. Within the Quadratic OU class, we consider two settings:

- Easy: We set $A = 0.2I$, $P = 0.2I$, $Q = 0.1I$, $\sigma_0 = I$, $\lambda = 1$, $T = 1$, $x_{\text{init}} \sim 0.5\mathcal{N}(0, I)$.

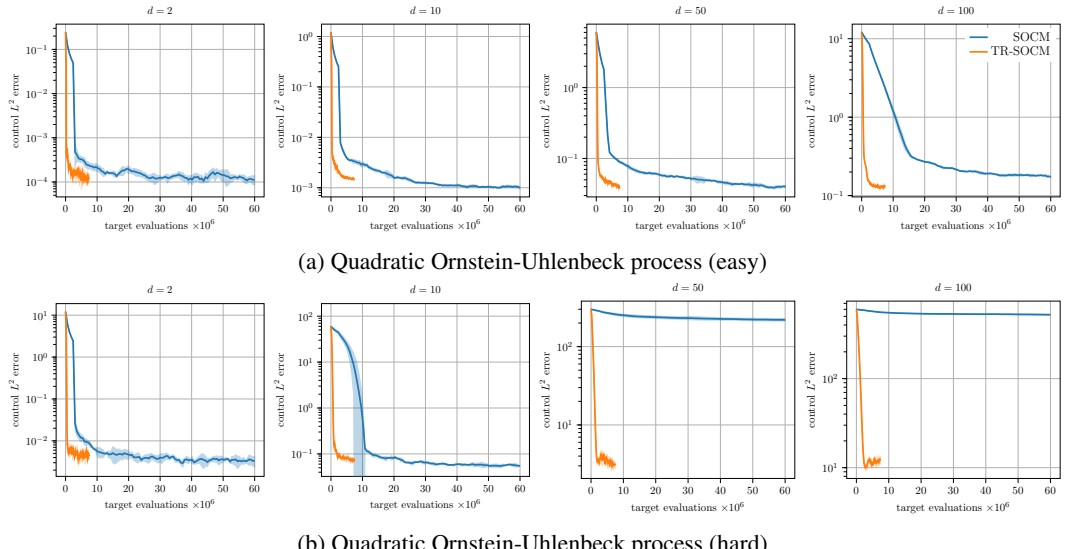

(a) Quadratic Ornstein-Uhlenbeck process (easy)

(b) Quadratic Ornstein-Uhlenbeck process (hard)

Figure 11: Control $L^2$ error as a function of the number of target evaluations for the quadratic OU problem across varying problem dimensionalities $d$. All results are averaged across four random seeds.

- Hard: We set $A = I$, $P = I$, $Q = 0.5I$, $\sigma_0 = I$, $\lambda = 1$, $T = 1$, $x_{\text{init}} \sim 0.5\mathcal{N}(0, I)$.

## L.3 Results

We compare the performance of SOCM and its trust-region-based variant (TR-SOCM) on the quadratic Ornstein–Uhlenbeck (OU) problem across varying problem dimensionalities $d$. Both approaches rely on importance sampling, which is known to be challenging in high-dimensional settings. This experiment highlights the role of trust regions in scaling to such regimes. Results are presented in Fig. 11.

In low-dimensional settings ($d \leq 10$), both methods perform comparably, although TR-SOCM exhibits significantly better sample efficiency. As the dimensionality increases ($d \geq 50$), the performance of SOCM deteriorates markedly, while TR-SOCM continues to achieve low control error. For the more challenging variant of the quadratic OU problem, SOCM fails to meaningfully improve upon its initialization, whereas TR-SOCM demonstrates consistent error reduction.

These results suggest that trust regions are particularly beneficial in high-dimensional and difficult problem settings, where they provide stability and improved performance.

