# OpenReview forum: "Trust Region Constrained Measure Transport in Path Space for Stochastic Optimal Control and Inference"
_NeurIPS.cc/2025/Conference — NeurIPS 2025 spotlight_

### Official Review · Reviewer_TLot · 2025-07-01

**Clarity:** 4
**Significance:** 3
**Originality:** 3
**Rating:** 5
**Confidence:** 2

**Summary:**

This paper proposes a trust region method for stochastic optimal control (SOC) problems by reframing SOC as a measure transport problem in path space. The core idea is to iteratively minimize the KL divergence between controlled path measures while constraining the KL divergence between consecutive iterates within a "trust region". The authors demonstrate that this approach yields a geometric annealing path with equidistant steps in Fisher-Rao distance, which contributes to stable optimization. The method is shown to be efficient across diverse applications, including diffusion-based sampling, molecular transition path sampling, and diffusion model fine-tuning, consistently outperforming baselines with significant gains in stability and sample efficiency.

**Questions:**

Apart from the comments in the weakness section, I have the following questions:
1. Could the authors explain why their proposed methods TR-SOCM and TR-LV are scalable to high dimensions (e.g., in Figure 2)?
2. Have the authors tried shrinking $\epsilon$, which is commonly used in optimization?
3. How is the length of sequence $I$ affect the algorithmic performance?

**Ethical Concerns:**

["NO or VERY MINOR ethics concerns only"]

**Final Justification:**

The authors have cleared my doubts. I believe this work makes a meaningful contribution to the field of stochastic optimal control. Therefore, I raised my score to 5 and recommend acceptance of this paper.

**Limitations:**

Yes.

**Quality:**

3

**Strengths And Weaknesses:**

Strengths
1.  One interesting finding in this work is that it elegantly links trust regions to geometric annealing, resulting in equidistant steps in Fisher-Rao distance. This theoretically grounded approach directly addresses the instability often encountered when prior and target measures differ substantially, by systematically approaching the target measure gradually.
2. The proof in this work appears correct. The authors provide a clear interpretation of each theory with easy-to-digest examples.
3. The true region SOC algorithm (Algorithm 1) is flexible with two compatible loss functionals, log-variance divergence and SOC matching loss, which are designed to work with buffer reuse. This "off-policy" optimization, where the $\lambda$-dual can be fitted from already-simulated trajectories, is a key enabler for enhanced efficiency.

Weaknesses
1. The authors claim "minimal computational overhead" for the dual optimization of Lagrange multipliers. However, empirical comparisons with baselines are only qualitatively sketched, not based on running time. It would be better to see the benefits under the same computational cost.
2. Hyper-parameter guidance on the effect of $\epsilon$ and buffer size is missing, which could critically affect algorithmic performance.

---

> ### Author Rebuttal · Authors · 2025-07-30
>
> We thank the reviewer for taking the time to review our work and for the many helpful comments and suggestions. We hope the following replies address the questions and concerns raised.
>
> ---
> > The authors claim "minimal computational overhead" for the dual optimization of Lagrange multipliers. However, empirical comparisons with baselines are only qualitatively sketched, not based on running time. It would be better to see the benefits under the same computational cost.
>
> We thank the reviewer for the helpful suggestion and have accordingly added quantitative results on running time. Specifically, we evaluate two aspects:
>
> (1) Overhead from dual optimization:
> For the dual optimization step, we employed a line search algorithm based on a bracketing method. In the diffusion-based sampling experiments, for example, computing $\lambda^*$ from 50k importance weights took approximately 0.014 seconds per optimization (averaged over 10k trials on an RTX 3060 GPU). Performing 150 such updates during training, the total overhead from dual optimization amounts to only 2.1 seconds, which is negligible relative to the overall training time.
>
> (2) Impact of the replay buffer in fine-tuning:
> We also compared TR-LV, which uses a buffer, against AM (which does not use a buffer). We observed a significant improvement in efficiency when using a buffer:
> – Without a buffer, fine-tuning experiments averaged 1 day and 23 minutes.
> – With a buffer, the same experiments required only 11 hours and 9 minutes on average.
> This clearly demonstrates that incorporating a replay buffer can substantially reduce computational cost in practice.
>
> We thank the reviewer for this helpful suggestion and will include the corresponding runtime comparisons in the revised manuscript.
>
> ---
> > Hyper-parameter guidance on the effect of $\varepsilon$ and buffer size is missing, which could critically affect algorithmic performance.
>
> We appreciate the reviewer’s observation regarding the importance of hyperparameter guidance for both the trust-region bound $\varepsilon$ and the buffer size.
>
> Trust-region bound $\varepsilon$:
> We discuss the role of $\varepsilon$ in Appendix E.3, “Influence of Trust Region Bounds”, where we highlight its connection to the variance of the importance weights and the resulting effective sample size. This theoretical relationship often provides an intuitive basis for selecting $\varepsilon$. In many of our experiments, we used $\varepsilon = 0.1$, which corresponds approximately to an effective sample size of 0.8 and showed good empirical performance.
>
> Buffer size:
> For transition path sampling, we adopted buffer sizes consistent with prior work. In the fine-tuning experiment, we evaluated several buffer sizes and found that a size of 100 offered a good balance between performance and computational efficiency, as larger buffer sizes did not yield significant improvements.
>
> In contrast, for diffusion-based sampling, we observed that buffer size has a substantial impact on performance. To explore this further, we conducted an additional experiment on the GMM task using the TR-LV loss, varying both the buffer size and dimensionality. We report the control L2 error in the table below:
> | Buffer size    | $d=2$ | $d=50$ | $d=100$ | $d=200$ |
> |----------|----------|----------|----------|----------|
> | $10k$    | $0.001 \scriptstyle \pm 0.000$   | $7.721 \scriptstyle \pm 0.051$   | $21.656 \scriptstyle \pm 1.504$   | $70.606 \scriptstyle \pm 1.886$   |
> | $25k$    | $0.001 \scriptstyle \pm 0.000$   | $3.507 \scriptstyle \pm 0.995$   | $4.313 \scriptstyle \pm 0.634$   | $59.814 \scriptstyle \pm 3.83$   |
> | $50k$    | $0.000 \scriptstyle \pm 0.000$   | $0.014 \scriptstyle \pm 0.000$   | $0.026 \scriptstyle \pm 0.002$   | $24.178 \scriptstyle \pm 3.040$   |
> | $100k$    | $0.000 \scriptstyle \pm 0.000$   | $0.014 \scriptstyle \pm 0.000$   | $0.030 \scriptstyle \pm 0.001$   | $11.594 \scriptstyle \pm 4.157$   |
>
> The results show that larger buffer sizes consistently lead to lower control error, particularly as dimensionality increases.
>
> We thank the reviewer for this valuable suggestion. We will incorporate these results and practical guidelines into the revised manuscript.
>
>
> ---
> > Could the authors explain why their proposed methods TR-SOCM and TR-LV are scalable to high dimensions (e.g., in Figure 2)?
>
> Both, the SOCM and LV loss do not require differentiation through simulations which makes them more memory efficient and therefore more scalable. For the LV loss we refer to Proposition 5.7 in [1], which shows beneficial scaling properties of this loss in high dimensions. Further, note that the additional use of trust regions allows losses that require importance weighting - such as the SOCM loss - to scale to high-dimensions as well (see e.g. Fig. 11) since the trust region keeps the variance of the importance weights approximately constant (see Figure 10).
>
> However, we agree that the statement about scalability in Figure 2 could potentially be misleading. We, therfore, changed it to "We observe that our trust region methods (TR-SOCM and TR-LV) are the only methods that perform well in high dimensions."
>
> ---
> > Have the authors tried shrinking $\varepsilon$, which is commonly used in optimization?
>
> Shrinking the trust-region bound is indeed a promising idea that could potentially enhance performance. However, to maintain simplicity and avoid introducing additional hyperparameters, such as a shrinking schedule, we opted to use a fixed value for $\varepsilon$ in our current implementation. We acknowledge this as a valuable direction and highlight it as a potential extension for future work. At the same time, however, we want to highlight a tradeoff between a small $\varepsilon$ and a potentially long runtime: the smaller $\varepsilon$, the larger is the number of the overall iterations $I$, which might result in better convergence to the optimal control $u^*$, but at the same time typically increases the overall runtime.
>
>
> ---
> > How is the length of the sequence $I$ affect the algorithmic performance?
>
> The length of the sequence is intimately connected with the choice of the trust-region bound $\varepsilon$: Small trust-region bounds require more iterations to approach the target and therefore larger $I$. Conversely, chosing large values of $\varepsilon$ requires fewer iterations and therefore a smaller $I$.
>
> As such, $I$ represents a trade-off between computational cost and performance. To see this, we kindly refer the reviewer to the supplementary material (Figure 10), where the left subfigure shows consistent performance improvements with smaller $\varepsilon$ values. However, the middle subfigure highlights that these settings require more target evaluations and thus larger $I$ to reach convergence. (For reference, $(\beta_i)_i$ denotes the annealing sequence, where $1 - \beta_i = 0$ indicates that the target distribution has been reached.)
>
> ---
>
> ### References
>
> [1] Nüsken, N., & Richter, L. (2021). Solving high-dimensional Hamilton–Jacobi–Bellman PDEs using neural networks: perspectives from the theory of controlled diffusions and measures on path space. Partial differential equations and applications, 2(4), 48.
>
> ---

---

### Official Review · Reviewer_r2Du · 2025-07-02

**Clarity:** 3
**Significance:** 3
**Originality:** 3
**Rating:** 5
**Confidence:** 1

**Summary:**

In this paper, the authors propose a novel method to solve stochastic optimal control problems. The idea of the paper is to divide the problem into constrained sub-problems, where the constraints maintain the solution of the actual problem closer to the previous one (having a constraint on the KL divergence between the solutions of two adjacent sub-problems). Doing this, the authors can reuse the sampled trajectories of the previous problem to solve the following one, reducing the target evaluations.

**Questions:**

- The approach appears sound and is applied across multiple domains. How sensitive is the method to domain-specific hyperparameters or design choices?

- The experimental results demonstrate sample efficiency on Stable Diffusion, but the performance is similar to the baseline. Do the authors expect their method to outperform the baseline with more samples?

- The proposed approach relies on the use of constraints to improve robustness. Could the authors provide an ablation or analysis showing how different values of the $\epsilon$ parameters influence performance and convergence? Is there a clear trade-off? How to set the parameter?

**Ethical Concerns:**

["NO or VERY MINOR ethics concerns only"]

**Final Justification:**

After reading the rebuttal and discussion, I remain positive about this paper. The authors provided clear and detailed answers to my questions.

**Limitations:**

Yes

**Paper Formatting Concerns:**

No formatting issues.

**Quality:**

3

**Strengths And Weaknesses:**

## Strenghts

- The paper is well written and clearly presented, even for readers who are not experts in SOC problems.

- It proposes a sound and principled approach to solving SOC problems.

- The authors demonstrate the versatility of the method by applying it to several tasks and comparing it against existing approaches.

- I thoroughly enjoyed reading the paper. Although the topic lies outside my area of expertise, I believe the method has potential utility in many AI applications.

## Weaknesses

- In the experiments, it would be great to show the contribution of the constrained in the performance of the method. Since the main idea is that the constraints will help in finding a more robust solution, it would be nice to have a trade-off between the $\epsilon$ parameters and the number of iterations.

- In the experiment on Stable Diffusion, the algorithm achieves similar performance to the baseline but with fewer samples. It is not clear if with more samples the algorithm will achieve better performance than the baseline.

---

> ### Author Rebuttal · Authors · 2025-07-30
>
> We thank the reviewer for taking the time to review our work and for the many helpful comments and suggestions. We hope the following replies address the questions and concerns raised.
>
> ---
> > The approach appears sound and is applied across multiple domains. How sensitive is the method to domain-specific hyperparameters or design choices?
>
> The domain-specific hyperparameters and design choices were largely adopted from the respective baselines. Specifically, for transition path sampling, we followed the settings used in [1]; for fine-tuning, we adopted the configuration from [2]; and for diffusion-based sampling, we partially followed the setup in [3]. We kindly refer the reviewer to Appendices E.1, F.1, and H.1 for further details on these design choices.
>
> Regarding the trust-region bound $\varepsilon$, we find that our method is generally robust across different domains. For all diffusion-based sampling tasks, we consistently used $\varepsilon = 0.1$, which performed well without additional tuning. In the case of transition path sampling and fine-tuning, we observed that smaller trust-region bounds can yield further improvements. We attribute this robustness to the theoretical relationship between $\varepsilon$ and the variance of the importance weights, as discussed in Appendix E.3, “Influence of Trust Region Bounds”. This relationship holds independently of the specific application domain (see also our response to the final question for further details).
>
> ---
> > The experimental results demonstrate sample efficiency on Stable Diffusion, but the performance is similar to the baseline. Do the authors expect their method to outperform the baseline with more samples?
>
> Our primary goal in the fine-tuning experiments was to demonstrate improved sample efficiency, i.e., achieving competitive performance with significantly fewer samples. As noted, our method matches the baseline’s performance while requiring substantially less computational effort.
>
> We are currently uncertain whether additional samples would allow our method to consistently outperform the baseline in terms of final performance, particularly given the strength of the baseline itself. However, since we observe consistent improvements under equal computational budgets in other domains, we plan to further investigate this question in the context of fine-tuning experiments.
>
> ---
> > The proposed approach relies on the use of constraints to improve robustness. Could the authors provide an ablation or analysis showing how different values of the $\varepsilon$ parameters influence performance and convergence? Is there a clear trade-off? How to set the parameter?
>
> We have added such an ablation in Appendix E.3, “Influence of Trust Region Bounds” in the context of diffusion-based sampling. Indeed, in that experiment there is a clear trade-off between computational cost and performance: Smaller trust-region bounds lead to improved performance as shown in the left plot of Figure 10, but take longer to approach the target (see middle plot of Figure 10). We agree that this trade-off deserves more visibility, and have therefore integrated the ablation study in the main part of the manuscript.
>
> To guide the choice of $\varepsilon$, we have further analyzed its effect in Appendix E.3 by examining its relationship to the variance of the importance weights and the corresponding effective sample size. This connection often provides a practical and intuitive way to set $\varepsilon$. In many of our experiments, we used $\varepsilon = 0.1$, which corresponds approximately to an effective sample size of 0.8.
>
> ---
>
> We hope that our response addresses the reviewer’s question, and we would be glad to provide further clarification if needed.
>
> ---
>
> ### References
>
> [1] Seong, Kiyoung, et al. "Transition Path Sampling with Improved Off-Policy Training of Diffusion Path Samplers. ICLR 2025
>
> [2] Domingo-Enrich, Carles, et al. "Adjoint matching: Fine-tuning flow and diffusion generative models with memoryless stochastic optimal control." ICLR 2025
>
> [3] Domingo-Enrich, Carles, et al. "Stochastic optimal control matching." NeurIPS 2024
>
> ---

---

> > ### Comment · Reviewer_r2Du · 2025-08-03
> >
> > I would like to thank you the reviewers for their detailed response.
> > I am happy to maintain my score.

---

### Official Review · Reviewer_31NG · 2025-07-02

**Clarity:** 4
**Significance:** 3
**Originality:** 3
**Rating:** 5
**Confidence:** 3

**Summary:**

Stochastic Optimal Control (SOC) problems have appeared within numerous sampling algorithms within recent years. As solution of SOC problems can be approached by minimizing a statistical divergence in path-measure space, when the starting path measure of the SOC problem (often the one corresponding to the reference process) differs greatly from the optimal path measure, solving the SOC problem in "one shot" can be difficult. This work proposes a trust region approach to SOC, wherein a sequence of SOC problems
$$
u_{i+1} = \underset{u \in \cal U}{\text{argmin}} D_{\rm KL} (\mathbb P^u \mid \mathbb Q) \quad \text{s.t. } D_{\rm KL}(\mathbb P^u \mid \mathbb P^{u_i}) < \varepsilon,
$$
are solved such that the optimal path measure $\mathbb Q$ is approached gradually via a sequence of controls $u_1, \dots, u_I$. The constraint $D_{\rm KL}(\mathbb P^u \mid \mathbb P^{u_i}) < \varepsilon$ ensures that $u_{i+1}$ does not change too drastically from $u_i$, the idea being that it will be more stable to solve $I$ problems of the form above than to solve one problem that attempts to go directly from $\mathbb P^0$ to $\mathbb Q$. The authors show that the sequence of path measures induced by the trust region scheme is the geometric annealing path between $\mathbb P^0$ and $\mathbb Q$ and that the constraint $D_{\rm KL}(\mathbb P^u \mid \mathbb P^{u_i}) < \varepsilon$ provides a way to automatically choose step-sizes such that the waypoints along the path are approximately equidistant in the Fisher-Rao metric. Starting from the constrained problem above, the authors use a Lagrangian approach to derive tractable a expression for the optimal Lagrange multiplier $\lambda_i$, which parametrizes the optimal control $u_i$. As the optimal control is also parametrized by the gradient of the value function, which is a conditional expectation,  the authors propose computational approaches for learning $u_i$ given $\lambda_i$ without computing the value function directly. The numerical examples demonstrate robust performance, particularly in comparison to SOC approaches without trust regions, on a variety of tasks, including sampling from unnormalized densities, transition path sampling, and fine-tuning of diffusion models.

**Questions:**

* The result in Proposition 2.2 is very nice. After reading it and thinking about it for a moment it made sense to me because I know that the geometric mixture can be characterized variationally. I'll use standard probability measures instead of path measures here, but if $\pi_0, \pi_1 \in \mathcal P(\mathbb R^d)$, then $\pi_t \propto \pi_0^{1-t}\pi_1^t$ is the minimizer of
$$
\pi_t = \underset{\eta \in \mathcal P(\mathbb R^d)}{\text{argmin}} \quad (1 - t)\mathrm D_{\rm KL}(\eta \| \pi_0) + t \mathrm D_{\rm KL}(\eta \| \pi_1), \quad t \in [0,1].
$$
This result can be found in, e.g., [Amari 2016]. So, given this characterization, it makes sense that requiring equispaced movement in KL would land one on the geometric annealing path. This is more of a comment than a question, but perhaps a result like this holds for path measures as well and could be connected to your result in Proposition 2.2, or be used to arrive at the trust-region method from a different angle?

* Proposition 2.3 reminds me of a similar result in [Chopin et al. 2024], where they show that change in any $f$-divergence controls the change in Fisher Information along the geometric annealing path and use this fact to propose time-stepping schemes based on Fisher Information (Section 3.1/Proposition 2). Your result is applicable to path measures as well, but it is probably worth citing [Chopin et al. 2024] to make the connection.

* This is minor, but in Algorithm 1 should $i$ be incremented on every iteration? And should $u^{i+1}$ be returned instead of $u_i$? I'm a little bit confused by the indexing.

### References
* Amari, S. I. (2016). Information geometry and its applications (Vol. 194). Springer.
* Chopin, N., Crucinio, F. &amp; Korba, A.. (2024). A connection between Tempering and Entropic Mirror Descent. <i>Proceedings of the 41st International Conference on Machine Learning</i>, in <i>Proceedings of Machine Learning Research</i> 235:8782-8800 Available from https://proceedings.mlr.press/v235/chopin24a.html.

**Ethical Concerns:**

["NO or VERY MINOR ethics concerns only"]

**Final Justification:**

This paper presents a compelling new algorithmic approach for solving SOC problems grounded in rigorous, interesting mathematics and justified by impressive empirical performance across a range of challenging benchmark problems. The two clarity issues I flagged have been resolved, and I feel that the discussion which has arisen out of the rebuttal discussions will further strengthen the final version of the paper. I therefore recommend this paper for acceptance in NeurIPS 2025.

**Limitations:**

Yes

**Quality:**

4

**Strengths And Weaknesses:**

## Strengths
* The paper is clear and well-written. I enjoyed reading it and learned a lot in the process.

* While geometric annealing is a common design choice in algorithms for sampling from unnormalized densities (SMC, recent transport-based approaches, etc.), this is the first time I've encountered it for path measures. Moreover, one of the most critical components of using geometric annealing is effectively picking the annealing schedule. The trust-region-derived objective that is optimized to pick each $\lambda$ here is nice and I imagine it could inform other methods which use geometric annealing.

* The derivations of the methods are rigorous and elegant.

* The trust region method proposed can be applied to any algorithm which uses SOC, including many recent sampling SOC-based sampling methods.

* The numerical examples are challenging and span three different problem types (sampling, transition-path sampling, and diffusion model fine-tuning). The benefit of using the trust region SOC approach over regular SOC is clear from the numerical examples, in that it often leads to better performance and requires fewer target evaluations/energy evaluations/trajectories.

## Weaknesses
* The explanation of how one goes from the cross-entropy loss to the SOCM loss is very brief and relies on prior knowledge of adjoint-matching. I know that the details are in the appendix, but building a little more intuition for readers who may be unfamiliar with adjoint-matching would help with readability.

---

> ### Author Rebuttal · Authors · 2025-07-30
>
> We thank the reviewer for taking the time to review our work and for the many helpful comments and suggestions. We hope the following replies address the questions and concerns raised.
>
> ---
> > [...] a result like this holds for path measures as well and could be connected to your result in Proposition 2.2, or be used to arrive at the trust-region method from a different angle?
>
> Indeed, Proposition 2.2 can also be seen as an instance of the variational characterization you mention with the key difference that $\pi_0$ is replaced with $\pi_i$. To make this more explicit, consider the following optimization objective, which shares the same critical points in $u$ as $\mathcal{L}\_{\mathrm{TR}}^{(i)}(u, \lambda_i)$ (Eq. (5)):
> $$
> \mathcal{L}(u) = \tfrac{1}{1+\lambda_i}D\_{\text{KL}}\left(\mathbb{P}^u | \mathbb{Q} \right) + \tfrac{\lambda_i}{1+\lambda_i} D\_{\text{KL}}(\mathbb{P}^u | \mathbb{P}^{u_i} ).
> $$
> Next, defining $\alpha_i = \tfrac{1}{1+\lambda_i}$, we arrive at
> $$
> \mathcal{L}(u) = \alpha_i D\_{\text{KL}}\left(\mathbb{P}^u | \mathbb{Q} \right) + (1-\alpha_i) D\_{\text{KL}}(\mathbb{P}^u | \mathbb{P}^{u_i} ).
> $$
> Solving $u_{i+1} = \text{argmin}\_u \mathcal{L}(u)$ gives
> $$
> \frac{\mathrm d\mathbb{P}^{u_{i+1}}}{\mathrm  d\mathbb{P}} \propto \left(\frac{\mathrm  d\mathbb{P}^{u_{i}}}{\mathrm  d\mathbb{P}}\right)^{1-\alpha_i}\left(\frac{\mathrm  d\mathbb{Q}}{\mathrm  d\mathbb{P}}\right)^{\alpha_i} \quad \iff \quad \frac{\mathrm  d\mathbb{P}^{u_{i+1}}}{\mathrm  d\mathbb{P}^{u_i}} \propto \left(\frac{\mathrm  d\mathbb{Q}}{\mathrm  d\mathbb{P}^{u_i}}\right)^{\alpha_i}
> $$
> which is equivalent to Eq.(7) in Proposition 2.2.
>
> We thank the reviewer for highlighting this connection. We will add additional details and references to the manuscript.
>
> ---
> > Proposition 2.3 reminds me of a similar result in [Chopin et al. 2024] [...]
>
> We agree that the work of Chopin et al. (2024) is highly relevant to our approach. In particular, the connection between geometric annealing and entropic mirror descent is intriguing. We are currently investigating whether similar connections can be established in the context of path measures, which we believe could offer valuable theoretical insights.
>
> ---
> > This is minor, but in Algorithm 1 should be incremented on every iteration? And should $u_{i+1}$ be returned instead of $u_i$?
>
> We thank the reviewer for pointing this out. We have corrected the indexing and incrementing accordingly in the revised manuscript.
>
> ---
> > The explanation of how one goes from the cross-entropy loss to the SOCM loss is very brief and relies on prior knowledge of adjoint-matching.
>
> We agree with the reviewer that the SOCM loss would benefit from further clarification. Accordingly, we have expanded the explanation in the revised manuscript to improve clarity and provide additional intuition.
>
> ---
>
> Once again, we sincerely thank the reviewer for the valuable feedback and would be happy to provide any further clarifications if needed.

---

> > ### Comment · Reviewer_31NG · 2025-08-03
> >
> > I thank the authors for their reply and look forward to reading the updated manuscript. I will maintain my score, as I have already recommended acceptance of this paper.

---

### Decision · Program_Chairs · 2025-09-17

**Decision:**

Accept (spotlight)

**Comment:**

This paper proposes a method for solving stochastic optimal control problems though a sequence of constrained problems. The idea is to gradually approach the optimal solution though a series of constrained problems, to ensure stability. All reviewers agreed the work is novel, and of high quality and importance.